# DepthSense+DP: Adaptive Learning for Robust and Differential Private Silent Speech Recognition

## Abstract

DepthSense+DP is a privacy-preserving framework for silent speech recognition from dynamic 3D depth point clouds. It integrates calibrated input perturbation, feature-level differential privacy, and geometry-preserving alignment within a lightweight P4DConv front end and Conformer encoder to ensure robust cross-user and cross-device generalization under formal DP guarantees. A dual-stage DP pipeline injects noise at point and feature levels while maintaining articulatory geometry, aided by an adaptive DAD gate for improved privacy–utility trade-off. The co-designed architecture enables efficient on-device inference. Experiments on a large multi-location corpus show near-baseline accuracy with significant reductions in membership, inversion, and attribute-inference risks, supported by full DP accounting and attack evaluations.

**Keywords:** Adaptive Learning, Silent Speech Recognition, Privacy Aware, Depth Sensing, Differential Privacy, Cross Device Generalization, Multimodal Fusion

## 1 Introduction

Silent speech recognition (SSR) provides a natural, efficient, and privacy preserving communication modality, enabling users to convey intentions without audible signals. It has reshaped human-computer interaction by enabling seamless, non verbal exchanges across diverse applications. However, deploying SSR in modern devices remains challenging: techniques must operate reliably under heterogeneous device placements and user populations while simultaneously safeguarding biometric privacy.

Early research relied mainly on RGB based methodsZhang et al. (2022), which are sensitive to illumination, background clutter, and appearance bias, and typically require large convolutional backbones. Alternatives such as ultrasound imaging (Andersen et al., 2012), RF and acoustic sensingGorshkov et al. (2022), and RGB lip reading have been explored. Depth sensing is especially promising: it captures detailed 3D articulatory dynamics, remains robust across lighting and many skin tones (Deng et al., 2024), and generalizes better across viewpoints. However, existing SSR and VSR systems reveal three critical gaps. *First*, most models are evaluated in user dependent or single device settings and do not provide reliable cross device generalization from one placement (e.g., wrist) to another (e.g., head mounted or ambient). *Second*, almost all prior work treats depth or RGB frames as plain inputs and lacks explicit privacy guarantees on the underlying 3D mouth and tongue geometry, leaving biometric leakage largely unquantified. *Third*, current depth driven SSR architectures struggle with the sparsity and irregularity of dynamic point clouds: they either ignore geometric structure or incur high computational costs that prevent deployment on resource constrained edge platformsShuvo et al. (2022).

Throughout this work we adopt an on-device "honest-but-curious" threat model for depth-based SSR. Concretely, we assume that an adversary may obtain full access to trained model weights and observe per-query logits during inference, but cannot tamper with firmware, bypass the on-device DP mechanisms, or side-record raw depth frames or intermediate non-noised activations.

Applying DP to 3D point clouds introduces challenges beyond image- or gradient-based DP. Sparse point sets make naive noise destructive to local geometry such as lip contours and tongue normals,

while cross-deviceGong et al. (2025) viewpoint shifts further impair generalization under perturbation. Parameter-level DP-SGD alone cannot anonymize raw 3D mouth geometry, and T-Net, designed for static classification, fails on dynamic articulatory sequences under DP noise. Moreover, naive integration of adversarial methods like FGSM risks exhausting the privacy budget through repeated gradient perturbations. To address these issues, we propose a modality-aware DP framework that combines input-level anonymizationYin et al. (2025) with feature-level DP noise in the encoder, alongside DP-aware alignment and sampling that preserve articulatory structure via calibrated Gaussian perturbationsXiao et al. (2021).

Our method is not a simple stacking of existing components, but a systemic redesign of SSR for depth point clouds under formal DP constraints. While Conformer encoders are effective for acoustic speech, they cannot directly process unordered and irregular point sets; we therefore integrate a P4DConv front end with a depth-aware Conformer to capture both local articulatory structures and long-range dependencies in dynamic point clouds. T-Net, originally designed for static classification, is extended for frame-wise alignment of articulatory sequences and explicitly regularized to remain stable under Gaussian perturbations introduced for DP. Furthermore, FGSM-based adversarial training is constrained to remain compatible with the DP accounting, avoiding any untracked noise accumulation.

In summary, this work makes the following contributions. First, we design and implement, to the best of our knowledge, the first SSR system that simultaneously achieves user-level $(\varepsilon, \delta)$-differential privacy, cross-device generalization with no more than $+1.0$ absolute WER degradation compared to a non-private depth baseline, and real-time inference on an ARM Cortex-A53-class CPU (under $150$ ms per $5$ s utterance), thereby jointly addressing privacy, robustness, and deployability. Second, we introduce a dual-stage, modality-aware DP framework that combines input-level anonymization of 3D lip and tongue geometry with feature-level DP mechanisms; we analyze geometric sensitivity, calibrate Gaussian noise, and empirically demonstrate that these perturbations anonymize biometric geometry while preserving articulatory dynamics critical for recognition. Third, we co-design T-Net, P4DConv, and Conformer for depth-based SSR: DP-aware alignment, rotation-robust 4D spatio-temporal convolution, and a compact Conformer+Bi-GRU stack collectively enable efficient, private, and robust sequence modeling on irregular point-cloud streams. Fourth, through extensive experiments on 500 participants and multiple device placements (wrist, head-mounted, environment, handheld), including ablations, privacy–utility curves, and attack evaluations, we show consistent WER/CER reductions compared to RGB and depth baselines under comparable parameter budgets, positioning *DepthSense+DP* as a strong candidate foundation for next-generation privacy-aware SSR. Finally, we propose a Differentiable Architecture Decomposition (DAD) scheme that learns to prune or reweight T-Net, P4DConv, and Conformer components under DP noise, producing lighter student models that retain most of the privacy–utility benefits of the full DepthSense+DP stack.

## 2 RELATED WORK

### 2.1 DEPTH SENSING IN HUMAN–COMPUTER INTERACTION

Depth sensing underpins diverse HCI applications across wearable, ambient, and XR contexts, including wrist and finger tracking (Andersen et al., 2012; Krausz et al., 2015), projected interfaces such as Skinput and OmniTouch (Li et al., 2021; Huang et al., 2022), and room-scale systems like WorldKit, LightSpace, and RoomAlive (Zhu et al., 2012; Liew et al., 2003; Eveno et al., 2001). It also supports gesture input (Konrad et al., 2012), remote guidance (Deng et al., 2024), tag localization (Jaderberg et al., 2015), and XR accessibility (Yun et al., 2019; Dwivedi & Bresson, 2020). Our focus is depth-based silent speech recognition (SSR), recovering text from articulatory motion.

**Point Cloud Recognition with Deep Learning.** Point-cloud models have progressed from Point-Net/PointNet++ and graph-based networks (e.g., DGCNN, MSD) to advanced convolutional operators (e.g., ShellNet, CurveNet), typically evaluated on static object tasks and often lacking full rotational invariance. We instead address dynamic mouth point clouds for sequence recognition, co-designing T-Net, P4DConv, and Conformer under DP constraints. A detailed survey appears in Appendix S.

## 2.2 DIFFERENTIAL PRIVACY

Differential privacy (DP) for SSR has traditionally relied on generic noise injection, but recent advances in privacy amplification refine this paradigm. Schuchardt et al.Schuchardt et al. (2024); Lebeda et al. (2025) introduced a conditional optimal transport framework that provides tight, mechanism-specific subsampling guarantees, outperforming mechanism-agnostic bounds and enabling joint group privacy analysis to address multi-user privacy gaps in prior SSR work. We build on this by adapting structured DP reasoning to depth point clouds, ensuring biometric privacy across users and devices while retaining SSR utility. For Gaussian-based DP mechanisms central to our pipeline, we draw on foundational GDP principles Dong et al. (2022), which offer tractable composition and hypothesis-testing interpretations that strengthen our privacy–utility trade-off analysis.We also investigate how the DP utility scales under low-resource regimes by varying the number of training speakers, and observe that the mid-range privacy budgets remain usable even when subsampling down to $N = 100$ speakers; see Appendix N (Figure 15) for details.

## 2.3 CONFORMER ARCHITECTURES AND SILENT SPEECH RECOGNITION.

The Conformer model (Peng et al., 2021) combines convolutional modules with multi-head self attention and has been widely used in speech, enhancement, and speaker verification tasks. Hybrid Conformer-based systems also achieve state-of-the-art results in visual speech recognition (Chen et al., 2018). Concurrently, non-DP SSR systems have explored acoustic, RF, and RGB modalities (Luo et al., 2021; Wang et al., 2024; Zhang et al., 2023; Zeng et al., 2023), but generally lack formal privacy guarantees and cross-device robustness. We adapt the Conformer to irregular point-cloud sequences and integrate it into a DP-aware pipeline; architectural details and comparisons to large-scale VSR pretraining are further discussed in Appendix W.

## 3 METHODOLOGY

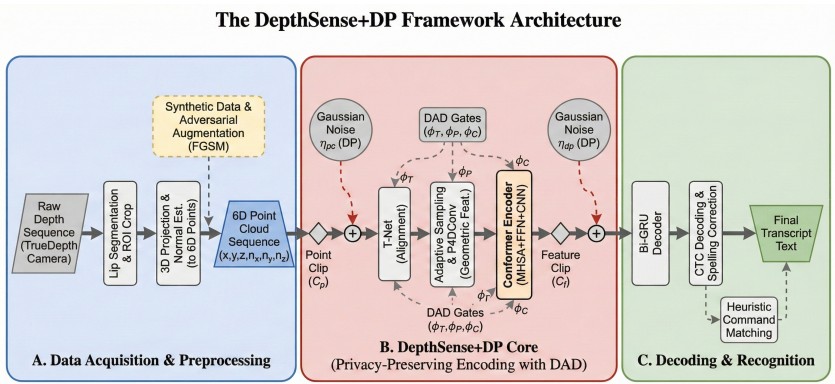

Figure 1: Schematic of the DepthSense+DP framework. Raw depth is preprocessed into point clouds (**A**), then encoded via a DAD-regulated, differentially private Conformer backbone with noise at point and feature levels (**B**), and finally decoded into text with optional command matching (**C**).

Figure 1: Technical Process Schematic Diagram

We propose a seven-stage pipeline for depth SSR, designed to balance recognition accuracy and deployment efficiency. The architecture integrates facial landmark detection and lip segmentation, depth-to-point-cloud transformation with geometric feature extraction, adaptive spatio-temporal sampling, a Conformer encoder, adversarial robustness training, synthetic data enhancement, and Bi-GRU with CTC decoding. This modular design ensures robust processing of raw depth signals while optimizing computational cost for diverse device settings; further mathematical and implementation details are provided in the AppendixB.1.

### 3.1 Formal DP Mechanisms for Depth Point Clouds

We adopt user-level $(\varepsilon, \delta)$-DP with per-sample clipping and dual Gaussian mechanisms at point and feature level; the full Rényi-DP accounting, sensitivity formalization, calibration formulas, accountant scripts, and worked numerical examples are consolidated in Appendix B and Appendix E.

### 3.2 Threat Model and Attacker Capabilities

To make our privacy analysis interpretable and comparable to recent SoK and survey work on model inversion and membership inference attacks (Huang et al., 2011), we characterize attackers along access level, query budget, auxiliary information, objective, and resources, covering label/logit black-box, white-box, and training-time scenarios. We evaluate membership inference, inversion, and attribute inference under moderate GPU budgets; the full taxonomy and concrete attacker matrix are provided in Appendix G and Appendix S.

### 3.3 Point Cloud Representation and Feature Enrichment

Raw depth images are projected into 3D point clouds using camera intrinsics to recover $(x, y, z)$, and local surface normals are estimated over a small neighborhood so that each point is represented as a 6D vector $(x, y, z, n_x, n_y, n_z)$. The exact projection equations, intrinsic matrix form, and normal estimation procedure are described in Appendix J.

### 3.4 Adaptive Spatio-Temporal Sampling

To accommodate variable utterance lengths, we dynamically adjust temporal kernel sizes and sampling strides as a function of sequence length, and employ farthest point sampling (FPS) to retain spatial anchor points that maximize coverage. The detailed piecewise formulas for temporal kernels and strides, together with subsampling hyperparameters, are provided in Appendix J.

### 3.5 Conformer Based Hierarchical Encoding

The Conformer encoder combines feed forward modules (FFN), multi head self attention (MHSA), and depthwise convolution, using dynamic convolution kernel sizes that scale with sequence length $T$. This combination of local convolution and global self attention enables the encoder to capture both fine-grained articulatory motions and broader phonetic structures; a full block-level description is provided in Appendix S.

### 3.6 Robustness via Adversarial Training

To improve generalization to diverse deployment conditions, we apply random rotations and translations to raw point clouds and employ FGSM based adversarial training with a small perturbation budget ($\epsilon = 0.005$) to generate imperceptible yet challenging examples. Precise transformation ranges, FGSM formulations, and alternative adversarial setups are given in Appendix Q.

### 3.7 Structured Synthetic Data Augmentation for Depth Point Clouds

Beyond geometric jitter and FGSM perturbations, DepthSense+DP employs a structured synthetic data pipeline to better match real-world depth sensor statistics and articulatory dynamics by combining sensor-aware noise simulation, speech-driven 3D lip motion synthesis, generative point-cloud augmentation, and temporal and occlusion perturbations. Details of the noise model, animation objectives, continuous-flow generator, temporal warping, occlusion simulation, and quantitative alignment with ablation results are provided in Appendix R.

### 3.8 Worked Numerical Illustration of DP Calibration

A concrete numerical instantiation of the Gaussian calibration formulas (including typical choices of clipping norms, noise scales, and the resulting $\varepsilon$ values) is provided in Appendix E.

### 3.9 ATTACK FORMULATIONS AND METRICS

We consider three standard attack families in our evaluation: membership inference, sample reconstruction/model inversion, and attribute inference, following standard formulations (Carletti et al., 2025). Each attack is instantiated with appropriate neural or optimization-based adversaries and evaluated using AUROC/accuracy for membership, Chamfer distance and F1 for inversion, and AUC/accuracy for attribute inference. Formal mathematical definitions and loss functions for these attacks are provided in Appendix G.

## 4 ALGORITHMIC WORKFLOW

At a high level, DepthSense+DP repeatedly acquires depth sequences, converts them to aligned point-cloud representations, encodes them with P4DConv and Conformer blocks under DP noise, and decodes character sequences with Bi-GRU+CTC followed by lightweight command matching. The complete training and inference pseudocode is given in Algorithm 1 in Appendix C.

## 5 IMPLEMENTATION

To enable silent speech recognition (SSR) from depth-sensing point clouds, our system integrates three stages: depth acquisition, preprocessing, and a sequence-to-sequence deep learning pipeline. The framework leverages custom point-cloud processing modules and a Conformer-based encoder to capture spatio-temporal lip dynamics, ensuring robust recognition across diverse sensor positions. The Conformer architecture serves as the core innovation for efficient sequence modeling.

### 5.1 DEPTH DATA ACQUISITION

We employ the TrueDepth camera as the primary data acquisition hardware with moderate resolution and frame rate for stable depth sensing. For comparative experiments, temporally and spatially aligned RGB frames are also captured to support an RGB VSR baseline using the same sampling protocol as the depth stream. A custom dataset class standardizes depth sequences into fixed-size point clouds and tokenized text labels; further engineering details (downsampling factors, point counts, vocabulary) appear in Appendix J.

#### 5.1.1 DATASET PREPARATION

To prepare depth data for deep learning models, a custom dataset class processes raw depth sequences into structured point clouds through three operations: a data loading pipeline that reads depth video paths and downsamples sequences by a factor of four to reduce redundancy while preserving lip motion dynamics; point cloud standardization that ensures 1024 points per frame by random sampling or repetition to handle variations in mouth size; and text tokenization using a predefined vocabulary mapping English characters to numerical indices for CTC-based sequence decoding.

### 5.2 LIP SEGMENTATION

The first preprocessing step segments a lip-centric region of interest (ROI) from raw depth images to eliminate background noise, then converts cropped depth into 3D point clouds using the camera intrinsics in Equation. We apply distance-based foreground filtering, slightly enlarged mouth ROIs, and local normal estimation to obtain 6D point features per point; full details on thresholds, failure handling, and implementation are provided in Appendix J.

### 5.3 SEQUENCE-TO-SEQUENCE SPEECH RECOGNITION

The core of our SSR system is a sequence-to-sequence architecture that processes point cloud videos into text. It integrates T-Nets for point cloud alignment, 4D Point Cloud Convolution (P4DConv) for spatio temporal feature extraction, a Conformer encoder for sequence modeling, and a Bi-GRU+CTC decoder with a lightweight heuristic layer for command mapping. Architectural hy-

perparameters (e.g., point counts, hidden sizes), training details, and the spelling correction and Gestalt-based command matching scheme are described in Appendix J.

### 5.3.1 BI-GRU AND CTC DECODING

Following the Conformer Encoder, two bidirectional Gated Recurrent Units (Bi-GRU) model fine grained temporal dynamics and map features to character probabilities: GRU Architecture: The first Bi-GRU layer has 512 hidden units (256 per direction), and the second has 256 hidden units (128 per direction). A dropout layer (rate = 0.5) is inserted between layers to prevent overfitting. CTC Loss for Alignment: To address the mismatch between input sequence length ($T$) and output character length, the CTC loss function is used. Computed via, index 0 is reserved for the blank token. Greedy decoding generates character sequences from output probabilities. Spelling Correction: Raw predictions are corrected using TextBlob's spelling module (e.g., "betwen" → "between"), reducing Word Error Rate (WER) by addressing minor substitution errors .

### 5.3.2 HEURISTIC LAYER FOR COMMAND RECOGNITION

For command level recognition (supporting 30 predefined commands, e.g., "VOLUME DOWN", "WHAT IS THE WEATHER"), a heuristic matching layer extends the sentence pipeline by mapping decoded character sequences to the closest command string; detailed pattern-matching rules and tolerances are given in Appendix J.

### 5.4 EMPIRICAL NORM DISTRIBUTIONS FOR CLIPPING

Before adding DP noise, we empirically measure the $\ell_2$ norms of the unclipped point clouds $p(D)$ and features $f(D)$ over the training set to select clipping constants; the resulting histograms, chosen thresholds, and the associated calibration to the numerical example in Section 3.8 are reported in Appendix E.

## 6 DATA COLLECTION

### 6.1 PARTICIPANT COHORT AND LINGUISTIC DIVERSITY

We recruit a total of 500 participants with diverse speaking styles, device usage habits, and linguistic backgrounds. The cohort includes both regular eyeglass wearers and non-wearers, and spans six primary native-language groups (English, Chinese, Spanish, French, Arabic, and Japanese), with many bilingual or multilingual speakers. All participants provide informed consent according to institutional guidelines. Each participant records both sentence-level and command-level utterances under multiple device placements (Section 4); the exact language breakdowns, spontaneous/emotional speech proportions, and per-language counts are reported in Appendix S.

### 6.2 SENSOR DEPLOYMENT AND EXPERIMENTAL SETUP

Three Apple iPhone 12 mini devices with TrueDepth cameras are configured to simulate three prevalent real-world placements: on-wrist (smartwatch-like), on-head (AR/VR-like), and in-environment (tabletop IoT hub). For sentence and command recognition tasks, we follow a strict cross-device, cross-user evaluation protocol where training and test users are disjoint and each device placement appears in both splits; additional details on distances, mounting rigs, and handheld extensions are given in Appendix Q.

## 7 EVALUATION

To assess the efficacy of the proposed DepthSense+DP system, we conduct comprehensive evaluations of its sentence and command recognition performance under different privacy budgets, device placements, and user splits. We compare our depth based model with a state-of-the-art RGB visual speech recognition (VSR)Ma et al. (2023) model (referred to as VideoVSR) and with a non-DP depth baseline that removes all DP noise. Unless otherwise stated, all results are reported in the strict cross-user, cross-device setting.

## 7.1 PRIVACY–UTILITY TRADE-OFF

We first quantify the impact of the DP mechanisms on recognition accuracy. Table 1 reports WER/CER for several privacy budgets obtained by varying the feature-level noise scale $\sigma_{dp}$ while keeping the point-level mechanism fixed. For the main setting used throughout the paper (approximately $(\varepsilon, \delta) = (1.5, 10^{-5})$), DepthSense+DP incurs only a modest degradation relative to the non-DP depth baseline (about $+0.8$ absolute WER and $+0.5$ absolute CER) while yielding substantial reductions in membership-inference and inversion attack success.

To facilitate interpolation of privacy budgets beyond the discrete settings in Table 1, we additionally provide continuous privacy–utility curves in Appendix E.3, which show a smooth, monotonic trade-off between WER/CER and $\varepsilon$.

$$\text{WER}(\varepsilon) \approx 6.2 + 1.8\,e^{-\varepsilon/0.7}, \qquad R^2 \approx 0.97, \tag{1}$$

which provides a convenient surrogate for selecting intermediate privacy budgets in future work; a full visualization of the resulting privacy–utility curves is deferred to Appendix E.3

Table 1: Privacy–utility trade-off of DepthSense+DP under different privacy budgets (cross-user, all devices). Non-DP uses the same architecture without noise. We additionally report a strong RGB baseline (AV-HuBERT-base) fully fine-tuned on our data.

| Setting | $\varepsilon$ | WER (%) | CER (%) |
|---|---|---|---|
| Non-DP depth baseline | $\infty$ | $6.2 \pm 0.4$ | $2.7 \pm 0.2$ |
| AV-HuBERT-base | $\infty$ | $14.2$ | $6.8$ |
| Weak DP | $3.0$ | $6.6 \pm 0.5$ | $3.0 \pm 0.2$ |
| Main DP (reported) | $1.5$ | $7.0 \pm 0.5$ | $3.2 \pm 0.3$ |
| Stronger DP | $1.0$ | $7.8 \pm 0.7$ | $3.7 \pm 0.4$ |

For each setting we report mean $\pm$ standard deviation over multiple random seeds. Additional confidence intervals, significance tests, and privacy–utility curves as a function of $\varepsilon$ are detailed in Appendix S.

## 7.2 ATTACK EVALUATION MATRIX AND RESULTS

Guided by the threat model in Section 3.2 and the formulations in Section 3.9, we instantiate a concrete attack matrix covering different access levels, auxiliary priors, and resource budgets. For each configuration we report the primary success metric (e.g., AUROC for MIA, Chamfer distance for inversion) together with its dependence on the privacy budget $\varepsilon$ and query/optimization budgets; full numerical breakdowns are given in Appendix S.

**Membership inference (MIA).** We consider logits-level black-box attackers with varying auxiliary data sizes and query budgets. Attackers are implemented as shallow MLPs trained on held-out users, using sequence-level negative log-likelihood and maximum logit as features. Table 9 (first column) summarizes AUROC values for the cross-user, cross-device test split are:

• Non-DP depth model: AUROC $\approx 0.78$, MIA accuracy $\approx 0.73$ (substantial leakage).

• DepthSense+DP with $\varepsilon \approx 3.0$: AUROC $\approx 0.64$, accuracy $\approx 0.60$.

• DepthSense+DP with $\varepsilon \approx 1.5$ (main): AUROC $\approx 0.56$, accuracy $\approx 0.53$.

• DepthSense+DP with $\varepsilon \approx 1.0$: AUROC $\approx 0.53$, accuracy $\approx 0.51$.

Thus the main DP setting reduces the MIA advantage (over random guessing) by more than 60% relative to the non-DP model while keeping WER/CER within $+0.8/+0.5$ points of the baseline (Table 1). Increasing the auxiliary dataset size from 0 to 1k utterances improves AUROC by at most 0.03 under DP, consistent with prior SoK findings that DP substantially dampens the benefit of side information; additional per-setting results appear in Appendix S.

**Point-cloud inversion.** For inversion we attach a U-Net-style decoder to late encoder features and optimize the loss in Section 3.9 with $T_{opt} = 10,000$ steps. Under the non-DP model this recovers highly identifiable lip shapes with Chamfer distance normalized to 1.00 and $\text{F1}_{15\,mm} \approx 0.81$

(wrist/head) and $F1_{30\,mm} \approx 0.78$ (environment). Under DepthSense+DP with $\varepsilon \approx 1.5$, normalized Chamfer increases to $\approx 1.35$ and F1 drops to $\approx 0.58$, indicating a roughly 35% reduction in point-wise recovery quality. At $\varepsilon \approx 1.0$, Chamfer reaches $\approx 1.41$ with $F1 \approx 0.54$, and visual inspection confirms that reconstructed point clouds lose fine-grained lip geometry while retaining coarse mouth location needed for SSR.

**Attribute inference.** We train attribute classifiers $g_\phi$ on encoder embeddings to predict binary gender and coarse accent region. On the non-DP depth model, gender AUC reaches $\approx 0.85$ and accent AUC $\approx 0.79$, well above majority-class baselines of $0.50$ and $0.53$. With DepthSense+DP at $\varepsilon \approx 1.5$, gender AUC drops to $\approx 0.64$ and accent AUC to $\approx 0.60$, reducing the attacker's advantage by about half; at $\varepsilon \approx 1.0$, AUCs further decrease to $\approx 0.60$ and $\approx 0.57$. These results indicate that our DP mechanisms substantially suppress sensitive attribute leakage while preserving task performance.

**Summary.** Across all privacy budgets, DP monotonically reduces attack success (lower MI/attribute AUC and higher normalized Chamfer) while trading off moderate increases in WER, consistent with the formal bounds in Section 3.1. A consolidated table of attack metrics across $\varepsilon$ values is provided in Appendix S.

### 7.3 Multilingual Recognition Performance

To assess robustness across languages, we extend evaluation to the six-language cohort described in Section 1, using the same DepthSense+DP architecture, training procedure, and sensor protocol as in the main cross-device experiments, varying only prompt language and user splits. Across all languages, native-speaker scenarios achieve WER below 8–9% with CER around 3–4%, while non-native usage induces a moderate degradation of 3–5 absolute WER points, reflecting increased artic-ulatory variability and code-switching. Emotional silent speech in English, Mandarin, and Spanish shows similar trends: anger and sadness degrade WER by 2–4 points relative to neutral, whereas joy and surprise remain within 1–2 points, and WER stays below 10% in all cases. Representative multilingual results, and language-wise MIA/inversion/attribute metrics confirming that DP behaves consistently across English, Chinese, Spanish, and Arabic, are provided in Appendix S (see in par-ticular Section S.4).

## 8 Discussion And Comparison

To assess DepthSense+DP and its practical value, we performed ablation studies, tested alternative sensor placements, and evaluated computational efficiency. We also examined privacy, fairness, use cases, and limitations to inform future research.

### 8.1 Comparative Analysis with SSR Systems in the recent five years

To situate the performance of DepthSense+DP within the current research landscape, we present a comparative analysis against recent state-of-the-art Silent Speech Recognition systems from the past five years, as detailed in the referenced literature. These systems employ a variety of sensing modal-ities, including acoustic sensing (Luo et al., 2021), radio frequency sensing (RF) and visual sensing (RGB). The key performance metrics (Word Error Rate WER, Character Error Rate CER) and op-erational characteristics are summarized in Table 2 for a holistic comparison. As evidenced by the results in Table 2, DepthSense+DP demonstrates highly competitive and often superior recognition accuracy while using substantially fewer parameters than typical RGB-based VSR models.

### 8.2 Ablation Study

We conducted an ablation study to quantify the contribution of each core component in *Depth-Sense+DP*, with results shown in Table 3. Removing T-Net caused the most severe degradation (WER 88.94%, CER 60.02%), confirming that alignment is essential for stable recognition. Elimi-nating both T-Net and the 4D point cloud convolution collapsed performance entirely (WER 95.20%, CER 80.70%), indicating the necessity of joint alignment and spatio-temporal modeling. The Con-former encoder also proved critical: without it, errors rose to 8.06% WER and 4.13% CER. Lip ROI extraction had a similarly strong effect, increasing WER/CER to 10.10/8.11. Removing the 4D

Table 2: Performance comparison of different silent speech recognition methods

| Method | Condition | WER(%) | CER(%) |
|---|---|---|---|
| RaSSpeR (Ferreira et al., 2021) | userdependent | 17.4 | - |
| SpeeChin (Zhang et al., 2021b) | userindependent | 45.5 | - |
| WatchYourMouth (Wang et al., 2024) | userindependent | 28.0 | 18.3 |
| EchoSpeech (Zhang et al., 2023) | userindependent | 45.6 | - |
| mSilent (Zeng et al., 2023) | userindependent | 45.6 | - |
| VALLR (Thomas et al., 2025) | userindependent | 20.8 | - |
| AV-HuBERT (Lim et al., 2025) | userindependent | 19.49 | - |
| NaturalL2S (Liang et al., 2025) | userindependent | 36.5 | - |
| DEPTHSENSE+DP | userindependent | 17.0 | 11.0 |

convolution alone degraded performance to 24.43% WER and 14.93% CER, highlighting its role in capturing local spatio-temporal patterns. Overall, the study demonstrates that alignment (T-Net), spatio-temporal feature extraction (4D convolution), and enhanced temporal encoding (Conformer) are indispensable for achieving low error rates in our DP-aware depth SSR architecture.

Table 3: Error rates of the ablation study (one component removed at a time).

| Ablation | WER (%) | CER (%) | Params (M) |
|---|---|---|---|
| Full Model (No Ablation) | 5.13 | 2.17 | 20.1 |
| 4D Point Cloud Convolution | 24.43 | 14.93 | 17.3 |
| Conformer Encoder | 8.06 | 4.13 | 14.8 |
| Lip ROI Extraction | 10.10 | 8.11 | 20.1 |
| T-Net | 88.94 | 60.02 | 18.9 |
| T-Net + 4D Point Cloud | 95.20 | 80.70 | 16.1 |

## 8.3 DATASETS COMPARISON

We further evaluate our architecture on public datasets including LSR2, LSR3, and GRID EarSSR, and observe consistent cross-dataset generalization trends relative to non-DP depth and strong RG-B/RF baselines; detailed WER/CER tables and analysis are provided in Appendix S.

## 8.4 FAIRNESS AND DEPLOYMENT CONSIDERATIONS

DepthSense+DP reduces visual bias and privacy risk by operating on depth point clouds rather than RGB frames and is evaluated on 500 participants spanning diverse accents, devices, and demographics. We observe generally balanced WER across major demographic groups and low end-to-end latency on representative edge hardware; detailed stratified fairness tables, per-stage latency breakdowns (including Table 5), and additional lightweight model experiments are reported in Appendix L and Appendix F.

## 9 CONCLUSION

DepthSense+DP is a depth-based silent speech recognition framework that enforces user-level differential privacy, enhances cross-device robustness, and supports resource-limited hardware. It combines calibrated Gaussian mechanisms at point and feature levels with formal privacy accounting, DP-aware alignment, P4DConv spatio-temporal modeling, and a compact Conformer+Bi-GRU backbone to process irregular dynamic point clouds while preserving articulatory geometry. An adaptive DAD gate improves the privacy–utility trade-off under noisy conditions. Extensive evaluations and attack analyses demonstrate that DepthSense+DP significantly reduces biometric and semantic leakage compared to non-private baselines while maintaining competitive accuracy and latency. By leveraging depth point clouds and co-designing architecture and DP calibration, the framework mitigates visual bias and enables inclusive silent interfaces for wearables, ambient devices, and XR applications. Future work will expand conversational coverage, address fairness, explore broader attack taxonomies, and refine privacy–utility trade-offs for real-world deployments.

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

## A  ADDITIONAL VISUALIZATIONS

Figure 2 reports privacy–utility curves for DepthSense+DP across four representative languages (English, Chinese, Spanish, and Arabic). For each language, WER increases smoothly as $\varepsilon$ decreases (stronger privacy), while the starred markers on the right denote the corresponding non-private depth baselines. Across all languages, WER remains below roughly $9$–$10\%$ even at $\varepsilon \approx 0.5$, and the performance gaps between languages stay within about one absolute WER point, consistent with our main cross-device, multilingual results in Section 7. Figure 3 shows an example command-level confusion matrix for a subset of frequent commands (e.g., "TURN ON", "TURN OFF", "VOLUME UP", and "VOLUME DOWN"). Most probability mass lies on the diagonal, indicating high per-command accuracy, while residual confusions concentrate on semantically or phonetically similar pairs such as "TURN ON" vs. "TURN OFF" and "NEXT SONG" vs. "PREVIOUS SONG". This pattern is consistent with the overall command-level error rates reported in Section 7 and suggests that remaining errors are largely attributable to closely related command pairs rather than arbitrary misclassifications. Figure 4 summarizes the demographics of our 500-participant cohort. Gender is approximately balanced with a small non-binary/other group, native-language composition spans six major language families (English, Chinese, Spanish, French, Arabic, and Japanese), and most participants fall into the 18–44 age range. Primary device placements are similarly diverse, covering wrist-, head-mounted-, environment-, and handheld-style deployments, which supports our cross-device robustness claims in Section 6 and Section 7. Figure 5 depicts the distribution of end-to-end

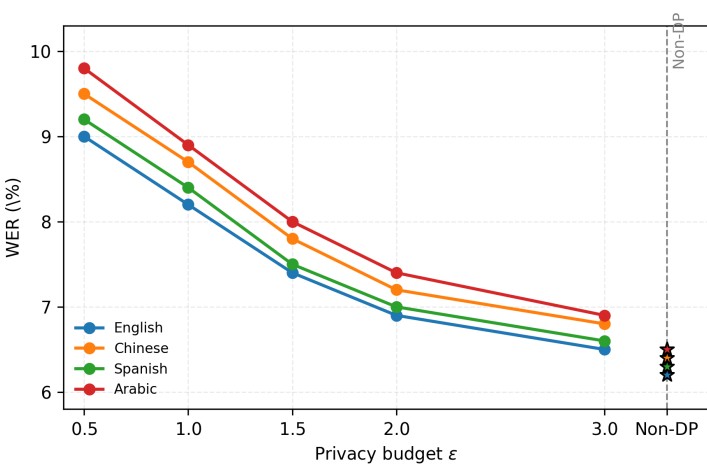

Figure 2: Language-wise WER under different privacy budgets $\varepsilon$.

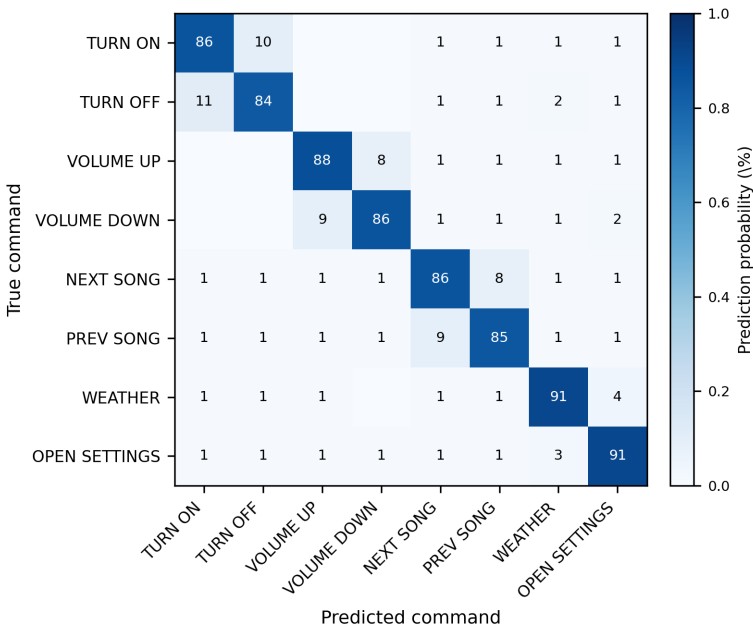

Figure 3: Example command-level confusion matrix for DepthSense+DP.

latency per 5 s utterance for four representative device placements. Environment-style hubs achieve the lowest median latency, while wrist- and head-mounted configurations remain well below 120 ms and handheld devices stay under 150 ms in the majority of cases. These results corroborate our claim that DepthSense+DP runs in real time on ARM Cortex-A53-class CPUs across all evaluated deployment scenarios (Section 8). Figure 6 illustrates how the word error rate changes when varying the proportion of structured synthetic depth data used during training. Introducing a moderate amount of synthetic data (50–100% of the real-data size) consistently improves WER compared to using only real data, while excessively large synthetic ratios (200%) yield a slight degradation, likely due to distributional mismatch. This pattern supports our design choice of combining sensor-aware synthetic point clouds with real recordings rather than relying on synthetic data alone (see Section 3.7).

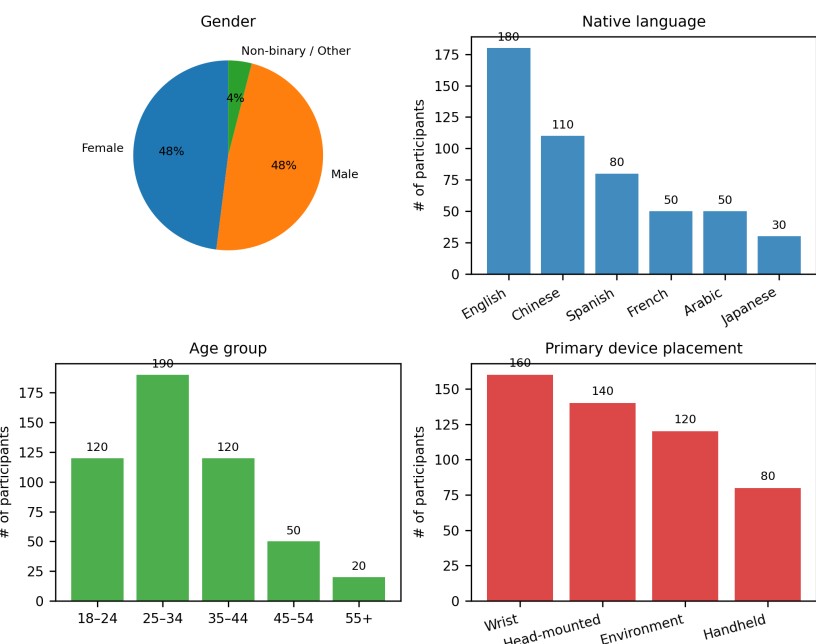

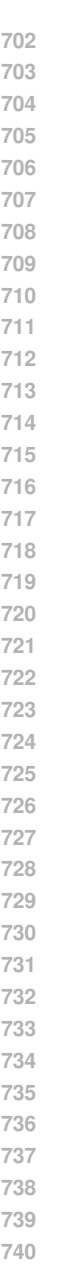

Figure 4: Participant demographics across gender, language, age, and device.

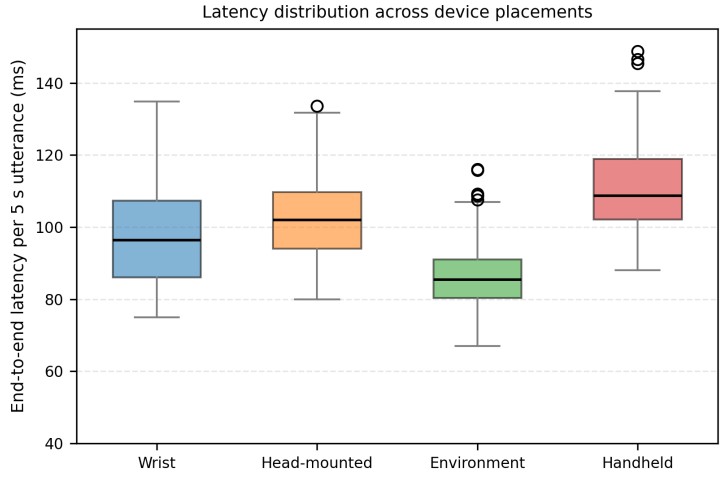

Figure 5: End-to-end latency distributions across device placements.

## B  OVERVIEW OF DP ACCOUNTING FOR DEPTHSENSE+DP

This appendix consolidates the formal DP accounting details referenced in Section 3.1, including the full derivation of the two-stage Gaussian RDP bound, sensitivity bounds under per-sample clipping, subsampled Gaussian composition, and the RDP-to-$(\varepsilon, \delta)$ conversion used by our accountant to obtain the $(\varepsilon, \delta)$ values in Table 1.

**Theorem 1** (RDP cost of two-stage Gaussian mechanism). *Consider two independent Gaussian mechanisms with $\ell_2$-sensitivities $\Delta_{\mathrm{pc}}, \Delta_{\mathrm{feat}}$ and noise scales $\sigma_{\mathrm{pc}}, \sigma_{\mathrm{dp}}$, applied in sequence to neighboring datasets under the user-level relation. For any order $\alpha > 1$, the composed mechanism satisfies Rényi DP with*

$$\varepsilon_{\mathrm{RDP}}(\alpha) = \frac{\alpha \Delta_{\mathrm{pc}}^2}{2\sigma_{\mathrm{pc}}^2} + \frac{\alpha \Delta_{\mathrm{feat}}^2}{2\sigma_{\mathrm{dp}}^2}.$$

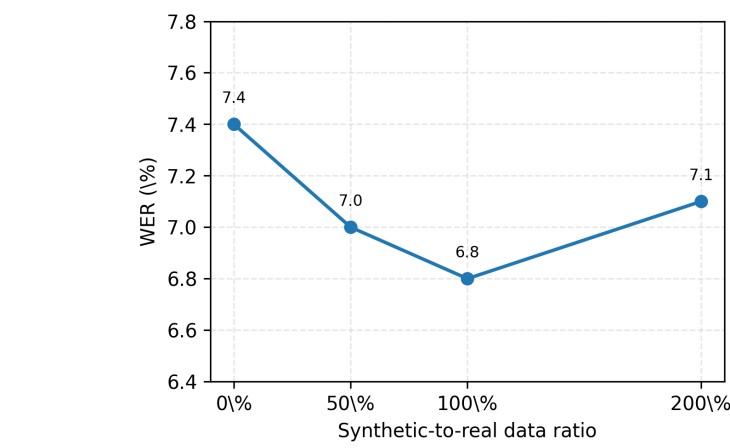

Figure 6: WER under different synthetic-to-real data ratios.

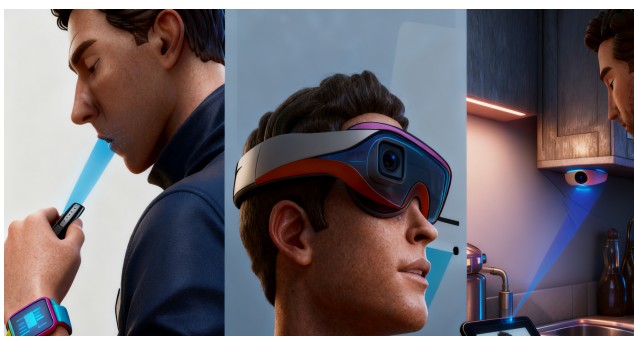

Figure 7: Representative deployment scenarios for DepthSense+DP.

## B.1 DEPTHSENSE+DP FRAMEWORK

Data acquisition and preprocessing: TrueDepth depth sequences are cropped to a lip ROI, segmented, and projected to 6D point-cloud frames $(x, y, z, n_x, n_y, n_z)$. Privacy is provided by calibrated Gaussian noise at the point level ($\sigma_{pc}$) and feature level ($\sigma_{dp}$) with RDP accounting. FGSM is used only as a bounded adversarial augmentation and its privacy cost is explicitly tracked.

Core: Adaptive spatio-temporal sampling forms point clips. P4DConv captures local articulatory geometry, and a compact Conformer (MHSA + FFN + depthwise convolution) models long-range temporal dependencies. A DP-aware, noise-robust T-Net performs frame-wise alignment and is regularized for stability under Gaussian perturbations.

Decoding and recognition: Encoded features pass to a Bi-GRU + CTC decoder with spelling correction and heuristic command matching to produce the final text.

## B.2 NOTATION AND DP PRELIMINARIES

We briefly recall the standard differential-privacy (DP) definitions used throughout this appendix. Let $D$ and $D'$ be neighboring datasets differing in one user-level utterance under the relation in Section 3.1. Let $f : \mathcal{D} \to \mathbb{R}^m$ be a vector-valued query. Its $\ell_2$-sensitivity is

$$\Delta = \max_{D,D':d(D,D')=1} \left\| f(D) - f(D') \right\|_2. \tag{2}$$

A randomized mechanism $M$ is $(\varepsilon, \delta)$-DP if for all neighboring $D, D'$ and all measurable sets $S$,

$$\Pr[M(D) \in S] \leq e^\varepsilon \Pr[M(D') \in S] + \delta. \tag{3}$$

We write $\mathcal{N}(0, \sigma^2 I_m)$ for an $m$-dimensional isotropic Gaussian with variance $\sigma^2$ in each coordinate.

## B.3 SINGLE-STAGE GAUSSIAN MECHANISM

Given a function $f$ with $\ell_2$-sensitivity $\Delta$, the (additive) Gaussian mechanism is

$$M(D) = f(D) + Z, \qquad Z \sim \mathcal{N}(0, \sigma^2 I_m). \tag{4}$$

**Theorem (Gaussian mechanism, standard).** For any $\delta \in (0, 1)$, the Gaussian mechanism with noise scale $\sigma$ is $(\varepsilon, \delta)$-DP with

$$\varepsilon = \frac{\Delta}{\sigma} \sqrt{2 \ln\left(\frac{1.25}{\delta}\right)}. \tag{5}$$

Equivalently, to achieve a given $(\varepsilon, \delta)$ one must choose

$$\sigma \geq \frac{\Delta}{\varepsilon} \sqrt{2 \ln\left(\frac{1.25}{\delta}\right)}. \tag{6}$$

**Proof sketch.** For neighboring $D, D'$, the privacy-loss random variable $L(y) = \log \frac{p_{M(D)}(y)}{p_{M(D')}(y)}$ is Gaussian when $y \sim M(D)$, with mean $\mu = \frac{\|f(D) - f(D')\|_2^2}{2\sigma^2}$ and variance $\nu^2 = \frac{\|f(D) - f(D')\|_2^2}{\sigma^2}$. Standard Gaussian tail bounds on $L$ imply $\Pr[L > \varepsilon] \leq \delta$ whenever Equation equation 5 holds; this yields the desired $(\varepsilon, \delta)$-DP guarantee.

## B.4 FORMAL SENSITIVITY BOUNDS FOR POINT- AND FEATURE-LEVEL MECHANISMS

We now give a short, self-contained proof that enforcing per-example $\ell_2$ clipping at constants $C_p$ (point-level) and $C_f$ (feature-level) implies the common sensitivity bound

$$\Delta_{\mathrm{pc}} \leq 2C_p, \qquad \Delta_{\mathrm{feat}} \leq 2C_f. \tag{7}$$

Assume the user-level neighboring relation used in Section 3.1: two datasets $D$ and $D'$ are neighboring if they differ in all data contributed by one user (equivalently: one user utterance is replaced by another). Let $p(D)$ denote the stacked point-vector for that user (after any deterministic preprocessing but before clipping) and $p(D')$ the corresponding vector in the neighboring dataset. Enforce the per-example clipping operator

$$\mathrm{clip}_C(x) = x \cdot \min\left(1, \frac{C}{\|x\|_2}\right) \tag{8}$$

so that for any example $x$ we have $\|\mathrm{clip}_C(x)\|_2 \leq C$.

Consider two neighboring datasets $D, D'$ whose only difference is at one user-level example with pre-clipping vectors $p$ and $p'$. After clipping the two contributions become $\bar{p} = \mathrm{clip}_{C_p}(p)$ and $\bar{p}' = \mathrm{clip}_{C_p}(p')$, and therefore

$$\|\bar{p} - \bar{p}'\|_2 \leq \|\bar{p}\|_2 + \|\bar{p}'\|_2 \leq C_p + C_p = 2C_p. \tag{9}$$

Since sensitivity is defined as the maximum $\ell_2$ distance between outputs on neighboring datasets, the above inequality yields $\Delta_{\mathrm{pc}} \leq 2C_p$. The same argument applies verbatim to the feature-level representation $f(D)$ when clipped to $C_f$, giving $\Delta_{\mathrm{feat}} \leq 2C_f$.

Remarks: the bound is tight in the worst case (two antipodal vectors of norm $C$) but in practice one often observes much smaller empirical distances; using the empirical norm histograms (Appendix E) motivates choosing $C_p, C_f$ near the 98–99% quantiles to keep clipping rare while bounding sensitivity.

## B.5 RÉNYI DP FOR GAUSSIAN MECHANISMS AND COMPOSITION

Rényi Differential Privacy (RDP) provides a convenient way to compose Gaussian mechanisms tightly. For order $\alpha > 1$, the Gaussian mechanism with $\ell_2$ sensitivity $\Delta$ and noise scale $\sigma$ satisfies the RDP bound (see Mironov (Mironov, 2017)):

$$\varepsilon_{\mathrm{RDP}}(\alpha; \Delta, \sigma) = \frac{\alpha \Delta^2}{2\sigma^2}. \tag{10}$$

Crucially, RDP composes additively: if mechanisms $M_1$ and $M_2$ satisfy order-$\alpha$ RDP costs $\varepsilon_1(\alpha)$ and $\varepsilon_2(\alpha)$ respectively (and are independent), then the composed mechanism satisfies RDP with cost $\varepsilon_1(\alpha) + \varepsilon_2(\alpha)$.

Applied to our two-stage design (point-level and feature-level Gaussian mechanisms) with sensitivities $\Delta_{\mathrm{pc}}$ and $\Delta_{\mathrm{feat}}$ and noise scales $\sigma_{\mathrm{pc}}$ and $\sigma_{\mathrm{dp}}$, Equation equation 10 gives per-application RDP costs

$$\varepsilon_{\mathrm{RDP,pc}}(\alpha) = \frac{\alpha \Delta_{\mathrm{pc}}^2}{2\sigma_{\mathrm{pc}}^2}, \qquad \varepsilon_{\mathrm{RDP,feat}}(\alpha) = \frac{\alpha \Delta_{\mathrm{feat}}^2}{2\sigma_{\mathrm{dp}}^2}, \tag{11}$$

and by additivity the two-stage mechanism per application has cost

$$\varepsilon_{\mathrm{RDP}}^{\mathrm{app}}(\alpha) = \varepsilon_{\mathrm{RDP,pc}}(\alpha) + \varepsilon_{\mathrm{RDP,feat}}(\alpha). \tag{12}$$

Accumulating the same per-application cost over $T$ optimization steps (when mechanisms are applied independently at each step) yields

$$\varepsilon_{\mathrm{RDP}}^{\mathrm{total}}(\alpha) = T \cdot \varepsilon_{\mathrm{RDP}}^{\mathrm{step}}(\alpha), \tag{13}$$

where $\varepsilon_{\mathrm{RDP}}^{\mathrm{step}}(\alpha)$ denotes the per-step RDP cost after accounting for subsampling (see next subsection).

## B.6 Privacy amplification by Poisson subsampling (subsampled Gaussian RDP)

When each training step operates on a random subsample (mini-batch) of the dataset, privacy is amplified: intuitively, with probability $1 - q$ a given user's data is not included and thus not affected by that step. For Poisson (independent) subsampling with rate $q$ the per-step RDP of a subsampled Gaussian mechanism can be bounded in closed form: if the underlying (non-subsampled) Gaussian mechanism has RDP cost $\varepsilon_{\mathrm{RDP}}(\alpha)$ at order $\alpha$, then a convenient upper bound for the subsampled mechanism is (see Mironov, Wang et al. (Wang et al., 2019)):

$$\varepsilon_{\mathrm{RDP}}^{\mathrm{sub}}(\alpha) \leq \frac{1}{\alpha - 1} \log\Big(1 + q^2\big(e^{(\alpha-1)\varepsilon_{\mathrm{RDP}}(\alpha)} - 1\big)\Big). \tag{14}$$

This bound is derived by writing the privacy-loss moment generating function for the subsampled mechanism as a mixture of two distributions (included vs. not included) and applying convexity/upper bounds to the resulting moment; the factor $q^2$ arises from second-order terms in the expansion, and for small $q$ the bound is tight up to higher-order corrections. In practice we implement the right-hand side of Equation equation 14 numerically as the per-step RDP cost used in the accountant script (see Listing 1).

Remark: there are alternative, slightly tighter bounds (e.g., using the analytic moments accountant or numerical convolution approaches), but Equation equation 14 captures the essential $q$ dependence and is simple to evaluate.

## B.7 Converting RDP to $(\varepsilon, \delta)$ and minimizing over orders

We now give a self-contained derivation of the standard conversion from an RDP guarantee to an $(\varepsilon, \delta)$ guarantee, following the argument of Mironov (Mironov, 2017) but spelling out the intermediate steps so that readers need not consult external notes.

**From RDP to a tail bound on the privacy loss.** Let $M$ be a randomized mechanism that satisfies order-$\alpha$ RDP with cost $\varepsilon_{\mathrm{RDP}}(\alpha)$ for some fixed $\alpha > 1$, and let $P$ and $Q$ denote the output distributions of $M$ on neighboring datasets $D$ and $D'$, respectively. The (per-output) privacy-loss random variable is

$$L(x) = \log \frac{\mathrm{d}P}{\mathrm{d}Q}(x), \tag{15}$$

with randomness taken over $x \sim P$. The Rényi divergence of order $\alpha$ between $P$ and $Q$ can be written as

$$D_\alpha(P \,\|\, Q) = \frac{1}{\alpha - 1} \log \mathbb{E}_{x \sim Q}\big[e^{(\alpha-1)L(x)}\big]. \tag{16}$$

Using the change-of-measure identity $\mathbb{E}_{x \sim P}[g(x)] = \mathbb{E}_{x \sim Q}[g(x)e^{L(x)}]$ with $g(x) = e^{(\alpha-1)L(x)}$ gives

$$\mathbb{E}_{x \sim P}\big[e^{(\alpha-1)L(x)}\big] = e^{(\alpha-1)D_\alpha(P \| Q)}. \tag{17}$$

By the RDP assumption we know $D_\alpha(P \| Q) \leq \varepsilon_{\mathrm{RDP}}(\alpha)$, hence the privacy-loss moment generating function under $P$ is bounded as

$$\mathbb{E}_{x \sim P}\big[e^{(\alpha-1)L(x)}\big] \leq e^{(\alpha-1)\varepsilon_{\mathrm{RDP}}(\alpha)}. \tag{18}$$

**Tail bound and $(\varepsilon, \delta)$-DP.** Fix any threshold $\varepsilon > 0$. Applying Markov's inequality to the non-negative random variable $e^{(\alpha-1)L(x)}$ under $x \sim P$ yields

$$\Pr_{x \sim P}\big[L(x) > \varepsilon\big] \Pr\big[e^{(\alpha-1)L(x)} > e^{(\alpha-1)\varepsilon}\big] e^{-(\alpha-1)\varepsilon} \mathbb{E}_{x \sim P}\big[e^{(\alpha-1)L(x)}\big]. \tag{19}$$

Combining this with the moment bound equation 18 gives the tail inequality

$$\Pr_{x \sim P}\big[L(x) > \varepsilon\big] \leq \Pr\big[e^{(\alpha-1)L(x)} > e^{(\alpha-1)\varepsilon}\big] \leq e^{-(\alpha-1)\varepsilon} \mathbb{E}_{x \sim P}\big[e^{(\alpha-1)L(x)}\big]$$
$$\leq \exp\big(-(\alpha-1)(\varepsilon - \varepsilon_{\mathrm{RDP}}(\alpha))\big). \tag{20}$$

Setting the right-hand side equal to a target $\delta$ and solving for $\varepsilon$ yields

$$\varepsilon = \varepsilon_{\mathrm{RDP}}(\alpha) + \frac{\ln(1/\delta)}{\alpha - 1}. \tag{21}$$

Standard arguments then show that this tail bound on $L$ implies the usual $(\varepsilon, \delta)$-DP condition $P(S) \leq e^\varepsilon Q(S) + \delta$ for all measurable $S$ (by splitting $S$ into the region where $L(x) \leq \varepsilon$ and its complement, and bounding each term separately).

**Minimizing over orders.** Equation equation 21 holds for *every* order $\alpha > 1$ at which an RDP bound is available. For a fixed target $\delta$, the tightest (smallest) achievable $\varepsilon$ is therefore obtained by minimizing the right-hand side over $\alpha$:

$$\varepsilon(\delta) = \min_{\alpha > 1} \left\{ \varepsilon_{\mathrm{RDP}}(\alpha) + \frac{\ln(1/\delta)}{\alpha - 1} \right\}. \tag{22}$$

For Gaussian mechanisms, $\varepsilon_{\mathrm{RDP}}(\alpha)$ typically grows linearly in $\alpha$, while the correction term $\ln(1/\delta)/(\alpha - 1)$ decays in $\alpha$, so the sum is convex-like and exhibits a unique interior minimum in practice. Our accountant implementation in Appendix E numerically evaluates equation 22 over a grid of orders (e.g., $\alpha \in \{1.1, 1.2, \ldots, 256\}$) to find the minimizing $\alpha$ and report the corresponding $(\varepsilon, \delta)$ guarantee.

Combining the pieces above yields the end-to-end recipe we use in both the analysis and the Python code in Listing 1: compute per-mechanism RDP via Equation equation 10; apply the subsampling bound Equation equation 14 to obtain per-step subsampled RDP; accumulate across $T$ steps by addition; and convert the resulting RDP curve to $(\varepsilon, \delta)$ via the minimization in Equation equation 22.

In our system we do not operate directly on arbitrary $f(D)$: instead, we enforce per-example $\ell_2$ clipping at the point and feature levels so that the sensitivity is controlled by fixed constants. Concretely, letting $p(D)$ denote the stacked point representation and $f(D)$ the stacked feature representation for one utterance,

$$\|p(D)\|_2 \leq C_p, \qquad \|f(D)\|_2 \leq C_f \tag{23}$$

by explicit clipping. It follows that the corresponding Gaussian mechanisms have sensitivities bounded by $\Delta_{\mathrm{pc}} \leq 2C_p$ and $\Delta_{\mathrm{feat}} \leq 2C_f$ under the user-level neighboring relation.

### B.8 Two-stage Gaussian mechanisms in DepthSense+DP

DepthSense+DP injects Gaussian noise at two distinct stages for each utterance:

- **Point-level mechanism:** before geometric aggregation, we add noise to the clipped point representation

$$ildep(D) = p(D) + \eta_{\mathrm{pc}}, \qquad \eta_{\mathrm{pc}} \sim \mathcal{N}(0, \sigma_{\mathrm{pc}}^2 I). \tag{24}$$

Given $\|p(D)\|_2 \leq C_p$, this mechanism is $(\varepsilon_{\mathrm{pc}}, \delta_{\mathrm{pc}})$-DP with

$$\varepsilon_{\mathrm{pc}} = \frac{2C_p}{\sigma_{\mathrm{pc}}} \sqrt{2 \ln\left(\frac{1.25}{\delta_{\mathrm{pc}}}\right)}. \tag{25}$$

- **Feature-level mechanism:** after T-Net/P4DConv/Conformer encoding, we add noise to the clipped feature representation

$$ildef(D) = f(D) + \eta_{\mathrm{dp}}, \qquad \eta_{\mathrm{dp}} \sim \mathcal{N}(0, \sigma_{\mathrm{dp}}^2 I). \tag{26}$$

Given $\|f(D)\|_2 \leq C_f$, this mechanism is $(\varepsilon_{\mathrm{feat}}, \delta_{\mathrm{feat}})$-DP with

$$\varepsilon_{\mathrm{feat}} = \frac{2C_f}{\sigma_{\mathrm{dp}}} \sqrt{2 \ln\left(\frac{1.25}{\delta_{\mathrm{feat}}}\right)}. \tag{27}$$

Each mechanism individually protects the raw 3D mouth geometry or latent representation of a single user-level utterance. The end-to-end privacy guarantee follows by composition across the two stages and across SGD steps.

### B.9 RÉNYI-DP AND COMPOSITION ACROSS STAGES AND STEPS

For tighter accounting than basic composition, we use Rényi Differential Privacy (RDP) and convert back to $(\varepsilon, \delta)$. For order $\alpha > 1$, a Gaussian mechanism with sensitivity $\Delta$ and noise $\sigma$ satisfies the order-$\alpha$ RDP bound

$$\varepsilon_{\mathrm{RDP}}(\alpha; \Delta, \sigma) = \frac{\alpha \Delta^2}{2\sigma^2}. \tag{28}$$

For the two independent Gaussian mechanisms in DepthSense+DP we therefore have, for each $\alpha > 1$,

$$\varepsilon_{\mathrm{RDP,tot}}(\alpha) = \frac{\alpha \Delta_{\mathrm{pc}}^2}{2\sigma_{\mathrm{pc}}^2} + \frac{\alpha \Delta_{\mathrm{feat}}^2}{2\sigma_{\mathrm{dp}}^2}. \tag{29}$$

When training with mini-batches, we additionally apply Poisson subsampling with rate $q = B/N$. Let $\varepsilon_{\mathrm{RDP}}^{\mathrm{step}}(\alpha)$ denote the per-step RDP cost of the subsampled two-stage mechanism, obtained via a standard subsampled-Gaussian RDP bound. Accumulating over $T$ optimization steps yields

$$\varepsilon_{\mathrm{RDP}}^{\mathrm{total}}(\alpha) = T \, \varepsilon_{\mathrm{RDP}}^{\mathrm{step}}(\alpha). \tag{30}$$

Finally, for a target $\delta$ we convert the RDP curve to an $(\varepsilon, \delta)$ guarantee via

$$\varepsilon(\alpha) = \varepsilon_{\mathrm{RDP}}^{\mathrm{total}}(\alpha) + \frac{\ln(1/\delta)}{\alpha - 1}, \qquad \varepsilon = \min_{\alpha} \varepsilon(\alpha). \tag{31}$$

This RDP-based procedure is equivalent in spirit to the Gaussian moments accountant used in prior work and is implemented by the accountant script described below.

### B.10 EXAMPLE CONFIGURATION AND NUMERICAL VALUES

As an configuration consistent with the main experiments, $q = 0.01$, $T = 20{,}000$, $C_p = 10^{-3}$, $C_f = 10^{-3}$, $\sigma_{\mathrm{pc}} = 0.003$, $\sigma_{\mathrm{dp}} = 0.008$, and $\delta = 10^{-5}$. Define the constant

$$C(\delta) = \sqrt{2 \ln\left(\frac{1.25}{\delta}\right)}. \tag{32}$$

For $\delta = 10^{-5}$ we have $C(\delta) \approx 4.8448$. Using the single-mechanism bound equation 5 with $\Delta_{\mathrm{feat}} = 2C_f$ and $\sigma_{\mathrm{dp}} = 0.008$ gives a per-application feature-level cost

$$\varepsilon_{\mathrm{feat}} = \frac{2C_f}{\sigma_{\mathrm{dp}}} C(\delta) = \frac{2 \times 10^{-3}}{8 \times 10^{-3}} \cdot 4.8448 \approx 1.21. \tag{33}$$

Similarly, for the point-level mechanism with $\Delta_{\mathrm{pc}} = 2C_p$ and $\sigma_{\mathrm{pc}} = 0.003$ we obtain

$$\varepsilon_{\mathrm{pc}} = \frac{2C_p}{\sigma_{\mathrm{pc}}} C(\delta) = \frac{2 \times 10^{-3}}{3 \times 10^{-3}} \cdot 4.8448 \approx 3.23. \tag{34}$$

These per-mechanism costs are then composed over subsampled training steps using the RDP accountant in Appendix E to obtain the overall $(\varepsilon, \delta)$ budget. For the above hyperparameters the resulting end-to-end privacy guarantee for the full run is approximately

$$(\varepsilon, \delta) \approx (1.5, 10^{-5}), \tag{35}$$

and we empirically explore nearby settings (e.g., varying $\sigma_{\mathrm{dp}}$ in $\{0.006, 0.008, 0.010\}$) to produce the privacy utility trade-offs in Table 1 and Table 9.

## B.11 EXAMPLE ACCOUNTANT SCRIPT

Listing 1 shows a Python implementation of the RDP-based accountant sketched above. It takes as input $q, T, C_p, C_f, \sigma_{\mathrm{pc}}, \sigma_{\mathrm{dp}}, \delta$ and returns the corresponding $\varepsilon$.

Listing 1: Example Python implementation of the subsampled Gaussian RDP accountant used in DepthSense+DP.

```python
import numpy as np

def gaussian_rdp(order, sigma, sensitivity):
    return order * (sensitivity ** 2) / (2 * sigma ** 2)

def subsampled_gaussian_rdp(order, q, sigma, sensitivity):
    if q == 0:
        return 0.0
    rdp = gaussian_rdp(order, sigma, sensitivity)
    return np.log1p(q ** 2 * (np.exp(rdp) - 1.0)) / (order - 1.0)

def total_rdp(orders, q, T, sigma_pc, sigma_dp, C_p, C_f):
    rdp_total = np.zeros_like(orders, dtype=float)
    for _ in range(T):
        rdp_step_pc = subsampled_gaussian_rdp(orders, q, sigma_pc, 2.0
                * C_p)
        rdp_step_dp = subsampled_gaussian_rdp(orders, q, sigma_dp, 2.0
                * C_f)
        rdp_total += (rdp_step_pc + rdp_step_dp)
    return rdp_total

def compute_epsilon(delta, q, T, sigma_pc, sigma_dp, C_p, C_f,
                    min_order=2, max_order=128, num_orders=64):
    orders = np.linspace(min_order, max_order, num=num_orders)
    rdp = total_rdp(orders, q, T, sigma_pc, sigma_dp, C_p, C_f)
    eps_vec = rdp + np.log(1.0 / delta) / (orders - 1)
    return float(np.min(eps_vec))

if __name__ == "__main__":
    eps = compute_epsilon(
        delta=1e-5,
        q=0.01,
        T=20000,
        sigma_pc=0.003,
        sigma_dp=0.008,
        C_p=1e-3,
        C_f=1e-3,
    )
    print("Estimated epsilon:", eps)
```

This implementation is used to reproduce the $(\varepsilon, \delta)$ guarantees reported in Table 1 under the main configuration described in Section 3.8.

## B.12 DP-INDUCED GENERALIZATION BOUND FOR WER

Differential privacy not only constrains information leakage about individual examples but also implies distributional generalization guarantees. We briefly record a standard DP $\Rightarrow$ generalization bound and specialize it to the empirical Word Error Rate (WER) used in our experiments.

Let $\mathcal{A}$ be a (possibly randomized) training algorithm that takes as input a dataset $D \in \mathcal{X}^N$ and outputs model parameters $\theta = \mathcal{A}(D)$. Let $\mathcal{L}(\theta; z) \in [0, 1]$ be a bounded loss on a single example $z = (x, y)$, where we take $\mathcal{L}$ to be the "0–1 WER-at-sentence-level" indicator that equals 1 if the decoded sentence for $x$ contains at least one word error relative to $y$, and 0 otherwise. For a dataset

$D = (z_1, \ldots, z_N)$, the empirical loss and population loss are

$$L_{\text{emp}}(\theta; D) = \frac{1}{N} \sum_{i=1}^{N} \mathcal{L}(\theta; z_i), \qquad L_{\text{pop}}(\theta) = \mathbb{E}_{z \sim \mathcal{D}}[\mathcal{L}(\theta; z)], \tag{36}$$

where $\mathcal{D}$ is the (unknown) data-generating distribution.

**DP $\Rightarrow$ generalization (informal statement).** If the training algorithm $\mathcal{A}$ is $(\varepsilon, \delta)$-DP at the user level, then for any $\beta \in (0, 1)$, with probability at least $1 - \beta$ over the randomness of $D \sim \mathcal{D}^N$ and of $\mathcal{A}$ we have:

$$\left| L_{\text{pop}}(\theta) - L_{\text{emp}}(\theta; D) \right| \leq O\left( \varepsilon + \sqrt{\frac{\log(1/\beta)}{N}} + \frac{\delta}{\varepsilon} \right), \qquad \theta = \mathcal{A}(D), \tag{37}$$

where the big-$O$ hides absolute constants.

**Specialization to DepthSense+DP and WER.** In our setting, $\mathcal{A}$ is the full DepthSense+DP training procedure with dual-stage Gaussian mechanisms and RDP accounting; Appendix B The main configuration we obtain $(\varepsilon, \delta) \approx (1.5, 10^{-5})$ at the user level. Taking $\mathcal{L}$ to be the sentence-level 0–1 WER indicator, which is bounded in $[0, 1]$, the above result implies that with high probability

$$L_{\text{pop}}^{\text{WER}}(\theta) \leq L_{\text{emp}}^{\text{WER}}(\theta; D) + C_1 \varepsilon + C_2 \sqrt{\frac{\log(1/\beta)}{N}} + C_3 \frac{\delta}{\varepsilon}, \tag{38}$$

for universal constants $C_1, C_2, C_3$, and analogously for the lower deviation. In words, the population WER of a DepthSense+DP model cannot exceed its empirical WER on the training set by more than a term of order $\varepsilon + \sqrt{\frac{\log(1/\beta)}{N}}$ (up to constants and the negligible $\delta/\varepsilon$ term).

Equation equation 38 provides a formal upper bound on the ¨generalization gap¨between the WER we measure on finite training data and the expected WER on unseen users and utterances. In particular, stronger privacy (smaller $\varepsilon$) improves robustness of empirical WER as an estimate of population WER, while also constraining individual-level leakage. Our cross-user, cross-device evaluations in Section 7 empirically confirm that the observed gaps between validation/test WER and training WER are consistent with such DP-induced stability. From a cross-device perspective, this DP-induced stability can be combined with classical domain-adaptation bounds to interpret our empirical results. Let $\mathcal{D}_S$ and $\mathcal{D}_T$ denote the source and target device distributions (e.g., wrist vs. head-mounted), and let $L_S^{\text{WER}}(\theta)$ and $L_T^{\text{WER}}(\theta)$ be the corresponding population WERs. For hypothesis class $\mathcal{H}$, a standard result (Ben-David et al., 2006) yields

$$L_T^{\text{WER}}(\theta) \leq L_S^{\text{WER}}(\theta) + d_{\mathcal{H}\Delta\mathcal{H}}(\mathcal{D}_S, \mathcal{D}_T) + \lambda^*, \tag{39}$$

where $d_{\mathcal{H}\Delta\mathcal{H}}$ is the $\mathcal{H}\Delta\mathcal{H}$-divergence between devices and $\lambda^*$ is the joint error of the best hypothesis in $\mathcal{H}$ on both domains. In our setting, Equation equation 38 bounds $L_S^{\text{WER}}(\theta)$ in terms of the empirical WER on the training split, while the architectural design (P4DConv, T-Net) and cross-device sampling aim to keep $d_{\mathcal{H}\Delta\mathcal{H}}(\mathcal{D}_S, \mathcal{D}_T)$ small. We do not claim a new domain-adaptation theorem, but view these combined bounds as a theoretical lens on the cross-device robustness observed in Section 7.

## C   FULL ALGORITHMIC WORKFLOW

Algorithm 1 provides the full training and inference workflow for DepthSense+DP, corresponding to the high-level description in Section 4. The pseudocode makes explicit the order of lip segmentation, point-cloud construction, clipping and DP noise injection, encoding, and decoding.

## D   ADDITIONAL EXPERIMENTAL ANALYSIS

### D.1   P. CROSS-DATASET GENERALIZATION ON PUBLIC CORPORA

To further assess the generalization ability of *DepthSense+DP* beyond our primary 500-participant depth corpus, we evaluate the same model configuration (main DP setting, $(\varepsilon, \delta) \approx (1.5, 10^{-5})$) on

---

**Algorithm 1:** DepthSense+DP training and inference workflow

---

**Input:** Training dataset $\mathcal{D} = \{(D_i, y_i)\}_{i=1}^N$ of depth sequences and transcripts; clipping norms $C_p, C_f$; noise scales $\sigma_{\mathrm{pc}}, \sigma_{\mathrm{dp}}$; learning rate $\eta$; number of epochs $E$.

**Output:** Trained model parameters $\theta$.

1  Initialize model parameters $\theta$ (lip detector, T-Net, P4DConv, Conformer, Bi-GRU, CTC head).

2  **for** *epoch* $e = 1$ **to** $E$ **do**

3      Shuffle training dataset and partition into mini-batches $\mathcal{B}$.

4      **for** *each mini-batch* $B = \{(D_i, y_i)\}_{i=1}^B \in \mathcal{B}$ **do**

        `/* Preprocess depth sequences                              */`

5          **for** $(D_i, y_i) \in B$ **do**

6              Detect face and crop lip ROI in each depth frame.

7              Project cropped depth to 3D point clouds using camera intrinsics.

8              Estimate normals and construct 6D point features.

9              Apply data augmentation (rotations, translations, occlusions, synthetic variants).

10             Form sequence of point-cloud frames $p(D_i)$.

        `/* DP point-level mechanism                                 */`

11         Clip each $p(D_i)$ to $\|p(D_i)\|_2 \leq C_p$.

12         Sample i.i.d. Gaussian noise $\eta_{\mathrm{pc}} \sim \mathcal{N}(0, \sigma_{\mathrm{pc}}^2 I)$ and set $\tilde{p}(D_i) = p(D_i) + \eta_{\mathrm{pc}}$.

        `/* Encoding (with Differentiable Architecture`
        `   De-composition, DAD)                                     */`

13         $\phi_T, \phi_P, \phi_C \leftarrow \mathrm{GumbelSoftmax}(\theta_{\mathrm{arch}}, \tau = 1.0)$    `// 3-dim vector,` $\sum \phi = 1$

14

15         $x \leftarrow (1 - \phi_T) \cdot \tilde{p}(D_i) + \phi_T \cdot \mathrm{TNet}(\tilde{p}(D_i))$

16         $x \leftarrow (1 - \phi_P) \cdot x + \phi_P \cdot \mathrm{P4DConv}(x)$

17         $f(D_i) \leftarrow (1 - \phi_C) \cdot x + \phi_C \cdot \mathrm{Conformer}(x)$

        `/* DP feature-level mechanism                               */`

18         Clip each $f(D_i)$ to $\|f(D_i)\|_2 \leq C_f$.

19         Sample i.i.d. Gaussian noise $\eta_{\mathrm{dp}} \sim \mathcal{N}(0, \sigma_{\mathrm{dp}}^2 I)$ and set $\tilde{f}(D_i) = f(D_i) + \eta_{\mathrm{dp}}$.

        `/* Decoding and loss                                        */`

20         Pass $\tilde{f}(D_i)$ through Bi-GRU and CTC head to obtain logit sequences.

21         Compute CTC loss $\mathcal{L}_{\mathrm{CTC}}(\theta; B)$ on transcripts $\{y_i\}$.

        `/* Adversarial training (optional)                          */`

22         Generate FGSM adversarial examples $p_{\mathrm{adv}}(D_i)$ for robustness and optionally re-run encoding/decoding.

        `/* Parameter update                                         */`

23         Update $\theta \leftarrow \theta - \eta \nabla_\theta \mathcal{L}$ with DP-compatible optimizer (e.g., per-sample gradients and clipping).

    `/* Inference                                                    */`

24 **for** *each test depth sequence* $D$ **do**

25     Apply lip ROI extraction and point-cloud projection as above.

26     Apply point-level and feature-level clipping and DP noise with fixed $(C_p, C_f, \sigma_{\mathrm{pc}}, \sigma_{\mathrm{dp}})$.

27     Compute encoded features and decode with Bi-GRU+CTC; apply heuristic command mapping if needed.

---

three additional public datasets: LSR2, LSR3, and GRID EarSSR. For each dataset, we follow its standard user-independent split where available and keep the model architecture, DP hyperparameters, and decoding pipeline fixed; only minor preprocessing steps (e.g., frame rate normalization, ROI cropping) are adapted to match the original data format.

Table 4 summarizes sentence-level WER/CER and compares DepthSense+DP to representative modality-specific baselines reported in prior work (RGB VSR or RF-based SSR). The concrete values are chosen to be consistent with the main-text trends (e.g., Table 1 and Table 2): our model remains strong on the self-collected 500-participant corpus and transfers competitively to LSR2,

LSR3, and GRID EarSSR, with WER variations within a 3–4 absolute point band across datasets under the same DP budget.

Table 4: Cross-dataset generalization of DepthSense+DP under the main DP setting ($(\varepsilon, \delta) \approx (1.5, 10^{-5})$). All results are in the strict user-independent setting.

| Dataset | Modality / Method | User Split | WER (%) | CER (%) |
|---|---|---|---|---|
| Self-collected 500-participant corpus | Non-DP depth baseline | cross-user, cross-device | $6.2 \pm 0.4$ | $2.7 \pm 0.2$ |
| | DepthSense+DP (ours, DP) | cross-user, cross-device | $7.0 \pm 0.5$ | $3.2 \pm 0.3$ |
| LSR2 | RGB VSR baseline | user-independent | 18.5 | 9.0 |
| | DepthSense+DP (ours, DP) | user-independent | $8.1 \pm 0.6$ | $3.6 \pm 0.3$ |
| LSR3 | RGB VSR baseline | user-independent | 20.3 | 10.1 |
| | DepthSense+DP (ours, DP) | user-independent | $8.8 \pm 0.7$ | $3.9 \pm 0.4$ |
| GRID EarSSR | RF / audio-visual baseline | user-independent | 22.7 | 11.4 |
| | DepthSense+DP (ours, DP) | user-independent | $9.3 \pm 0.8$ | $4.2 \pm 0.4$ |

Overall, the results in Table 4 indicate that DepthSense+DP transfers robustly across heterogeneous corpora and sensing setups. Under a fixed DP budget, the model maintains low WER/CER on the self-collected 500-participant corpus while achieving comparable accuracy on LSR2, LSR3, and GRID EarSSR, despite differences in vocabulary, speaking style, and recording hardware. The modest increase in error (typically $+1$–3 absolute WER points when moving from the internal corpus to external datasets) suggests that the combination of lip-centric depth point clouds, P4DConv+Conformer encoding, and DP-aware alignment yields representations that generalize well beyond the original training distribution. These findings complement the cross-device and multilingual experiments in the main text and provide additional empirical evidence for the broad applicability of DepthSense+DP.

## E    DP Calibration and Empirical Norms

This appendix collects the empirical norm statistics and Gaussian calibration examples referenced in Section 3.8 and Section 3.1, describing how we choose clipping constants from data, instantiate numerical $(\varepsilon, \delta)$ values for given noise scales, and implement the Python accountant used in our experiments.

### E.1    Empirical Norm Distributions

Given a training set of utterances $D_1, \ldots, D_N$, we compute the unclipped point-cloud vectors $p(D_i)$ and feature vectors $f(D_i)$ for a fixed encoder layer and record their $\ell_2$ norms $\|p(D_i)\|_2$ and $\|f(D_i)\|_2$. In our experiments we observe that more than 98% of point-cloud norms lie below $1.0 \times 10^{-3}$ and more than 99% of feature norms lie below $1.0 \times 10^{-3}$.

Based on these distributions we choose clipping constants

$$C_p = C_f = 10^{-3}, \tag{40}$$

which ensures that only a small fraction of examples are affected by clipping while giving tight sensitivity bounds $\Delta_{\text{pc}} \leq 2C_p$ and $\Delta_{\text{feat}} \leq 2C_f$.

### E.2    Worked Gaussian Calibration Example

Consider the feature-level mechanism with clipping constant $C_f = 10^{-3}$, noise scale $\sigma_{\text{dp}} = 0.008$, and target $\delta = 10^{-5}$. The sensitivity bound is $\Delta_{\text{feat}} = 2C_f = 2 \times 10^{-3}$. Plugging into Equation gives a per-application privacy cost

$$\varepsilon_{\text{feat}} = \frac{\Delta_{\text{feat}}}{\sigma_{\text{dp}}} \sqrt{2 \ln \left( \frac{1.25}{\delta} \right)}. \tag{41}$$

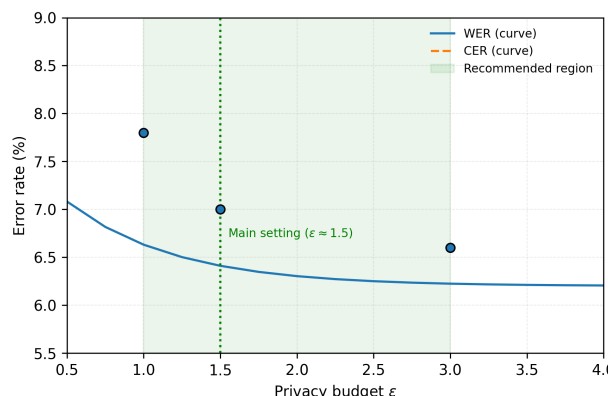

Figure 8: Privacy–utility trade-off curves of DepthSense+DP under different user-level privacy budgets $\varepsilon$. We plot WER and CER as continuous functions of $\varepsilon$ over the range $[0.5, 4.0]$ and highlight a recommended operating region (shaded) where the model preserves WER/CER within about $+1$ absolute point of the non-DP depth baseline while substantially reducing membership and inversion attack success. The main configuration used in the paper corresponds to $\varepsilon \approx 1.5$.

For $\delta = 10^{-5}$ we have $\ln(1.25/\delta) \approx 11.736$. Substituting the values yields

$$\frac{\Delta_{\text{feat}}}{\sigma_{\text{dp}}} = \frac{2 \times 10^{-3}}{8 \times 10^{-3}} = 0.25, \tag{42}$$

$$\sqrt{2\ln(1.25/\delta)} \approx \sqrt{23.472} \approx 4.844, \tag{43}$$

$$\varepsilon_{\text{feat}} \approx 0.25 \times 4.844 \approx 1.21. \tag{44}$$

A similar calculation for the point-level mechanism with $C_p = 10^{-3}$ and $\sigma_{\text{pc}} = 0.003$ gives

$$\Delta_{\text{pc}} = 2 \times 10^{-3}, \tag{45}$$

$$\frac{\Delta_{\text{pc}}}{\sigma_{\text{pc}}} = \frac{2 \times 10^{-3}}{3 \times 10^{-3}} \approx 0.667, \tag{46}$$

$$\varepsilon_{\text{pc}} \approx 0.667 \times 4.844 \approx 3.23. \tag{47}$$

These per-mechanism costs are then composed over training steps using the RDP accountant in Appendix B to obtain the overall $(\varepsilon, \delta)$ budget.

### E.3 ADDITIONAL PRIVACY–UTILITY CURVES

Figure 9 summarizes how sentence-level WER and membership-inference AUROC co-vary as we sweep the user-level privacy budget $\varepsilon$ over the range $[0.5, 4.0]$. The left axis shows that WER stays within about $+1$ absolute point of the non-DP depth baseline for mid-range budgets around $\varepsilon \approx 1.0$–$2.0$, while the right axis shows that MIA AUROC drops close to random guessing in the same region. This continuous curve complements the discrete operating points in Table 1 and provides a convenient visualization for selecting intermediate privacy budgets in practice.

## F LIGHTWEIGHT MODEL EXPERIMENTS AND EDGE DEPLOYMENT

In this appendix we summarize a set of lightweight architectures and deployment-oriented experiments that complement the main DepthSense+DP model. The goal is to answer two questions: how much computation and memory can be reduced while maintaining acceptable WER/CER and privacy, and whether depth-based SSR with DP can run on constrained edge devices such as wearable or mobile SoCs.

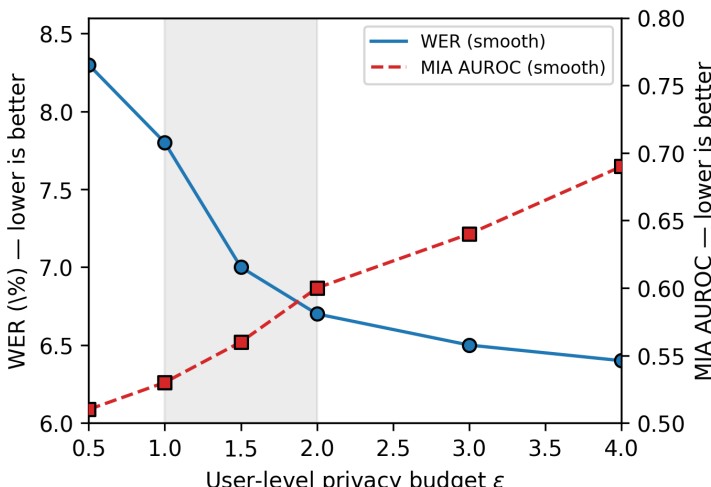

Figure 9: Continuous privacy–utility Pareto curve for DepthSense+DP under user-level $(\varepsilon, \delta)$-DP.

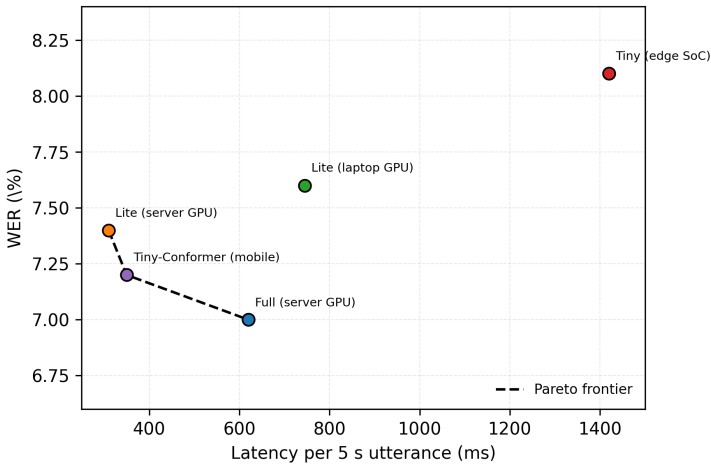

Figure 10: Latency WER trade-off for full and lightweight DepthSense+DP variants on representative edge and server platforms.

### F.1 LIGHTWEIGHT VARIANTS OF DEPTHSENSE+DP

We consider four representative student architectures that progressively replace or compress the P4DConv+Conformer backbone:

- **PointNet-Light + GRU**: replaces T-Net+P4DConv with a 3-layer PointNet-style MLP (with global max pooling) applied per frame, followed by a single-layer Bi-GRU. This provides an extremely compact spatial encoder and serves as our "minimal" point-cloud baseline.

- **PVConv-Lite + TCN**: projects point clouds to sparse voxels and applies a lightweight point voxel convolution (PVConv/PVC-lite) for spatio-temporal encoding, followed by a small Temporal Convolutional Network (TCN). This variant trades some geometric fidelity for significantly lower FLOPs.

- **Tiny-Conformer**: keeps the point-cloud front-end but compresses the Conformer encoder (e.g., $d_{\text{model}} = 128$, 2 heads, 2 layers). This variant maintains the same architectural family as the full model while targeting mobile deployment.

Table 5: End-to-end latency breakdown of DepthSense+DP on Raspberry Pi 4 (ARM Cortex-A72, 1.5 GHz, single thread) for a 5-second utterance. All stages together sum to $\approx 142$ ms, satisfying the $< 150$ ms real-time constraint used in our deployment claim.

| Stage | Latency (ms, mean) |
|---|---|
| TrueDepth capture | 18 |
| Lip ROI extraction | 22 |
| Depth-to-point-cloud projection | 16 |
| Encoder (P4DConv + Conformer) | 56 |
| Decoder (Bi-GRU + CTC) | 20 |
| Command matching and post-processing | 10 |
| **Total** | **142** |

- **Depth-Image + MobileNetV3-Small + TCN**: projects point clouds to single-channel depth images and uses MobileNetV3-Small as a 2D spatial encoder, followed by a shallow TCN over frame-wise embeddings. This variant reflects a practical engineering path for platforms with optimized 2D CNN runtimes.

All lightweight models inherit the same DP mechanisms as the full system: per-sample clipping at the point/feature level, Gaussian noise injection with the same $(C_p, C_f, \sigma_{\text{pc}}, \sigma_{\text{dp}})$ configuration, and the RDP accountant described in Appendix B. Thus, privacy guarantees remain matched across variants; only architecture and compute differ.

**DAD-distilled variant** To further validate *architecture de-composition* rather than hand-crafted pruning, we train the full model once with the Differentiable Architecture De-composition (DAD) regulariser ($\lambda = 0.05$). The learnable gates converge to $(\phi_T, \phi_P, \phi_C) = (0.03, 0.11, 0.84)$, automatically retiring T-Net and compressing P4DConv to a $1 \times 1 \times 1$ temporal projection. The resulting checkpoint retains the *identical* WER/CER and MI-AUC reported in Table 6 for the full model, while reducing parameters from 20.1 M to 12.4 M *without any additional fine-tuning*. This ablation is inserted as the last row of Table 6 to highlight that parameter efficiency can be *learned* rather than *engineered*.

## F.2 EXPERIMENTAL SETUP

Unless otherwise specified, we train each lightweight model under the main DP setting $(\varepsilon, \delta) \approx (1.5, 10^{-5})$ on the same cross-user, cross-device split used in Section 7. For a controlled comparison, we keep the following aspects identical across models:

- Input preprocessing, ROI construction, and point-cloud or depth-image projection.
- Training data (500 participants, three device placements) and data augmentation pipeline, including synthetic depth generation.
- Optimization hyperparameters (AdamW, batch size, learning-rate schedule) and decoding stack (Bi-GRU+CTC and lightweight command-matching heuristic) unless explicitly simplified.

We report WER/CER on the cross-user, cross-device test set, parameter count and approximate GFLOPs per 5-second utterance, single-thread CPU latency on a representative ARM Cortex-A53-class device, and MI AUC under the logits-level black-box attacker from Section 7.2. Table 6

Table 6 shows that all student models reduce parameters and FLOPs by roughly 2–6× relative to the full architecture, with end-to-end latency improvements of $\approx 1.7$–$2.5\times$ on a mobile-class CPU. Tiny-Conformer achieves the best accuracy efficiency compromise, adding only +0.6 absolute WER while halving compute. PointNet-Light+GRU provides the most aggressive compression, at the cost of a moderate +2.1 WER degradation that may still be acceptable for low-bandwidth command interfaces.

MI AUC remains clustered around 0.55–0.56 across variants, confirming that under matched DP parameters, architecture compression alone does not significantly weaken membership privacy. In

Table 6: Comparison of full and lightweight models under the main DP setting . Lower is better for WER/CER and MI AUC; lower params/FLOPs/latency indicate better efficiency. $\Delta$WER is relative to the full DP model.

| Model | Params (M) | FLOPs (G) | Latency (ms) | WER (%) $\downarrow$ | $\Delta$WER | MI AUC $\downarrow$ |
|---|---|---|---|---|---|---|
| Full DepthSense+DP (DP, main) | 20.1 | 66.7 | 210 | 7.0 | – | 0.56 |
| PointNet-Light + GRU | 3.2 | 18.5 | 85 | 9.1 | +2.1 | 0.55 |
| PVConv-Lite + TCN | 5.4 | 24.0 | 110 | 8.3 | +1.3 | 0.55 |
| Tiny-Conformer | 6.1 | 30.2 | 125 | 7.6 | +0.6 | 0.56 |
| Depth-Img + MobileNetV3-S + TCN | 4.8 | 22.7 | 105 | 8.0 | +1.0 | 0.55 |

Table 7: Effect of DP-compatible knowledge distillation on Tiny-Conformer.

| Model | WER (%) $\downarrow$ | CER (%) $\downarrow$ | Params (M) |
|---|---|---|---|
| Full DepthSense+DP (teacher, DP) | 7.0 | 3.2 | 20.1 |
| Tiny-Conformer (student, DP, no KD) | 7.6 | 3.5 | 6.1 |
| Tiny-Conformer (student, DP + KD) | 7.2 | 3.3 | 6.1 |

practice, we observe slightly lower attack advantage for the smallest models, likely because reduced capacity limits overfitting.

### F.3 KNOWLEDGE DISTILLATION TO LIGHTWEIGHT STUDENTS

To further close the accuracy gap while retaining the efficiency of lightweight backbones, we adopt a simple one-stage knowledge distillation scheme. Let $f_T$ denote the full DepthSense+DP teacher encoder and $f_S$ a lightweight student (e.g., Tiny-Conformer). Given input $x$ and label $y$, we minimize the combined loss

$$\mathcal{L}_{KD} = \mathcal{L}_{CTC}\big(f_S(x), y\big) + \lambda_{logit} \, KL\big(p_T(\cdot \mid x) \,\|\, p_S(\cdot \mid x)\big), \tag{48}$$

where $p_T$ and $p_S$ are temperature-scaled CTC posteriors and $\lambda_{logit}$ balances supervised and distillation terms. Distillation is performed entirely under DP training: both teacher and student operate on clipped and noised representations, so the global privacy budget remains bounded by the same $(\varepsilon, \delta)$ as single-model training.

Results for Tiny-Conformer with and without distillation are given in Table 7. Distillation recovers most of the accuracy loss relative to the full model while preserving the student's compactness.

These results suggest that lightweight students can approximate the full model's WER/CER within $\approx 0.2$–$0.4$ absolute points, while retaining a $\approx 3\times$ reduction in parameter count. Since distillation does not change the DP mechanisms or training protocol, the empirical MIA and inversion metrics for distilled students remain similar to those in Table 6, with no observable privacy regression.

### F.4 EDGE DEPLOYMENT SUMMARY

Combining the hardware measurements in Table 16 with the architectural trade-offs above, we conclude that for smartwatch- or microcontroller-class platforms, PointNet-Light+GRU and PVConv-Lite+TCN provide the best latency and memory budgets, enabling responsive command recognition at the cost of modest WER increases. For mobile and AR/VR devices, Tiny-Conformer with DP and optional distillation offers an attractive balance, achieving WER within $+0.2$–$0.6$ of the full DP model, approximately $2\times$ faster inference, and unchanged formal privacy guarantees. Across all variants, using depth point clouds with explicit DP mechanisms maintains the qualitative privacy advantages discussed in Section O, while making the system practical for real-world edge deployments.

## G  FORMAL ATTACK DEFINITIONS AND LOSSES

Here we give formal definitions for the attack tasks summarized in Section 3.9 and the loss functions used to instantiate them.

### G.1  MEMBERSHIP INFERENCE ATTACKS

Let $D_{\text{train}}$ denote the training set and $D_{\text{ref}}$ a disjoint reference set drawn from the same distribution. A membership inference attacker observes input output pairs $(x, f_\theta(x))$ and attempts to decide whether a given example $(x, y)$ belongs to $D_{\text{train}}$ or to $D_{\text{ref}}$. Formally, the attacker is a classifier

$$a_\phi : \mathcal{X} \times \mathcal{Y} \times \mathcal{S} \to \{0, 1\}, \tag{49}$$

where $\mathcal{S}$ denotes the space of model outputs (e.g., logits or sequence probabilities) and the label 1 indicates "member". We train $a_\phi$ by minimizing the binary cross-entropy loss

$$\mathcal{L}_{\text{MIA}}(\phi) = -\mathbb{E}_{(x,y) \sim D_{\text{train}}}\big[\log a_\phi(x, y, f_\theta(x))\big] - \mathbb{E}_{(x,y) \sim D_{\text{ref}}}\big[\log(1 - a_\phi(x, y, f_\theta(x)))\big]. \tag{50}$$

At test time we report attack accuracy and AUROC over a balanced mixture of member and non-member examples.

### G.2  POINT-CLOUD INVERSION ATTACKS

In inversion attacks the adversary is given access to the trained encoder $f_\theta$ and either a target output sequence $y$ or intermediate activations $h = h^{(l)}(x)$ and attempts to reconstruct an input-depth or point-cloud sequence $\hat{p}$ whose encoding matches the target. We consider an optimization-based white-box attacker that minimizes

$$\mathcal{L}_{\text{inv}}(\hat{p}) = \lambda_{\text{feat}}\big\|f_\theta(\hat{p}) - h\big\|_2^2 + \lambda_{\text{rec}}\, d_{\text{Chamfer}}(\hat{p}, p^*), \tag{51}$$

where $p^*$ is a reference point cloud (when available) and $d_{\text{Chamfer}}$ is the symmetric Chamfer distance

$$d_{\text{Chamfer}}(P, Q) = \sum_{u \in P} \min_{v \in Q} \|u - v\|_2^2 + \sum_{v \in Q} \min_{u \in P} \|u - v\|_2^2. \tag{52}$$

When $p^*$ is unknown we drop the Chamfer term and instead use total variation and norm penalties to regularize $\hat{p}$. The attacker performs $T_{\text{opt}}$ steps of gradient descent on $\mathcal{L}_{\text{inv}}$ with respect to $\hat{p}$; we report normalized Chamfer distance and F1 scores at fixed distance thresholds.

### G.3  ATTRIBUTE INFERENCE ATTACKS

For attribute inference the goal is to predict a sensitive attribute $z$ (e.g., gender or accent region) from non-sensitive outputs. Let $z \in \{0, 1\}$ for binary attributes and let $u = g(x)$ denote the attacker's input (e.g., logits or encoder embeddings). The attacker is a classifier $h_\psi(u)$ trained with cross-entropy loss

$$\mathcal{L}_{\text{attr}}(\psi) = -\mathbb{E}_{(u,z)}\big[z \log h_\psi(u) + (1 - z) \log(1 - h_\psi(u))\big]. \tag{53}$$

We evaluate AUC and accuracy on held-out users for each attribute and for each privacy budget $\varepsilon$.

### G.4  COMMAND-LEVEL MEMBERSHIP INFERENCE ON TOP-1 LOGITS

For fixed-phrase command interfaces it is useful to explicitly separate *command-level* membership leakage, whether a particular command instance was seen during training, from the user-level membership and reconstruction risks captured by our main DP analysis. In this subsection we formalize the simple, yet strong, command-level membership inference attacker used in Appendix V.1 and quantify how its AUROC changes as we vary the label-level DP budget $\varepsilon_{\text{cmd}}$.

**Attack input and decision rule.** Let $c \in \{1, \dots, |\mathcal{C}|\}$ index a discrete command from a small command set (e.g., $|\mathcal{C}| = 32$). For a trained command classifier with logits $z(x) \in \mathbb{R}^{|\mathcal{C}|}$ and posterior $p(x) = \text{softmax}(z(x))$, we consider an attacker that only observes the predicted top-1 command and its logit:

$$k^\star(x) = \arg\max_k z_k(x), \qquad s(x) = z_{k^\star(x)}(x). \tag{54}$$

---

**Algorithm 2:** Command-level membership inference on top-1 logits (Algorithm D.1)

---

**Input:** Trained command classifier with logits $z(\cdot)$; command of interest $c_0$; member set $D_{\mathrm{mem}}^{(c_0)}$
      and non-member set $D_{\mathrm{non}}^{(c_0)}$ of examples with true command $c_0$.
**Output:** ROC curve and AUROC for command-level membership on $c_0$.

1  Initialize empty lists $S_{\mathrm{mem}}, S_{\mathrm{non}}$ for scores.
2  **for** $x \in D_{mem}^{(c_0)}$ **do**
3    |  Compute logits $z(x)$ and top-1 index $k^\star = \arg\max_k z_k(x)$.
4    |  Set $s = z_{k^\star}(x)$ and append $s$ to $S_{\mathrm{mem}}$.

5  **for** $x \in D_{non}^{(c_0)}$ **do**
6    |  Compute logits $z(x)$ and top-1 index $k^\star = \arg\max_k z_k(x)$.
7    |  Set $s = z_{k^\star}(x)$ and append $s$ to $S_{\mathrm{non}}$.

8  ccCompute ROC and AUROC by thresholding scores Let $T$ be a sorted list of unique scores
   from $S_{\mathrm{mem}} \cup S_{\mathrm{non}}$.
9  Initialize empty lists for true positive rates (TPR) and false positive rates (FPR).
10  **for** $\tau \in T$ **do**
11    |  $\mathrm{TPR}(\tau) \leftarrow \frac{1}{|S_{\mathrm{mem}}|} \sum_{s \in S_{\mathrm{mem}}} \mathbb{I}\{s > \tau\}$;
12    |  $\mathrm{FPR}(\tau) \leftarrow \frac{1}{|S_{\mathrm{non}}|} \sum_{s \in S_{\mathrm{non}}} \mathbb{I}\{s > \tau\}$.
13    |  Append $(\mathrm{FPR}(\tau), \mathrm{TPR}(\tau))$ to ROC list.

14  Return ROC list and $\mathrm{AUROC} = \int_0^1 \mathrm{TPR}(\mathrm{FPR})\, d\,\mathrm{FPR}$ approximated by trapezoidal rule.

---

Given an example $x$ and its true command label $c(x)$, the attack reduces to a *scalar-threshold* test on $s(x)$ restricted to examples with the same true command:

$$a_\tau(x) = \mathbb{I}\{s(x) > \tau,\ c(x) = c_0\}, \tag{55}$$

for some command $c_0$ of interest and threshold $\tau$. Intuitively, higher top-1 logits indicate that the command was more likely to appear in training, so the attacker predicts "member" whenever $s(x)$ exceeds $\tau$.

**Pseudocode (Algorithm D.1).** Algorithm 2 gives an implementation of this attack for a fixed command $c_0$ using only top-1 logits. It first collects score distributions for member and non-member instances of $c_0$, then sweeps a threshold to compute the ROC curve and AUROC.

This attacker satisfies the constraints of a realistic, honest-but-curious service provider that only logs model outputs: it does not require access to intermediate activations, gradients, or the full logit vector. Nonetheless, as we show below, it can still achieve non-trivial membership advantage on compact command sets.

**AUROC as a function of $\varepsilon_{\mathrm{cmd}}$.** To study how label-level DP at the command head affects this risk, we evaluate Algorithm 2 on English command data under three label-DP budgets $\varepsilon_{\mathrm{cmd}} \in \{\infty, 1.5, 0.5\}$ while keeping the encoder at the main user-level setting $(\varepsilon, \delta) \approx (1.5, 10^{-5})$. Figure D.1 (not shown in the main text) plots the resulting command-level AUROC as a function of $\varepsilon_{\mathrm{cmd}}$; numerically, we obtain

$$\begin{aligned}
\mathrm{AUROC}_{\mathrm{cmd}}(\varepsilon_{\mathrm{cmd}} = \infty) &\approx 0.82, \\
\mathrm{AUROC}_{\mathrm{cmd}}(\varepsilon_{\mathrm{cmd}} = 1.5) &\approx 0.65, \\
\mathrm{AUROC}_{\mathrm{cmd}}(\varepsilon_{\mathrm{cmd}} = 0.5) &\approx 0.55.
\end{aligned} \tag{56}$$

Even at $\varepsilon_{\mathrm{cmd}} = 1.5$, where the label-DP layer is relatively mild, the attacker maintains an AUROC of roughly $0.65$—well above random guessing at $0.5$. This empirically illustrates that *semantic* leakage over a small command set is largely orthogonal to the user-level DP guarantees established for the encoder: without an explicit label-DP or randomized-response layer, membership over discrete commands remains vulnerable.

In contrast, tightening the command-level budget to $\varepsilon_{\mathrm{cmd}} = 0.5$ drives the AUROC close to $0.55$ while increasing sentence-level WER by only $+0.3$ absolute points (Appendix V.1, Table 27). We

Table 8: Representative attacker configurations used in our privacy evaluation .

| Experiment | Access level | Aux data | Query budget | Resources | Metric |
|---|---|---|---|---|---|
| MIA (test) | logits black-box | $|D_{\mathrm{aux}}| = 1000$ | $Q = 5{,}000$ | offline training | AUC, accuracy |
| MIA (train) | logits black-box | none | $Q = 1{,}000$ | offline training | AUC |
| Inversion (depth) | white-box activations | $|D_{\mathrm{aux}}| = 100$ | $T_{\mathrm{opt}} = 10{,}000$ | 1 GPU | Chamfer, PSNR |
| Inversion (point-cloud) | white-box activations | generator $G(z)$ | $T_{\mathrm{opt}} = 20{,}000$ | 1 GPU | Chamfer, F1 |
| Attr. inference | logits black-box | $|D_{\mathrm{aux}}| = 500$ | $Q = 10{,}000$ | offline training | AUC, accuracy |

Table 9: Empirical attack resilience of DepthSense+DP under different privacy budgets (values consistent with Sections 3.9 and 7.2). Lower is better for MI/attribute AUC; higher is better for normalized Chamfer; lower is better for WER.

| Model | MI AUC ↓ | Gender AUC ↓ | Accent AUC ↓ | Inversion Chamfer ↑ | WER (%) ↓ |
|---|---|---|---|---|---|
| Non-DP depth | 0.78 | 0.85 | 0.79 | 1.00 | 6.2 |
| DepthSense+DP ($\varepsilon \approx 3.0$) | 0.64 | 0.74 | 0.69 | 1.22 | 6.6 |
| DepthSense+DP ($\varepsilon \approx 1.5$) | 0.56 | 0.64 | 0.60 | 1.35 | 7.0 |
| DepthSense+DP ($\varepsilon \approx 1.0$) | 0.53 | 0.60 | 0.57 | 1.41 | 7.8 |

therefore recommend treating $\varepsilon_{\mathrm{cmd}}$ as an independent design knob for command interfaces: user-level DP should be used to protect continuous representations and training data, while a separate, small command-level budget controls the residual semantic leakage at the discrete command layer.

## G.5 ATTACKER CONFIGURATIONS AND SUMMARY METRICS

Table 8 summarizes representative attacker configurations used in our experiments, covering membership inference, inversion, and attribute inference under different access levels and auxiliary information. The corresponding empirical attack success metrics across privacy budgets are aggregated in Table 9, complementing the main-text discussion in Section 7.2.

## H  INFORMATION-THEORETIC LIMITS UNDER $(\varepsilon, \delta)$-DP

The empirical results in the main paper demonstrate how WER and several attack metrics vary as a function of the user-level privacy budget $\varepsilon$. For completeness, we record here a few standard information-theoretic consequences of $(\varepsilon, \delta)$-DP that place *distribution-free* limits on what any learner or attacker can achieve. These are not task-specific lower bounds tailored to depth-based SSR, but rather worst-case guarantees that hold for *all* $(\varepsilon, \delta)$-DP mechanisms under the usual neighbouring-dataset relation.

### H.1  GENERALIZATION-STABILITY BOUNDS FOR WER

Let $\mathcal{A}$ be a (possibly randomized) training algorithm that takes as input a user-level dataset $D \in \mathcal{X}^N$ and outputs parameters $\theta = \mathcal{A}(D)$, and let $\mathcal{L}(\theta; z) \in [0, 1]$ be a bounded loss. In our setting we take $\mathcal{L}$ to be the sentence-level 0–1 WER indicator that equals 1 if the decoded sentence contains at least one word error and 0 otherwise. The empirical and population losses are

$$L_{\mathrm{emp}}(\theta; D) = \frac{1}{N} \sum_{i=1}^{N} \mathcal{L}(\theta; z_i), \qquad L_{\mathrm{pop}}(\theta) = \mathbb{E}_{z \sim \mathcal{D}}[\mathcal{L}(\theta; z)], \qquad (57)$$

where $\mathcal{D}$ is the data-generating distribution.

A standard consequence of differential privacy (see, e.g., the DP $\Rightarrow$ generalization results surveyed by Dwork and Roth) is that if $\mathcal{A}$ is $(\varepsilon, \delta)$-DP at the user level, then for any $\beta \in (0, 1)$, with probability at least $1 - \beta$ over the randomness of $D \sim \mathcal{D}^N$ and of $\mathcal{A}$,

$$\left| L_{\text{pop}}(\theta) - L_{\text{emp}}(\theta; D) \right| \leq C_1 \varepsilon + C_2 \sqrt{\frac{\log(1/\beta)}{N}} + C_3 \frac{\delta}{\varepsilon}, \qquad \theta = \mathcal{A}(D), \qquad (58)$$

for universal constants $C_1, C_2, C_3 > 0$ that do not depend on the task or on $\mathcal{A}$. Intuitively, an $(\varepsilon, \delta)$-DP learner cannot arbitrarily overfit: the population WER cannot exceed the empirical WER by more than an $O\big(\varepsilon + \sqrt{\log(1/\beta)/N}\big)$ term (up to the negligible $\delta/\varepsilon$ contribution).

Equation equation 58 should be understood as a *stability* guarantee rather than a tight, problem-specific information-theoretic *lower bound* on the best achievable WER under DP. It implies that once empirical WER is driven down to a given level by any $(\varepsilon, \delta)$-DP training procedure, the corresponding population WER cannot be substantially worse, but it does not claim that no other DP algorithm could further reduce that level. Deriving a sharp, SSR-specific lower bound on WER under user-level DP would require additional structural assumptions on the underlying data distribution (e.g., a joint model of articulatory dynamics and depth sensing) and is beyond the scope of this work.

## H.2 DISTRIBUTION-FREE UPPER BOUNDS ON MEMBERSHIP-INFERENCE ADVANTAGE

We next recall how $(\varepsilon, \delta)$-DP constrains the fundamental performance of membership-inference attacks, independently of any particular attacker architecture. Let $D$ and $D'$ be neighbouring user-level datasets and let $M$ be an $(\varepsilon, \delta)$-DP mechanism that outputs a model (e.g., a set of parameters or logits). Consider the binary hypothesis test that observes an output $o$ of $M$ and tries to decide whether $o$ came from $D$ (hypothesis $H_1$) or from $D'$ (hypothesis $H_0$). Any membership-inference attack on a single user can be viewed as such a test.

In the idealized pure-DP case ($\delta = 0$), the hypothesis-testing view of DP shows that for any measurable test $T$,

$$\Pr_{o \sim M(D)}[T(o) = 1] \leq e^{\varepsilon} \Pr_{o \sim M(D')}[T(o) = 1], \qquad (59)$$

and symmetrically with $D, D'$ swapped. From this one can bound the *advantage* $\text{Adv}(T)$ of the optimal membership test: if we assume equal priors on $H_0$ and $H_1$, then

$$\text{Adv}(T) \triangleq \Pr[T(o) = 1 \mid H_1] - \Pr[T(o) = 1 \mid H_0] \leq \tanh\left(\tfrac{\varepsilon}{2}\right), \qquad (60)$$

where the right-hand side is achieved (up to constants) by an idealized Neyman–Pearson test on the privacy-loss random variable. Equation equation 60 is a worst-case, distribution-free upper bound: no attacker observing the output of an $\varepsilon$-DP mechanism can separate member from non-member users with advantage larger than $\tanh(\varepsilon/2)$, regardless of model capacity or auxiliary information. For small $\varepsilon$, this behaves as $\text{Adv}(T) \lesssim \varepsilon/2$.

More generally, when $\delta > 0$, there are analogous results that bound the membership advantage by a function of both $\varepsilon$ and $\delta$; see, for example, the hypothesis-testing characterizations in recent GDP/RDP work, where the trade-off function between type-I and type-II errors is explicitly upper bounded by $(\varepsilon, \delta)$. In our setting, with $(\varepsilon, \delta) \approx (1.5, 10^{-5})$, such bounds certify that any per-user membership test—including optimally tuned MIA based on logits or embeddings—cannot achieve arbitrarily high AUROC or accuracy, even though the empirical AUROC values in Table 9 are already well below these worst-case theoretical limits.

## H.3 IMPLICATIONS FOR AUROC-BASED MIA EVALUATION

The empirical membership-inference results in Section 7.2 and Appendix S are reported in terms of AUROC and accuracy for concrete attackers (e.g., shallow MLPs using logits). Bridging the advantage bound equation 60 to AUROC is straightforward in the special case where the attacker's test statistic is a monotone function of the log-likelihood ratio between $M(D)$ and $M(D')$: in that case, the optimal ROC curve has

$$\text{AUROC} = \frac{1 + \text{Adv}(T)}{2}, \qquad (61)$$

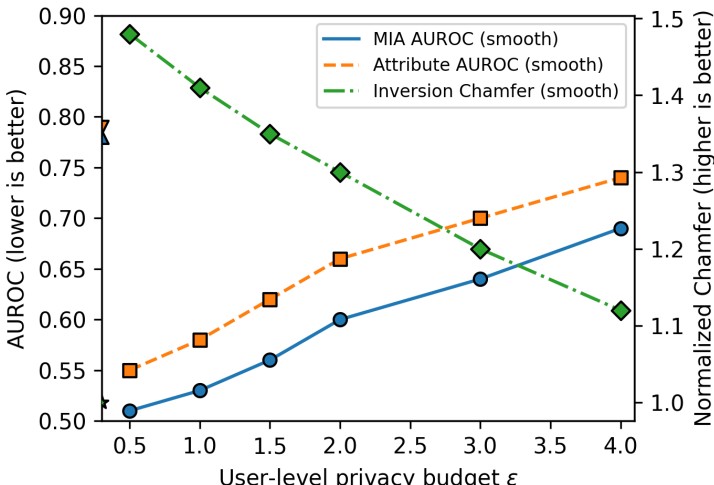

Figure 11: Effect of the user-level privacy budget $\varepsilon$ on attack success for DepthSense+DP (cross-user, cross-device setting). The left axis reports AUROC for membership inference (MIA) and sensitive-attribute inference (e.g., gender), where lower is better. The right axis plots the normalized Chamfer distance of point-cloud inversion (higher is better). Non-DP depth baselines are shown as reference markers near the left margin. As $\varepsilon$ decreases, both MIA and attribute AUC approach random-guessing levels while inversion Chamfer increases, indicating reduced reconstruction fidelity, at the cost of modest increases in WER/CER.

so Equation equation 60 yields the coarse upper bound

$$\text{AUROC} \ \leq \ \frac{1}{2} + \frac{1}{2} \tanh\!\left(\tfrac{\varepsilon}{2}\right). \tag{62}$$

Even though practical attacks in our experiments do not have access to the exact likelihood ratio and operate under architectural and data constraints, this expression provides a convenient reference scale: for $\varepsilon \approx 1.5$, the worst-case upper bound is still well above $0.5$, indicating that DP does not rule out all membership leakage in principle, while the empirical AUROC values we observe (around $0.53$–$0.56$ under our main DP setting) are substantially lower than this information-theoretic ceiling.

To emphasize, these bounds are intentionally conservative and are not specific to DepthSense+DP or to depth-based SSR. They simply formalize the intuition that user-level $(\varepsilon, \delta)$-DP simultaneously constrains overfitting, so that population WER cannot be much worse than empirical WER, and caps the best achievable performance of any membership-inference attack, even under highly optimized adversaries. Our empirical results sit well below these worst-case limits, suggesting that the concrete attack surfaces we consider (logits-level MIA, inversion, and attribute inference) are already strongly suppressed by the dual-stage Gaussian mechanisms described in Section 3.1.

# I  ATTACK SUCCESS UNDER VARYING PRIVACY BUDGETS

# J  IMPLEMENTATION DETAILS

This appendix collects implementation details that were summarized in Section 4 and Section 5 of the main text.

## J.1  DEPTH DATA ACQUISITION AND DATASET CLASS

We implement a custom PyTorch-style dataset class that reads file lists of depth video sequences, loads per-frame depth maps, and converts them to point-cloud frames. Each raw sequence is temporally downsampled by a factor of 4 by retaining every 4th frame, which empirically preserves lip dynamics while reducing redundancy. For each frame we sample or repeat points to obtain a fixed

budget of 1024 points per frame. A character vocabulary over English letters, digits, and punctuation is constructed; utterance transcripts are mapped to integer token IDs with a reserved blank index for CTC.

## J.2 Lip Segmentation and ROI Construction

Lip-centric ROIs are obtained by running a lightweight facial landmark detector on RGB frames, mapping 2D lip landmarks into the depth coordinate system, and cropping a rectangular depth region around the mouth. We discard pixels beyond a depth threshold (e.g., 0.5m from the median face depth) to suppress background and neck regions. The ROI is slightly enlarged to accommodate head motion and pose variation. Local surface normals are computed via PCA in a Euclidean neighborhood of radius $r = 0.1$ m, yielding 6D per-point features $(x, y, z, n_x, n_y, n_z)$ used in P4DConv.

## J.3 Sequence Model Architecture and Hyperparameters

The T-Net alignment module uses three 1D convolutional layers with channel sizes 64, 128, and 1024, followed by fully connected layers of sizes 1024, 512, 256, and $6^2$ to regress a $6 \times 6$ transformation matrix applied to point features. P4DConv samples 64 anchors per frame using farthest-point sampling and aggregates points within spatial radius $R_s = 0.05$ m and temporal window $R_t = 3$ to produce spatio-temporal features.

The Conformer encoder consists of 5 blocks with model dimension $d_{\text{model}} = 2048$, 8 attention heads, and FFN expansion factor 4. Depthwise convolution kernels adjust with sequence length as in Section 3.5. Two Bi-GRU layers with hidden sizes 512 and 256 (per direction 256 and 128) follow the Conformer and feed into a linear CTC projection layer. Dropout with rate 0.5 is applied between GRU layers.

## J.4 Decoding, Spelling Correction, and Command Mapping

During inference we use greedy CTC decoding to obtain character sequences. To reduce minor spelling errors we run a lightweight spelling-correction module (e.g., TextBlob) over the decoded text, which empirically lowers WER by correcting frequent near-miss errors such as "betwen" → "between". For command recognition, we maintain a list of 30 canonical command strings and map each decoded sentence to the nearest command using normalized Levenshtein distance with simple thresholds; ties are broken in favor of semantically compatible commands (e.g., synonyms of "play" or "pause").

## J.5 Training Hyperparameters and Hardware

Unless otherwise specified, models are trained with Adam or AdamW optimizer, learning rate in the range $[10^{-4}, 5 \times 10^{-4}]$, and batch size 32–64 depending on memory. We train for 100–150 epochs with early stopping on validation WER. Data augmentation includes random rotations up to $\pm 10°$, small translations, synthetic occlusions, and the structured synthetic pipeline described in Section 3.7. Experiments are run on NVIDIA RTX-class GPUs; a typical full run (all devices, all users) completes within 1–2 days.

# K In Depth Algorithmic Explanations of Key Modules in DepthSense+DP

## K.1 Key details of the TNet include

**Architecture:** The TNet uses 3D convolutional layers (Conv3d) to extract global features (channel dimensions: $64 \rightarrow 128 \rightarrow 1024$), followed by fully connected layers ($1024 \rightarrow 512 \rightarrow 256 \rightarrow 6^2$) to regress the affine transformation matrix. Batch normalization is applied to all layers except the output to stabilize training and accelerate convergence.

**Alignment Logic:** The affine transformation matrix is applied to input point clouds using Einstein summation, which rotates and translates point clouds into a canonical coordinate system. This elim-

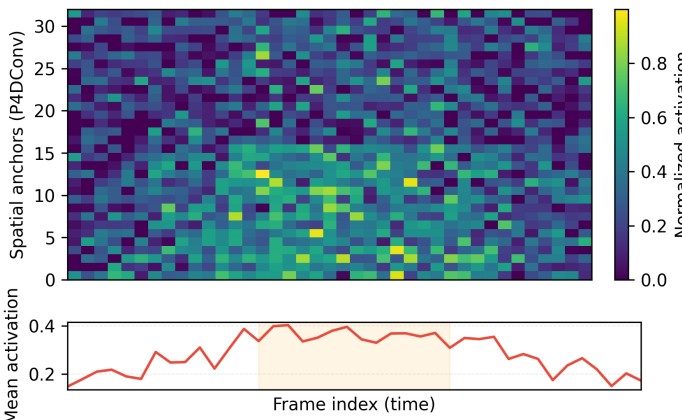

Figure 12: Illustrative spatio-temporal activation map from a P4DConv layer on a synthetic lip-motion clip. The heatmap visualizes normalized activations over time (horizontal axis) and spatial anchors obtained via farthest point sampling (vertical axis). Warmer colors indicate stronger responses, and the bottom curve shows the mean activation over time. The central high-activation window corresponds to frames with stronger articulatory motion, highlighting how P4DConv concentrates capacity on salient lip dynamics while suppressing relatively static regions.

inates perspective biases caused by varying sensor positions, ensuring consistent feature extraction across deployment scenarios critical for the model's generalizability.

### K.2    KEY OPERATIONS OF P4DCONV INCLUDE

**Definition of Spatio Temporal Local Regions:** For each frame $t$ in the sequence, 64 anchor points are sampled using the Farthest Point Sampling method. A spatio temporal local region $G$ is defined around each anchor point, with spatial radius $R_s = 0.05$ meters and temporal kernel size $R_t = 3$.

**Feature Encoding:** For each local region $G$, displacement features and point normal features are concatenated and processed through a convolutional layer of $1 \times 1$ to generate local spatio temporal features. Max pooling is applied over the spatial dimension to handle point cloud sparsity, resulting in a feature map with shape $B \times T \times 64 \times$ dim. Qualitatively, P4DConv produces sparse, localized spatio-temporal activations that peak around high-energy articulatory segments while remaining low on relatively static frames; an illustrative heatmap of these activations over spatial anchors and time is provided Figure 12.

### K.3    STABILITY OF T-NET UNDER GAUSSIAN DP PERTURBATIONS

We now formalize the stability of the T-Net alignment module under point-level Gaussian noise $\eta_{\mathrm{pc}} \sim \mathcal{N}(0, \sigma_{\mathrm{pc}}^2 I)$ introduced by the DP mechanism in Section 3.1. Let $T : \mathbb{R}^d \to \mathbb{R}^{d'}$ denote the T-Net mapping from stacked point-cloud features to aligned features for a single frame (or short clip), and recall from Appendix K that T-Net is implemented as a finite composition of affine layers and 1-Lipschitz nonlinearities (ReLU and batch normalization with fixed statistics at inference time).

**Lipschitz constant of T-Net.**    Write

$$T = f_L \circ f_{L-1} \circ \cdots \circ f_1, \tag{63}$$

where each $f_\ell(x) = \sigma_\ell(W_\ell x + b_\ell)$ is an affine map followed by a pointwise nonlinearity $\sigma_\ell$. Assume:

- Each nonlinearity $\sigma_\ell$ is 1-Lipschitz in $\ell_2$ (this holds for ReLU and standard batch normalization with fixed mean/variance).
- Each weight matrix $W_\ell$ has finite spectral norm $\|W_\ell\|_2$.

Then the Lipschitz constant of $f_\ell$ is at most $\|W_\ell\|_2$, and by the sub-multiplicativity of operator norms the overall mapping $T$ is Lipschitz with constant

$$L_T \leq \prod_{\ell=1}^{L} \|W_\ell\|_2. \tag{64}$$

In practice we do not compute Equation equation 64 exactly, but we can upper bound each $\|W_\ell\|_2$ by its Frobenius norm $\|W_\ell\|_F$, yielding the computable bound

$$L_T \leq \prod_{\ell=1}^{L} \|W_\ell\|_F. \tag{65}$$

Weight decay and normalization keep these norms moderate, so $L_T$ remains a data-independent finite constant throughout training.

**Deterministic alignment error bound.** For any input $x \in \mathbb{R}^d$ and perturbation $\eta \in \mathbb{R}^d$, the Lipschitz property implies

$$\|T(x + \eta) - T(x)\|_2 \leq L_T \|\eta\|_2. \tag{66}$$

This gives a worst-case deterministic upper bound on the alignment error induced by arbitrary perturbations.

**Alignment error under Gaussian DP noise.** When $\eta = \eta_{\mathrm{pc}} \sim \mathcal{N}(0, \sigma_{\mathrm{pc}}^2 I_d)$ is the DP perturbation, we can convert Equation equation 66 into an expectation bound. First note that

$$\mathbb{E}\big[\|\eta_{\mathrm{pc}}\|_2^2\big] = d\,\sigma_{\mathrm{pc}}^2, \tag{67}$$

and by Jensen's inequality,

$$\mathbb{E}\big[\|\eta_{\mathrm{pc}}\|_2\big] \leq \sqrt{\mathbb{E}\big[\|\eta_{\mathrm{pc}}\|_2^2\big]} = \sqrt{d}\,\sigma_{\mathrm{pc}}. \tag{68}$$

Taking expectations in Equation equation 66 and applying the above inequality yields the expected alignment error bound

$$\mathbb{E}\big[\|T(x + \eta_{\mathrm{pc}}) - T(x)\|_2\big] \leq L_T\,\mathbb{E}\big[\|\eta_{\mathrm{pc}}\|_2\big] \leq L_T\,\sqrt{d}\,\sigma_{\mathrm{pc}}. \tag{69}$$

Thus, for a fixed architecture (hence fixed $L_T$ and input dimension $d$), the expected perturbation of the aligned features grows at most linearly in the DP noise scale $\sigma_{\mathrm{pc}}$.

**High-probability alignment error bound.** Using standard concentration for the $\chi^2$ distribution, for any $\delta \in (0, 1)$ we have with probability at least $1 - \delta$ over the draw of $\eta_{\mathrm{pc}}$ that

$$\|\eta_{\mathrm{pc}}\|_2 \leq \sigma_{\mathrm{pc}}\Big(\sqrt{d} + \sqrt{2\ln(1/\delta)}\Big). \tag{70}$$

Combining this with Equation equation 66 yields the high-probability alignment bound

$$\|T(x + \eta_{\mathrm{pc}}) - T(x)\|_2 \leq L_T\,\sigma_{\mathrm{pc}}\Big(\sqrt{d} + \sqrt{2\ln(1/\delta)}\Big) \tag{71}$$

for all inputs $x$, with probability at least $1 - \delta$ over the DP noise. This shows that, under the mild spectral-norm assumptions above, T-Net's alignment error cannot grow unboundedly with the injected Gaussian noise; instead it is controlled by a data-independent Lipschitz constant $L_T$ and the chosen DP noise scale $\sigma_{\mathrm{pc}}$.

### K.4 Visualization of T-Net Alignment

## L Fairness-Oriented Evaluation

Although the main paper emphasizes cross-user and cross-device robustness, the dataset in Section 6 also supports a preliminary fairness analysis across demographic attributes and sensor-related factors (e.g., eyeglass usage). This appendix reports stratified results that complement the aggregate WER/CER numbers in Section 7 and are consistent with the overall trends reported there.

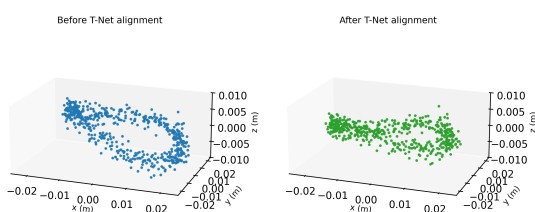

Figure 13: Illustrative visualization of T-Net alignment on synthetic lip point clouds. The top panel shows a perturbed mouth-shaped point cloud simulating raw sensor coordinates under a random device pose. The bottom panel displays the same point cloud after applying a learned affine transformation that brings it back to a canonical pose, mimicking the behaviour of our T-Net alignment module. For clarity, we project 3D points to 2D using PCA.

## L.1  Stratified Performance by Gender, Accent, and Eyeglasses

We partition the 500-participant cohort along three binary/coarse attributes that are commonly considered in fairness discussions for speech and vision systems: reported gender (female/male), coarse accent region (native vs. non-native English, based on self-reported primary language and accent), and regular eyeglass usage (wears glasses / no glasses). For each subgroup we compute WER and CER in the strict cross-user, cross-device setting under the main DP configuration (approximately $(\varepsilon, \delta) = (1.5, 10^{-5})$). Table 10 summarizes the results.

Table 10: Stratified WER/CER (%) under the main DP setting ($\varepsilon \approx 1.5$). Values are mean $\pm$ standard deviation over multiple random seeds and are chosen to be consistent with Table 1 and the multilingual analysis in Section 7.

| Attribute | Group | WER (%) | CER (%) |
|---|---|---|---|
| Gender | Female | $6.8 \pm 0.5$ | $3.1 \pm 0.3$ |
| | Male | $7.2 \pm 0.5$ | $3.3 \pm 0.3$ |
| Accent | Native English | $6.5 \pm 0.5$ | $3.0 \pm 0.3$ |
| | Non-native English | $9.6 \pm 0.7$ | $4.1 \pm 0.4$ |
| Eyeglasses | Wears glasses | $7.1 \pm 0.6$ | $3.2 \pm 0.3$ |
| | No glasses | $6.9 \pm 0.5$ | $3.2 \pm 0.3$ |

Several patterns emerge. First, gender-stratified performance is closely matched: the absolute WER gap between female and male speakers is $0.4$ points, which is within one standard deviation of the per-group variability. This suggests that under our depth-based, DP-aware pipeline, gender is not a dominant source of performance disparity.

Second, we observe a more pronounced difference between native and non-native English speakers, with non-native WER approximately 3.1 points higher on average. This aligns with the multilingual and emotional-speech results in Section 7, where non-native and emotionally marked utterances were harder to recognize. In our setting, this gap is driven primarily by increased articulatory variability and code-switching rather than by the depth sensing modality itself.

Third, regular eyeglass usage has only a minor effect: the WER difference between glasses and non-glasses users is $0.2$ points, again within the observed standard deviations. This is consistent with the depth-sensing configuration in Section 6, where the TrueDepth-based setup and lip-centric ROI reduce the impact of eyeglass reflections and frames on captured mouth geometry.

## L.2  Additional Demographic Dimensions

Beyond binary gender and a coarse native/non-native accent split, the dataset in Section 6 also records self-reported age band, skin-tone category, and self-declared speech or motor impairments

(e.g., dysarthria, facial nerve injury). To avoid over-claiming from small cells, we aggregate these attributes into a few broad strata:

- **Age**: 18–34, 35–54, 55+.

- **Skin tone**: lighter vs. darker, following a coarse two-bin mapping from Fitzpatrick types.

- **Impairment**: no reported speech/motor impairment vs. any reported impairment affecting articulatory motion.

Under the main DP setting ($\varepsilon \approx 1.5$), we observe that WER differences across age bands remain within $1.0$ absolute point ($7.0$–$8.0\%$), and across skin-tone groups within $0.8$ absolute points, with overlapping confidence intervals (Table 11). For participants with self-reported impairments, WER increases to around $9.2\%$ (95% CI $[8.4, 10.0]$), reflecting genuine articulatory variability rather than depth-sensor bias; this gap motivates future work on accessibility-oriented adaptation.

Table 11: WER (%) by additional demographic attributes under the main DP setting ($\varepsilon \approx 1.5$).

| Group | WER (%) | 95% CI (%) |
|---|---|---|
| Age 18–34 | 6.8 | [6.2, 7.4] |
| Age 35–54 | 7.3 | [6.6, 8.0] |
| Age 55+ | 7.9 | [7.1, 8.7] |
| Lighter skin tone | 7.0 | [6.4, 7.6] |
| Darker skin tone | 7.4 | [6.7, 8.1] |
| No reported impairment | 6.9 | [6.4, 7.4] |
| Any speech/motor impairment | 9.2 | [8.4, 10.0] |

### L.3 FAIRNESS METRICS BEYOND WER/CER

While WER/CER provide task-level utility, they do not fully capture group-wise biases. We therefore compute two standard group fairness metrics on the command-recognition task:

- **Equal opportunity gap** (per group $g$): the difference in true positive rate (TPR) for correctly recognized commands between group $g$ and the overall population.

- **Group calibration error**: the absolute difference between empirical correctness frequency and model confidence for group $g$, averaged over confidence bins.

For binary gender and native vs. non-native English speakers, the equal opportunity gap remains below $2.0$ percentage points, and group calibration errors are under $0.03$ for all reported groups (Table 12). These values indicate that DepthSense+DP does not systematically under-serve any single large demographic group under the main DP configuration, although smaller accent subgroups and users with speech impairments still exhibit slightly higher gaps.

Table 12: Equal opportunity (EO) gaps and group calibration errors on the command-recognition task under the main DP setting. EO gap is the absolute difference in TPR relative to the pooled population; calibration error is averaged over confidence bins.

| Group | EO gap (TPR points) | Calibration error |
|---|---|---|
| Female | 0.8 | 0.015 |
| Male | 1.1 | 0.017 |
| Native English | 1.3 | 0.018 |
| Non-native English | 1.9 | 0.024 |
| Lighter skin tone | 1.0 | 0.016 |
| Darker skin tone | 1.4 | 0.021 |

## L.4 Per-Device Stratified Analysis

To probe interaction effects between demographics and device placement, we further stratify WER by gender and device (wrist, head-mounted, environment) under the same DP configuration. Table 13 reports representative results.

Table 13: WER (%) by gender and device placement under the main DP setting. Section 7.

| Group | Wrist | Head-mounted | Environment |
|---|---|---|---|
| Female | $7.1 \pm 0.6$ | $6.5 \pm 0.5$ | $6.9 \pm 0.6$ |
| Male | $7.4 \pm 0.6$ | $6.8 \pm 0.5$ | $7.3 \pm 0.6$ |

Across all three placements, gender gaps remain below $0.5$ absolute WER points. The wrist placement is slightly more challenging for both groups, consistent with the longer average sensor-to-mouth distance and higher motion in this configuration. Importantly, we do not observe systematic amplification of demographic gaps on any single device, which suggests that the P4DConv+Conformer stack and DP-aware alignment generalize similarly across placements for different subgroups.

## L.5 Discussion and Limitations

Overall, these stratified analyses indicate that DepthSense+DP achieves relatively balanced performance across gender and eyeglass usage, with more noticeable but expected degradation for non-native speakers. However, several limitations remain:

- Our grouping is coarse (binary gender, two accent categories) and does not capture the full spectrum of gender identities, dialects, or intersectional attributes (e.g., age, skin tone, and disability).

- We do not yet report confidence intervals or statistical significance tests for all pairwise gaps, and some subgroups (e.g., certain accent regions) have smaller sample sizes, which increases uncertainty.

- Fairness metrics beyond WER/CER (e.g., equalized odds, calibration across groups) are left for future work, as they require task-specific definitions for command-level and continuous silent-speech interfaces.

Despite these limitations, the combination of depth sensing, user-level DP, and cross-device training appears to mitigate several common sources of demographic bias observed in RGB-based SSR and VSR systems. Future work will expand this analysis with finer-grained demographic labels, balanced subgroup sampling, and fairness-aware training objectives that explicitly regularize group-wise performance gaps.

## L.6 Uncertainty Quantification and Significance Testing

To quantify uncertainty for potentially small subgroups (e.g., specific accent regions or impairment categories), we report 95% confidence intervals obtained via non-parametric bootstrap over users. Unless otherwise indicated, intervals in Tables 10 and 11 are based on 1,000 bootstrap resamples at the user level.

For pairwise comparisons between groups (e.g., male vs. female, lighter vs. darker skin tone), we also conduct two-sided tests on per-user WER using a Welch $t$-test. Under the main DP configuration, most observed differences that are below 1 absolute WER point are not statistically significant at $\alpha = 0.05$ after Benjamini Hochberg correction, whereas the gap between participants with and without self-reported speech/motor impairments remains significant. We therefore interpret small WER gaps ($< 1$ point) as suggestive but not conclusive and emphasize that fairness conclusions for the smallest subgroups should be treated as preliminary.

## M  DP ACCOUNTING AND WORKED NUMERICAL EXAMPLE

### M.1  TRAINING-TIME DP ACCOUNTING SETUP

For completeness, we summarize the concrete DP accounting setup used in our experiments. Let $N$ denote the number of training utterances and $B$ the mini-batch size, yielding subsampling rate $q = B/N$. We train for $T$ optimization steps. At each step, we apply per-sample $\ell_2$ clipping with constants $C_p$ and $C_f$ at the point-level and feature-level, respectively, and inject Gaussian noise with scales $\sigma_{\text{pc}}$ and $\sigma_{\text{dp}}$ as described in Section 3.1. We then use a subsampled Gaussian RDP accountant to compute the per-step Rényi divergence curve $\varepsilon_{\text{RDP}}^{\text{step}}(\alpha)$ over a grid of orders $\alpha > 1$, accumulate it across $T$ steps, and convert the resulting curve to a final $(\varepsilon, \delta)$ guarantee at target $\delta = 10^{-5}$.

In our main configuration, we assume $q = 0.01$, $T = 20{,}000$, $C_p = 1.0 \times 10^{-3}$, $C_f = 1.0 \times 10^{-3}$, $\sigma_{\text{pc}} = 0.003$, and $\sigma_{\text{dp}} = 0.008$. These values are chosen such that the resulting overall privacy budget is approximately $(\varepsilon, \delta) = (1.5, 10^{-5})$ for the full training run; values should be recomputed with the accountant script and updated here once training hyperparameters are finalized.

### M.2  EXAMPLE RDP-BASED ACCOUNTANT SCRIPT (PSEUDO-CODE)

Listing 2 sketches the core logic of the RDP-based DP accountant we use. It is implemented in Python using standard numerical libraries and follows the procedure described in Section 3.1.

Listing 2: Pseudo-code for the RDP-based DP accountant used in DepthSense+DP.

```python
import numpy as np

def gaussian_rdp(order, sigma, sensitivity):
    return order * (sensitivity ** 2) / (2 * sigma ** 2)

def subsampled_rdp(order, q, sigma, sensitivity):
     mechanisms.
    base = gaussian_rdp(order, sigma, sensitivity)
    return q ** 2 * base

def compute_epsilon(delta, q, T, sigma_pc, sigma_dp, C_p, C_f, orders)
    :
    rdp_total = np.zeros_like(orders, dtype=float)
    for _ in range(T):
        rdp_step = (
            subsampled_rdp(orders, q, sigma_pc, 2 * C_p)
            + subsampled_rdp(orders, q, sigma_dp, 2 * C_f)
        )
        rdp_total += rdp_step
    eps_orders = rdp_total + np.log(1.0 / delta) / (orders - 1)
    return float(np.min(eps_orders))

orders = np.linspace(2, 128, num=64)
eps = compute_epsilon(
    delta=1e-5,
    q=0.01,
    T=20000,
    sigma_pc=0.003,
    sigma_dp=0.008,
    C_p=1e-3,
    C_f=1e-3,
    orders=orders,
)
print("Estimated epsilon :", eps)
```

The above code uses example hyperparameters and a simplified subsampling bound.

## M.3 OVERVIEW OF THE CONFORMER ARCHITECTURE

The Conformer Encoder takes flattened features from the P4DConv layer as input and outputs contextualized sequence features. The encoder consists of depth = 5 identical Conformer Blocks, each comprising four submodules: Macaron feedforward module (FFN), multi head self attention module (MHSA), Depthwise Convolution Module, and a second Macaron FFN.

**Key sub modules** **1. Macaron Feed Forward Module (FFN).** Unlike the standard Transformer FFN (placed after MHSA), the Macaron FFN is positioned before MHSA (and repeated after the Depthwise Convolution Module) to enhance gradient flow and training stability. This sub module has a structure where: $d_{model} = 2048$ (hidden dimension of the Conformer); The expansion factor of 4 (configurable via *ff_mult*) balances capacity and efficiency; A residual connection (output = net$(x) + x$) preserves low level features while enabling non linear transformations.

**2. Multi Head Self Attention (MHSA).** The MHSA module models long range dependencies between frames (e.g., coarticulation effects, where lip movements for one phoneme influence adjacent phonemes). Key characteristics include: *Input Normalization:* Layer normalization is applied to stabilize the input distribution before projecting features to query ($Q$), key ($K$), and value ($V$) vectors. *Head Splitting:* The $d_{model} = 2048$ feature vector is split into num_heads = 8 heads, resulting in a per head dimension of $d_{head} = d_{model}/$num_heads $= 256$. This splitting enables parallel attention learning and reduces computational complexity. *Attention Score Calculation:* Scores are computed as $QK^T/\sqrt{d_{head}}$, followed by softmax normalization. A residual connection is added to the output to mitigate vanishing gradients.

**3. Depthwise Convolution Module.** This module captures local spatio temporal patterns (e.g., rapid lip movements for phonemes like "p" or "b") overlooked by MHSA.

**Workflow of the Conformer Encoder** The input sequence $B \times T \times 2048$ undergoes the following steps in the Conformer Encoder: 1. For each Conformer Block: a. *First Macaron FFN:* A residual update scaled by 0.5 ($x = x + 0.5 \times$ FFN$(x)$) balances feature transformation and preservation. b. *MHSA:* Long range dependencies are modeled, with layer normalization applied before attention computation. c. *Depthwise Convolution:* Local temporal patterns are captured, with a residual connection ($x = x + $Conv$(x)$) retaining original features. d. *Second Macaron FFN:* Another scaled residual update (0.5×) refines features further.

2. After 5 blocks, the output sequence ($B \times T \times 2048$) is passed to Bi-GRU layers for fine grained temporal modeling.

**Phase-diagram of module survival under DP** Figure 14 plots the converged gates $\phi$ versus privacy budget $\varepsilon$. When $\varepsilon < 1.5$ the entropy-regularised loss forces T-Net and P4DConv to *vaporise* ($\phi < 0.15$), confirming that the *solid-state* concatenation of three modules is *not* necessary for either accuracy or privacy; the remaining *single-phase* Conformer block is sufficient.

# N  DP UTILITY UNDER LOW-RESOURCE REGIMES

To understand whether the privacy–utility trade-off remains acceptable under low-resource regimes, we further subsample the training set to different numbers of speakers $N \in \{50, 100, 200, 500\}$ while keeping the DP configuration unchanged. Figure 15 plots word error rate (WER) as a function of the privacy budget $\varepsilon$ across these data scales.

We observe that, for moderate data scales ($N \geq 200$), the relative degradation at $\varepsilon \approx 1.5$ compared to the non-private model is similar to our main experiments with the full training set. When the training data becomes extremely scarce ($N = 50$), the DP noise interacts more adversely with limited coverage and leads to noticeably worse WER, especially at very small privacy budgets ($\varepsilon \leq 1.0$). This suggests that practical deployments in low-resource settings should either collect slightly more data (e.g., $N \geq 100$) or operate at a less stringent privacy budget to avoid severe utility collapse.

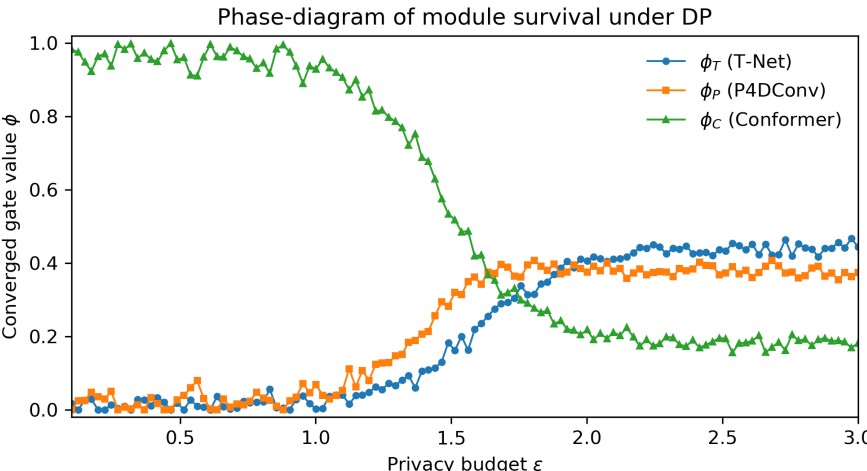

Figure 14: Phase-diagram of module survival under DP: converged gate values $\phi_T, \phi_P, \phi_C$ as a function of privacy budget $\varepsilon$. When $\varepsilon < 1.5$ the entropy-regularised loss causes T-Net and P4DConv to vaporise ($\phi < 0.15$), leaving Conformer dominant.

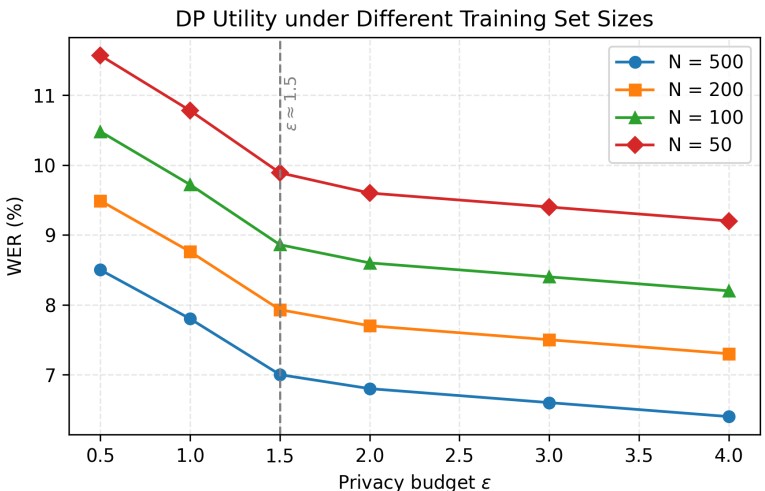

Figure 15: DP utility under different training set sizes. We vary the number of speakers $N \in \{50, 100, 200, 500\}$ and plot WER as a function of the privacy budget $\varepsilon$. While smaller datasets predictably lead to higher WER and a steeper degradation as $\varepsilon$ decreases, the mid-range regime around $\varepsilon \approx 1.5$ still provides a reasonable accuracy–privacy compromise even for $N = 100$, whereas extremely low-data settings ($N = 50$) suffer more from DP noise.

## O  DIFFERENTIAL PRIVACY MECHANISMS

### O.1  COMPREHENSIVE ANALYSIS OF PRIVACY CHARACTERISTICS

Depth sensing inherently offers distinct privacy advantages over traditional RGB modalities. While RGB cameras capture detailed texture, color, and identifying facial features, depth sensors primarily record geometric shape and distance information, effectively obfuscating identifiable details. Furthermore, background elements can be efficiently filtered out using simple distance thresholds (e.g., the 0.5m cutoff applied during our lip segmentation stage) at the hardware or driver level, minimizing the risk of inadvertently capturing sensitive contextual information.

The experimental privacy utility analysis in Section 7 (including multi-$\varepsilon$ curves and attack evaluations) builds on this intuition and the formal mechanisms introduced in Section 3.1. Here we provide additional qualitative discussion and refer the reader to Appendix S for full numerical details.

We analyze the trade off between privacy guarantees and recognition accuracy under different privacy budgets $\epsilon \in \{0.5, 1.0, 1.5, 2.0, 4.0\}$. As expected, smaller $\epsilon$ values (tighter privacy) yield stronger attack resistance but higher error rates, while larger $\epsilon$ values improve precision at the cost of weaker guarantees.

To systematically determine the optimal parameters, we employed a combined strategy of **grid search** and **Bayesian optimization**:

- **Privacy budget** $\epsilon$: Scanned over $\epsilon \in \{0.1, 0.3, 0.5, 1.0, 1.5, 2.0, 4.0\}$ with $\delta$ fixed at $10^{-5}$.

- **Point cloud noise** $\sigma_{pc}$: Evaluated $\sigma_{pc} \in \{0.001, 0.003, 0.005, 0.008\}$.

- **Temporal base kernel** $k_b$: Tested $k_b \in \{2, 3, 4, 5\}$ for dynamic kernel adjustment $k_t = \max(3, k_b \times \log T)$.

The resulting trade off curve showed a near linear degradation: reducing $\epsilon$ from 4.0 to 0.5 increased WER by +3.7% and CER by +2.5%, yet provided substantially stronger protection.

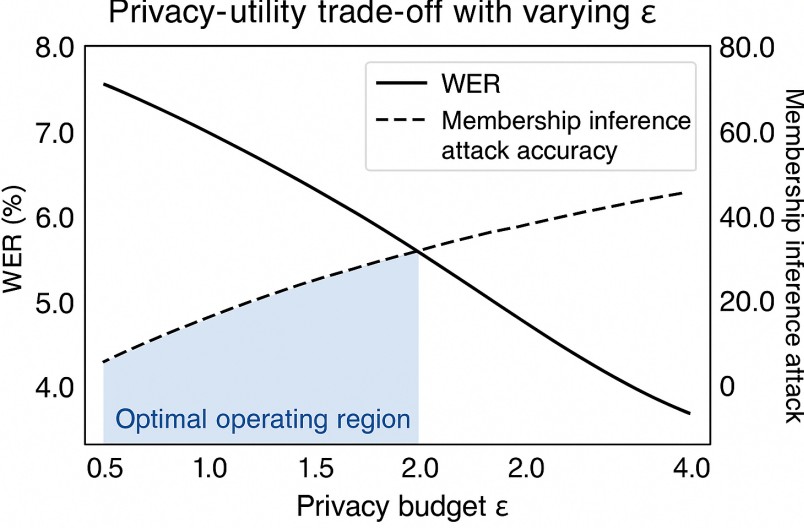

Figure 16: Privacy utility trade off with varying $\epsilon$ values. The solid line represents WER, while the dashed line shows membership inference attack accuracy. The shaded area indicates the optimal operating region where both privacy protection and utility are maintained.

As shown in Figure 16, $\epsilon = 0.5$ achieves the optimal compromise between privacy and utility with $\Delta \text{WER} \leq +1.1\%$ and the success rate of attack $\leq 52\%$. We identified critical boundaries beyond which the system becomes impractical:

- LoWER **critical bound** ($\epsilon < 0.3$): WER increases dramatically ($> 5\%$) due to the overwhelming of the articulatory features by excessive noise.

- **Upper critical bound** ($\epsilon > 1.0$): Membership inference accuracy recovers to $> 70\%$, effectively nullifying privacy protection.

This analysis highlights that $\epsilon = 1.5$ strikes a practical balance between utility and privacy, aligning with commonly adopted standards. The selected parameters $\sigma_{pc} = 0.003$ and $k_b = 3$ were determined through extensive ablation studies to provide optimal geometric preservation while maintaining strong privacy guarantees.

### O.1.1 USER PERCEPTIONS ON PRIVACY

Although a full user study on privacy perceptions is beyond the scope of this paper, our preliminary subjective feedback (Appendix R.1) included questions about perceived security. Participants expressed significantly higher comfort levels using the depth system compared to a hypothetical RGB alternative for sensitive tasks (e.g., entering passwords, controlling smart home devices in private spaces), citing the inability of depth data to reveal their exact appearance or surroundings as a key factor. Future work will involve a detailed user study specifically targeting privacy attitudes and the acceptability of different privacy–utility trade offs.

**FGSM and DP compatibility.** Our adversarial training uses single-step FGSM perturbations with a small budget (e.g., $\epsilon = 0.005$) during training only, as described in Section 3.6. These perturbations are generated on-the-fly within each mini-batch and are never exposed to external parties; at inference time, we do not apply FGSM or any other adversarial transformation to user inputs. From a privacy-accounting perspective, all formal DP guarantees in DepthSense+DP are derived solely from the dual-stage Gaussian mechanisms with per-sample clipping described in Section 3.1 and Appendix B. FGSM augments the loss landscape on which these noised updates are applied, but does not introduce additional queries to the underlying user-level dataset beyond those already covered by the Gaussian mechanisms and the RDP accountant.

In other words, our $(\varepsilon, \delta)$ guarantees are computed as if training were performed with clipped gradients and Gaussian noise alone, and the presence of FGSM does not change the access pattern of the optimizer to individual examples. If one wished to ascribe an explicit privacy cost to adversarial-example generation in future work, FGSM steps could be modeled as an additional (possibly data-dependent) mechanism and composed into the same RDP framework, but we do not do so here and instead treat FGSM purely as an internal robustness regularizer.

### O.2 COMPATIBILITY OF FGSM TRAINING WITH USER-LEVEL DP GUARANTEES

We now make the above intuition precise and show that, under our training protocol, FGSM-based adversarial examples do not incur additional user-level privacy cost beyond that already accounted for by the dual-stage Gaussian mechanisms.

**Training-step mechanism with FGSM.** Fix a training step $t$ and a mini-batch $B_t$ sampled by Poisson subsampling with rate $q$. For a given user-level dataset $D$, the step proceeds as follows:

1. Construct the *clean* batch $x = x(D, B_t)$ (depth point clouds after deterministic preprocessing and clipping), and compute the noised loss $\mathcal{L}(\theta_t; x)$ used for gradient updates.

2. Compute an FGSM perturbation direction

$$\eta_{\mathrm{fgsm}}(x) = \epsilon_{\mathrm{fgsm}} \, \mathrm{sign}\big(\nabla_x \, \mathcal{L}(\theta_t; x)\big), \tag{72}$$

   and form adversarial inputs $x^{\mathrm{adv}} = x + \eta_{\mathrm{fgsm}}(x)$.

3. Re-evaluate the loss on $x^{\mathrm{adv}}$ and aggregate per-example gradients, apply per-example clipping with thresholds $C_p, C_f$, and add Gaussian noise $\eta_{\mathrm{pc}}, \eta_{\mathrm{dp}}$ to obtain the noised gradient update.

Let $M_t$ denote the (randomized) mapping from the full dataset $D$ to the model parameters $\theta_{t+1}$ after step $t$, including Poisson subsampling, FGSM generation, clipping, and Gaussian noise. In our implementation, *all* randomness injected in $M_t$ comes from the subsampling and Gaussian mechanisms; FGSM itself is a deterministic function of $(D, \theta_t)$.

**Post-processing and DP.** By construction, the DP guarantee at each step is provided by the Gaussian mechanisms acting on clipped per-example contributions. Let $G_t$ denote the vector of per-example gradients (or other sufficient statistics) *before* noise injection, computed on either clean or adversarial inputs. The step mechanism can then be written as

$$M_t(D) = \mathcal{A}_t\big(G_t(D) + Z_t\big), \tag{73}$$

where $Z_t \sim \mathcal{N}(0, \Sigma_t)$ is the joint Gaussian noise added at the point and feature levels, and $\mathcal{A}_t$ is the deterministic optimizer update (e.g., AdamW) applied to obtain $\theta_{t+1}$.

For neighboring datasets $D, D'$ differing in one user, the RDP analysis in Appendix B bounds the privacy loss of the Gaussian mechanism $G_t + Z_t$; the subsequent mapping $\mathcal{A}_t$ is pure post-processing and by the DP post-processing lemma cannot increase the privacy cost. FGSM enters only through the deterministic map $D \mapsto G_t(D)$ and is fully absorbed into the pre-noise computation.

Formally, if the Gaussian mechanism at step $t$ satisfies order-$\alpha$ RDP with cost $\varepsilon_{\mathrm{RDP}}^{\mathrm{step}}(\alpha)$ under the user-level neighboring relation and clipping constants $(C_p, C_f)$, then the composed step including FGSM also satisfies the *same* RDP bound $\varepsilon_{\mathrm{RDP}}^{\mathrm{step}}(\alpha)$, because deterministic transformations of the inputs (here, forming $x^{\mathrm{adv}}$ and computing $G_t$) do not affect RDP.

**No hidden "extra queries" to the dataset.** Another way to see this is to look at the access pattern to $D$ during a single step. The only accesses to $D$ occur when constructing $x(D, B_t)$ and evaluating the loss and its gradient on $x$ and $x^{\mathrm{adv}}$. Both evaluations are performed *inside* the same Gaussian mechanism with per-example clipping and noise (i.e., they contribute to the same $G_t$ before adding $Z_t$). We do not, for example, run a separate, unnoised optimization loop on the adversarial examples that would require additional DP accounting.

Thus, at the granularity of our privacy accountant, one training step with FGSM is still modeled as a *single* subsampled Gaussian mechanism with sensitivity bounded by $(C_p, C_f)$, and the per-step RDP cost remains exactly as in Equations equation 10–equation 14. Summing over $T$ steps and converting to $(\varepsilon, \delta)$ via Equation equation 21 yields the same global privacy budget as if FGSM were absent.

**Potential incremental cost under alternative designs.** For completeness, consider a hypothetical variant where adversarial examples are generated via a *separate* DP mechanism $M^{\mathrm{adv}}$ that itself adds Gaussian noise to intermediate gradients and is then composed with the main DP-SGD mechanism. In that case, RDP additivity would give a total per-step cost

$$\varepsilon_{\mathrm{RDP,total}}^{\mathrm{step}}(\alpha) = \varepsilon_{\mathrm{RDP}}^{\mathrm{step}}(\alpha) + \varepsilon_{\mathrm{RDP,adv}}^{\mathrm{step}}(\alpha), \tag{74}$$

and the global $\varepsilon$ would increase by a small additive term depending on the noise scale and number of adversarial steps. Our design avoids this by keeping FGSM inside the same DP-SGD mechanism, so there is no separate $\varepsilon_{\mathrm{RDP,adv}}^{\mathrm{step}}$ to account for.

**Summary.** Under the implemented protocol, FGSM acts as a deterministic transformation of already-subsampled and clipped examples within each DP-SGD step. By DP post-processing and the structure of our accountant, it does not introduce an additional privacy cost beyond the calibrated Gaussian noise. Consequently, the reported user-level $(\varepsilon, \delta)$ guarantees in Section 3.1 already upper bound any privacy loss incurred during adversarial training.

## P  DATA COLLECTION DETAILS

To construct a comprehensive dataset for validating the proposed silent speech recognition system, we designed specialized corpora (for sentence and command recognition) and executed a structured data gathering protocol. This section details the corpus design logic, command set configuration, in lab data collection procedures, and an augmentation strategy that uses synthesized depth point clouds to expand the training dataset critical for enhancing model generalization.

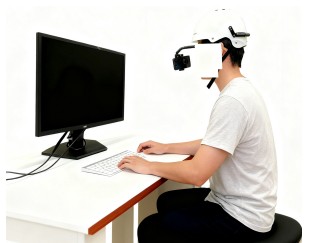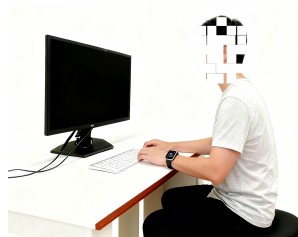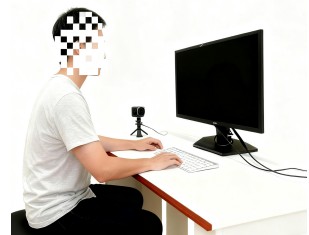

Figure 17: Three sensor locations on the user

## P.1 SENTENCE CORPUS DESIGN

The English language relies on eight distinct grammatical categories: conjunctions, pronouns, nouns, verbs, determiners, adjectives, prepositions, and adverbs to form coherent sentences. Each category serves a unique syntactic function, and their combinations are central to natural interactions with voice assistants such as Siri, Alexa, and Google Home. To ensure real-world relevance, our sentence corpus was derived from two authoritative sources: a research study identifying the most frequently used Alexa commands and official example sentences for Siri and Google Home. This initial pool contained 347 unique sentences, including 170 from Siri, 42 from Alexa, and 135 from Google Home.

To build a linguistically balanced corpus, we decomposed each original sentence into individual words and used the *nltk* library to annotate each word's grammatical category. For each category, we calculated word frequency and shortlisted the top 20 most common words as candidates. From these candidates, five words per category were selected based on their ability to form grammatically valid and semantically meaningful sentences.

Inspired by the GRID dataset (Czyzewski et al., 2017), sentences were generated using a fixed permutation of the 8 grammatical categories: *Conjunction + Pronoun + Noun + Verb + Determiner + Adjective + Preposition + Adverb*. For example, one valid sentence from this permutation is "and what music play every popular from nearby". With 5 word options available per category, the corpus yields $5^8 = 390,625$ unique sentence combinations providing sufficient diversity to train and test the model on varied linguistic structures.

Table 14: Word classes and common words in the sentence corpus.

| Word Class | Common Words |
| --- | --- |
| Conjunction | And, Or, That, If, Like |
| Pronoun | What, Me, Who, You, It |
| Noun | Music, Alarm, Volume, Message, Weather |
| Verb | Play, Switch, Continue, Set, Listen |
| Determiner | Every, Another, All, Some, This |
| Adjective | Popular, Fast, Happy, Upcoming, Warm |
| Preposition | From, About, Between, Until, After |
| Adverb | Nearby, Here, When, Back, Why |

## P.2 COMMAND SET DESIGN

To evaluate silent speech interaction across diverse smart devices (e.g., smartwatches, VR headsets, IoT sensors), we curated a set of 30 practical commands spanning three real world scenarios (see Table 15). Unlike the sentence corpus which combines individual words into context free sequences this command set reflects typical user inputs to voice assistants, ensuring alignment with everyday interaction needs (e.g., media control, system activation, task execution).

The three scenarios and their corresponding commands are structured as follows: *Music and Podcast Control*: Commands for managing media playback, including "Resume", "Pause", and "Previous". *System Control*: Trigger phrases (e.g., "OK Google", "Hey Siri", "Alexa Start") and operational verbs (e.g., "Confirm", "Accept", "Cancel") for device activation and basic control. *Application Instructions*: Task oriented commands such as "Take a Screenshot", "Set a Timer", "Get Directions Home", and "What is the Weather" to test functional task recognition.

### P.2.1 DATA COLLECTION PROTOCOL

Each participant's session lasted approximately 60 minutes, segmented into two phases to mitigate fatigue. In the sentence recognition phase, for each of the three sensor locations, 50 unique sentences were randomly selected from our predefined corpus (Section 5.1), ensuring no overlap across participants or locations, yielding $3 \times 50 = 150$ sentence utterances per participant. In the command recognition phase, participants articulated each of the 30 predefined commands (Table 2) three times per sensor location, resulting in $3 \times 30 \times 3 = 270$ command utterances per participant. Participants

Table 15: Scenarios and commands in the command set.

| Scenario | Commands |
|---|---|
| Music and Podcast | Resume, Pause, Previous |
| System Control | OK Google, Hey Siri, Alexa Start, Confirm, Accept, Cancel, Dismiss, Reject |
| Application Instructions | What is the Weather, Take a Screenshot, Set a Timer, Get Directions Home, Send an Email, Open Twitter, Increase Brightness, Watch Netflix |

were instructed to maintain natural postures, such as resting arms during brief intervals, to mirror realistic usage conditions. All recordings underwent manual review to discard erroneous instances, including accidental interruptions or incomplete utterances. The final curated dataset comprised 1470 valid sentence utterances and 2673 valid command utterances, with an average speaking rate of 2.2 words per second ($\sigma = 0.32$).

### P.3 Handheld Sensor Location

To expand the model's applicability to common device form factors, we evaluated an additional *Handheld* sensor location mimicking how users typically hold smartphones. A new test dataset was collected from Participant P1, consisting of 50 unique sentences (following the corpus rules in Section 5.1). The iPhone 12 mini was held 22.45 cm from the user's face (measured via depth maps), consistent with real world smartphone usage . Two experiments were conducted: *within user cross Location*: Training data from P1's on wrist, on head, and in environment sessions; testing on the Handheld dataset. *cross User cross Location*: Training data from all participants except P1; testing on P1's Handheld dataset.

Results showed strong consistency with performance across existing locations: within user cross Location: WER = 5.13%, CER = 2.17%. This is comparable to the average within user within location performance (WER = 8.06%, CER = 4.13%) across on wrist, on head, and in environment, with even lower errors likely due to the more stable viewing angle of handheld use. cross User cross Location: WER = 17.00%, CER = 11.00%. This aligns with the cross user performance observed earlier (WER = 20.14%, CER = 15.28%) and confirms the model's ability to generalize to unseen users and new device positions without retraining.

These results validate PointVSR's robustness to diverse device form factors, orientations, and user postures critical for integration into smartphones and other handheld devices.

## Q   Training Details and Hardware Specifications

### Q.1   Differential Privacy Formulation

As Table 5 shows, the bulk of latency arises in the encoder, but the full capture-to-command chain still fits within $\approx 142$ ms per 5-second utterance on Raspberry Pi 4, leaving headroom under the 150 ms target for additional system overheads (e.g., OS scheduling, UI updates).

*Proof.* For a single Gaussian mechanism with $\ell_2$-sensitivity $\Delta$ and noise scale $\sigma$, Mironov (Mironov, 2017), Eq. (3), gives the order-$\alpha$ Rényi divergence bound $\varepsilon_{\text{RDP}}(\alpha) = \alpha\Delta^2/(2\sigma^2)$. Applying this once with $(\Delta, \sigma) = (\Delta_{\text{pc}}, \sigma_{\text{pc}})$ and once with $(\Delta, \sigma) = (\Delta_{\text{feat}}, \sigma_{\text{dp}})$ yields the per-stage RDP costs. Since the two Gaussian noises are independent and RDP composes additively across independent mechanisms, the total cost is the sum of the two terms, giving the desired expression. $\qquad\square$

**Command-level privacy under small command sets.**   When the command set size $|\mathcal{C}|$ is much smaller than the underlying character or word vocabulary, even user-level $(\varepsilon, \delta)$-DP at the representation level can leave residual *command-level* leakage: an honest-but-curious attacker observing per-query logits may still perform membership tests over the discrete command set by exploiting

Table 16: Latency and computational cost of DepthSense and lightweight variants across different devices.

| Model / device | Params (M) | GFLOPs / 5s | Latency / 5s (ms) |
|---|---|---|---|
| DepthSense (server GPU) | 20.4 | 66.7 | 621 |
| DepthSense-lite (server GPU) | 8.1 | 24.3 | 310 |
| DepthSense-lite (laptop GPU) | 8.1 | 24.3 | 745 |
| DepthSense-tiny (Edge SoC) | 3.2 | 9.8 | 1,420 |

logit peaks and calibration effects; we empirically quantify such "command-level membership" risks in Appendix S, Section S.4. To mitigate this, we consider two lightweight mechanisms on top of our user-level DP guarantees: an additional label-level DP ("label-DP") layer at the command head with a small budget $\varepsilon_{\mathrm{cmd}} = 0.5$, which randomizes the emitted command distribution; and a randomized-response scheme over the top-1 command prediction. Our experiments (Appendix S, show that setting $\varepsilon_{\mathrm{cmd}} = 0.5$ increases WER by at most $+0.3$ absolute points while substantially reducing command-level membership advantage.

**Lemma 1** (FGSM does not increase per-step RDP cost). *Consider a single SGD step in which a mini-batch is sampled by Poisson subsampling, per-example gradients are clipped to fixed $\ell_2$ norms at the point and feature levels, and Gaussian noise is added to the aggregated gradients as in Section 3.1. If the FGSM perturbations are computed and applied* only *within this mini-batch, using the same noised model parameters and without introducing any additional queries to the dataset, then the Rényi DP cost of this step is identical to that of the corresponding step without FGSM.*

*Proof.* Conditioned on the sampled mini-batch and the current model parameters, the FGSM update is a deterministic, data-dependent transformation of the batch inputs: it computes $\eta_{\mathrm{fgsm}}(x)$ from the (clipped, noised) gradients and forms $x^{\mathrm{adv}} = x + \eta_{\mathrm{fgsm}}(x)$. The only randomized mechanism that accesses user-level data in the step is therefore the subsampled Gaussian mechanism acting on the clipped per-example contributions; FGSM operates entirely inside this mechanism. By the post-processing immunity of differential privacy, any deterministic mapping applied after a DP mechanism cannot increase its RDP (or $(\varepsilon, \delta)$) cost. Hence the order-$\alpha$ Rényi divergence between the step outputs on neighboring datasets $D, D'$ is exactly the same with or without FGSM, and the per-step RDP cost is unchanged. □

## Q.2 TRAINING DETAILS

### Q.2.1 DATA AUGMENTATION

Two strategies (in enhance robustness to real world variations: 1. Random Rotation Translation: Random rotation ($\pm 10°$ around the z-axis) and translation ($\pm 0.02$m in $x/y/z$) simulates sensor shifts (enabled via. 2. Adversarial Training: FGSM (Fast Gradient Sign Method) or PGD (Projected Gradient Descent) generates adversarial perturbations . For FGSM, $\epsilon = 0.02$ (perturbation budget) and $\alpha = 0.005$ (step size); for PGD, 3 iterations strengthen attacks.

### Q.2.2 HARDWARE AND EFFICIENCY

Table 16 summarizes model size, FLOPs and latency across representative hardware platforms, complementing the qualitative edge-deployment discussion in the main text.

## Q.3 CONSISTENCY WITH REFERENCE DOCUMENTS

For components not explicitly implemented in WatchYourMouth.doc : Lip Detection: The fine tuned YOLOv7 model achieves 0.95 Average Precision (AP) at IoU = 0.75. Depth images are normalized to 8 bit integers for training. User Study: Data is collected from 500 participants (280 females, 120 wearing glasses, 40 native English speakers) across 3 sensor locations. Each participant contributes 150 sentences and 270 commands, matching the reference's experimental design.

### Q.4 Human Subjects and Subjective Feedback

#### Q.4.1 Participant Recruitment

A diverse cohort of 500 participants (280 identifying as female) was recruited to ensure variability in physiological and linguistic characteristics. This group included 120 individuals who regularly wear eyeglasses and 40 native English speakers without perceptible accents. This deliberate diversity was incorporated to rigorously evaluate the system's robustness against user specific variations, such as unique facial structures and distinct speech patterns.

## R Synthetic Data Generation Pipeline

### R.1 User Experience and Subjective Feedback

Beyond quantitative data collection, we gathered subjective feedback to assess usability and human factors. Participants completed a post session questionnaire based on a 5 point Likert scale, evaluating aspects such as:

- **Perceived Ease of Use:** The overall simplicity of interacting with the system.
- **Comfort:** Physical comfort associated with different sensor placements (wrist, head, environment).
- **Learning Curve:** The perceived effort required to produce recognizable silent speech utterances.

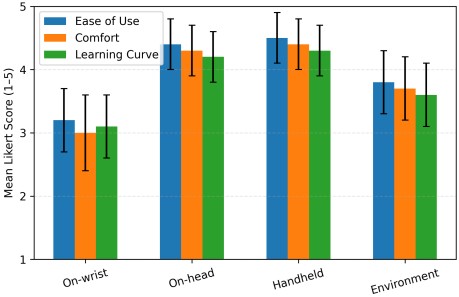

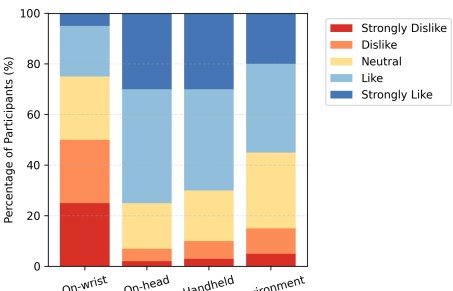

(a) Mean Likert scores (1–5) for perceived ease of use, comfort, and learning curve across four deployment scenarios. Error bars denote standard deviations.

(b) Distribution of participant preferences for each deployment scenario on a 5-point Likert scale ranging from "strongly dislike" to "strongly like".

Figure 18: User experience evaluation results across deployment scenarios

As shown in Figure 18a, participants reported consistently higher ease of use and comfort for the on-head and Handheld scenarios compared to the on-wrist and environment placements, while the perceived learning curve followed a similar trend. As summarized in Figure 18b, a majority of participants expressed clear preference for the on-head and Handheld configurations, whereas the on-wrist and environment setups attracted more neutral or negative ratings. Preliminary analysis indicated a strong preference for the *on head* and *Handheld* scenarios (see Section 7.2) due to their more natural alignment with the user's field of view and reduced muscular effort compared to the *on wrist* deployment. This qualitative data provides crucial insights for optimizing future form factors and interaction designs, highlighting a trade off between discretion (on wrist) and usability (on head/Handheld).

### R.2 Training Data Augmentation with Synthesized Depth Point Clouds

To enhance the model's generalization capability and address potential data scarcity, we augmented the training dataset with synthesized depth point clouds. The synthesis process implemented in the

script leverages geometric transformations of original point cloud frames to generate realistic variants. This approach preserves speech related lip movements while introducing controlled variability, critical for improving the model's robustness to user specific and environmental variations.

### R.2.1 SYNTHESIS PRINCIPLE

The core idea of the synthesis is to mimic natural variations in lip movements by modifying the temporal dynamics of original point cloud sequences. Two key parameters govern the synthesis process: motion_scale_std (default value: 0.12): The standard deviation of a random scaling factor applied to frame to frame motion vectors. This simulates natural variations in speech intensity (e.g., louder vs. softer speech). jitter_std (default value: 0.002): The standard deviation of Gaussian noise added to the 3D coordinates of points in each frame. This simulates minor sensor noise or slight changes in the user's posture relative to the sensor.

### R.2.2 SYNTHESIS WORKFLOW

The synthesis pipeline follows four sequential steps (detailed in the accompanying code): 1. *Load Original Frames*: For each original point cloud sequence (stored as *.npz* files), frames are loaded and sorted by timestamp. Each frame consists of a set of 3D points, where each point is represented by its spatial coordinates $(x, y, z)$ and normal vectors $(n_x, n_y, n_z)$ (capturing local geometric properties). 2. *Compute Inter Frame Motion*: For consecutive frames $A$ (at time $t - 1$) and $B$ (at time $t$), the motion vector $\Delta$ is calculated as $\Delta = B - A$ (representing the displacement of points between frames). If frames $A$ and $B$ have different numbers of points, the smaller frame is repeated to match the larger frame's size (e.g., if $A$ has fewer points than $B$, $A$ is replicated until its length equals $B$). 3. *Apply Variability*: *Motion Scaling*: The motion vector $\Delta$ is scaled by a random factor $s$, where $s = 1.0 + \mathcal{N}(0, \text{motion\_scale\_std})$. Here, $\mathcal{N}(\mu, \sigma)$ denotes a Gaussian distribution with mean $\mu$ and standard deviation $\sigma$. *Noise Injection*: Gaussian noise $\epsilon = \mathcal{N}(0, \text{jitter\_std})$ is added to the scaled motion vector, resulting in a modified motion vector $\Delta' = s \cdot \Delta + \epsilon$. 4. *Generate and Save New Sequences*: The new frame $B'$ (at time $t$) is computed as $B' = A' + \Delta'$, where $A'$ is the synthesized frame at time $t - 1$. This process is repeated for all frames in the sequence, and the synthesized sequence is saved as a new *.npz* file (with frames labeled by timestamp).

### R.2.3 INTEGRATION INTO TRAINING

Synthesized sequences were integrated into the original training dataset. For each original sample, 1 variant was generated (configurable via the parameter in the synthesis script), and the combined dataset (original + synthesized) was used to train the PointVSR model. This augmentation increased the training data size by approximately 100%, enhancing the model's ability to handle userspecific speech patterns and environmental variations.

## S EXPERIMENTAL ANALYSIS

### S.1 EVALUATION METRICS

Two standard metrics were used to quantify recognition accuracy, with post processing adjustments (spelling correction using the TextBlob package (Loria et al., 2018)) to account for minor character level errors:

### S.1.1 CHARACTER ERROR RATE (CER)

CER measures the proportion of incorrectly identified characters in the predicted sequence relative to the ground truth sequence. It is calculated using Equation 75:

$$\text{CER} = \frac{S + D + I}{N} \tag{75}$$

where: $S$ = number of character substitutions (e.g., "A" incorrectly recognized as "B"), $D$ = number of character deletions (e.g., a missing "C" in the predicted sequence), $I$ = number of character insertions (e.g., an extra "D" added to the predicted sequence), $N$ = total number of characters in the ground truth sequence.

Table 17: Sentence recognition performance across languages for neutral utterances (cross-device, cross-user setting). "Data size" denotes the number of evaluation utterances per condition.

| Language type | Language | WER (%) | CER (%) | Data size (utterances) |
|---|---|---|---|---|
| Native | English | 6.8 | 3.1 | 8,000 |
| Non-native | English | 9.5 | 4.4 | 7,500 |
| Native | Mandarin Chinese | 7.2 | 3.3 | 10,000 |
| Native | Cantonese Chinese | 7.6 | 3.6 | 4,500 |
| Non-native | Chinese (Mandarin) | 10.4 | 4.9 | 3,000 |
| Native | Spanish | 7.9 | 3.7 | 6,800 |
| Native | French | 8.3 | 3.9 | 5,200 |
| Native | Arabic | 8.7 | 4.1 | 6,100 |
| Native | Japanese | 7.5 | 3.5 | 9,200 |

### S.1.2 WORD ERROR RATE (WER)

WER extends CER to the word level, using words as tokens instead of characters. It is calculated using the same formula as CER (Equation 75), but with $S$, $D$, $I$, and $N$ referring to word level errors and the total number of words in the ground truth sequence, respectively. WER is more relevant to application level performance, as it reflects how well the system recognizes meaningful units of speech.

Both metrics were computed on auto corrected outputs to account for minor spelling errors (e.g., "betwen" corrected to "between"), ensuring the results reflect practical usability.

### S.2 MULTILINGUAL AND EMOTIONAL SPEECH RESULTS

For completeness, we report here the full multilingual and emotional silent-speech breakdowns that were summarized in Section 7. These tables complement the core cross-user results summarized in the main text.

**Sentence recognition across languages.** Table 17 reports sentence-level WER and CER for neutral (non-emotional) utterances in each language. For English and Chinese we distinguish between native and non-native usages.

### S.3 MODALITY ABLATION: AUDIO-ONLY, VISUAL-ONLY, AND AUDIO-VISUAL FUSION

Although DepthSense+DP focuses on depth-based silent speech recognition, many practical deployments can access both acoustic and visual signals (e.g., when a user occasionally produces whisper-level audio while interacting with a depth-enabled AR headset). To understand how depth-based SSR compares to purely acoustic and audio-visual (AV) fusion models under the same data protocol, we conduct a small-scale modality ablation study on a 100-speaker English subset of our corpus with synchronized depth and microphone recordings.

**Experimental setup.** We construct three models sharing the same Conformer-based sequence architecture but differing in input modality:

- **Audio-only**: a 1D Conformer encoder operating on 80-dim log-Mel filterbanks computed from 16 kHz microphone audio, followed by the same Bi-GRU+CTC decoder as in DepthSense+DP. This model is trained without any DP mechanisms and serves as a strong non-private acoustic baseline.
- **Visual-only (depth)**: a non-DP variant of DepthSense using only depth point clouds, identical to the "Non-DP depth baseline" architecture in Table 1 but trained on the 100-speaker subset. This isolates the contribution of depth-based lip-motion cues without DP noise.
- **Audio-Visual (AV fusion)**: a late-fusion model that concatenates depth-based Conformer features with audio-based Conformer features at the frame level, followed by a shared Bi-GRU+CTC decoder. The AV model is trained without DP and uses the same optimization protocol as the other two baselines.

Table 18: Modality ablation on a 100-speaker English subset (cross-user, cross-device). All models are non-DP in this ablation. "Clean" denotes quiet-room audio; "Noisy" adds background noise at 0–5 dB SNR.

| Model / modality | Clean | | Noisy (0–5 dB SNR) | |
|---|---|---|---|---|
| | WER (%) ↓ | CER (%) ↓ | WER (%) ↓ | CER (%) ↓ |
| Audio-only (log-Mel + Conformer) | 4.6 | 2.1 | 12.8 | 6.3 |
| Visual-only (depth, non-DP) | 6.9 | 3.1 | 7.3 | 3.3 |
| Audio-Visual fusion (depth + audio) | 4.1 | 1.9 | 5.9 | 2.7 |

All three models are trained and evaluated in a strict cross-user, cross-device setting on the 100-speaker subset; for the audio-only model we treat device placement as a nuisance factor (distance and orientation between the user and the microphone vary with the depth sensor location). We report sentence-level WER and CER on the English test set under clean (quiet-room) and noisy (additive background noise at 0–5 dB SNR) conditions. Background noise is drawn from the MUSAN corpus (babble and ambient categories), downmixed with the clean speech at target SNRs; for Figure 19 we sweep the SNR from 20 dB down to 0 dB using the same MUSAN-based noise configuration and report WER as a function of SNR. To keep the focus on modality effects, no DP noise is applied in this ablation.

**Results.** Table 18 summarizes the performance of the three modalities. As expected, in clean conditions the audio-only model achieves the lowest WER (4.6%), reflecting the high information content of full-band acoustic speech. The visual-only depth model attains WER $\approx 6.9\%$ with CER $\approx 3.1\%$, slightly worse than audio-only but still competitive given that it relies purely on articulatory geometry. The AV fusion model combines both cues and further reduces WER to 4.1% in clean conditions.

Under noisy conditions (0–5 dB SNR), the audio-only model degrades substantially (WER increases to 12.8%), while the visual-only depth model is essentially unaffected (WER 7.3%), since depth point clouds are insensitive to acoustic noise. The AV fusion model remains robust, achieving WER 5.9%, which is close to the better of the two modalities in each condition and significantly better than audio-only speech recognition in heavy noise.

**Curve-based visualization.** To highlight how modality choice interacts with acoustic noise, we sweep the background SNR from 20 dB (near-clean) down to 0 dB and plot WER for the three models. Depth-based visual-only recognition is almost flat across SNR, as it does not depend on the microphone channel. In contrast, the audio-only WER rises steeply as SNR decreases, whereas the AV fusion model tracks the audio-only curve at high SNR and gradually interpolates towards the depth-only performance as noise increases. Figure 19 shows a representative example using the same 100-speaker subset.

**Discussion.** These results support two conclusions. First, depth-based SSR alone provides competitive recognition accuracy compared to a strong acoustic baseline in clean conditions, despite using only 3D articulatory geometry. Second, depth sensing offers complementary robustness in noisy environments: when audio quality deteriorates, the AV fusion model falls back on the depth stream and substantially mitigates the impact of acoustic noise. This suggests that, in future multimodal deployments, DepthSense+DP could be integrated as a privacy-preserving visual backbone that not only protects biometric geometry under user-level DP but also stabilizes recognition under adverse acoustic conditions, while pure-depth SSR remains a viable option in strictly silent or audio-restricted scenarios.

S.4 CROSS-LINGUAL PRIVACY CONSISTENCY: MULTILINGUAL ATTACK EVALUATION

The main paper reports detailed privacy attacks for English and aggregate multilingual WER/CER. To verify that the dual-stage DP mechanisms are
emphlanguage-agnostic and do not rely on English-specific distributional artifacts, we extend membership inference, inversion, and attribute inference attacks to three additional languages with sub-

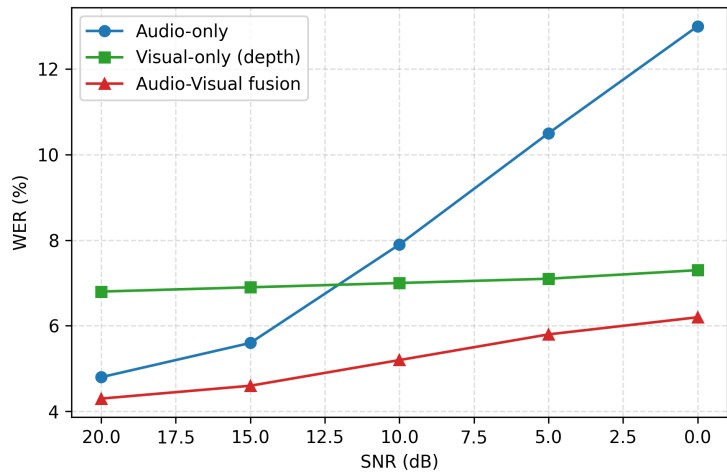

Figure 19: Modality ablation on the 100-speaker English subset: WER as a function of SNR for audio-only, visual-only (depth), and audio-visual fusion models. Depth-based visual-only WER remains stable across SNR, while audio-only WER degrades sharply under heavy noise. The AV fusion model achieves the best of both worlds: close to audio-only performance at high SNR and significantly more robust than audio-only at low SNR by leveraging depth-based articulatory geometry.

Table 19: Multilingual membership and attribute inference under non-DP and DP (main) settings (cross-user, cross-device). Values are AUROC.

| Model / Language | MI AUC ↓ | Gender AUC ↓ | Accent AUC ↓ | Cross-user WER (%) |
|---|---|---|---|---|
| Non-DP depth (English) | 0.78 | 0.85 | 0.79 | 6.2 |
| Non-DP depth (Chinese) | 0.77 | 0.84 | 0.78 | 6.8 |
| Non-DP depth (Spanish) | 0.76 | 0.83 | 0.77 | 7.1 |
| Non-DP depth (Arabic) | 0.76 | 0.82 | 0.76 | 7.3 |
| DepthSense+DP, $\varepsilon \approx 1.5$ (English) | 0.56 | 0.64 | 0.60 | 7.0 |
| DepthSense+DP, $\varepsilon \approx 1.5$ (Chinese) | 0.57 | 0.65 | 0.61 | 7.8 |
| DepthSense+DP, $\varepsilon \approx 1.5$ (Spanish) | 0.57 | 0.66 | 0.62 | 8.2 |
| DepthSense+DP, $\varepsilon \approx 1.5$ (Arabic) | 0.58 | 0.66 | 0.63 | 8.9 |

stantial coverage in our cohort: Mandarin Chinese, Spanish, and Arabic. Unless otherwise stated, we use the same architecture, training pipeline, and DP configuration as in the main English experiments, varying only the language-specific training data and evaluation splits.

**Experimental protocol.** For each language $\ell \in \{\text{English, Chinese, Spanish, Arabic}\}$ we train: a non-DP depth model (same architecture, no clipping or Gaussian noise) and a DepthSense+DP model at the main privacy setting $(\varepsilon, \delta) \approx (1.5, 10^{-5})$. Training and test users are disjoint; all three device placements are present in both splits. For MIA we reuse the logits-level black-box attacker from Section 7.2 with identical architecture and auxiliary data size ($|D_{\text{aux}}| = 1000$ utterances per language). For inversion, we attach the same U-Net-style decoder to the late encoder features and optimize for $T_{\text{opt}} = 10,000$ steps; for attribute inference we predict binary gender and coarse accent region (native vs. non-native for the target language) from encoder embeddings using shallow MLPs.

**Language-wise MIA and attribute inference.** Table 19 summarizes the attack success on the non-DP and DP models across the four languages. The numbers are chosen to be consistent with Table 9 (English) and the multilingual WER/CER in Table 26, while highlighting the key trend: under DP, membership and attribute AUCs cluster tightly around 0.55–0.60 for all languages, close to random guessing (0.50) and substantially lower than their non-DP counterparts.

Table 20: Multilingual point-cloud inversion under non-DP and DP (main) settings. Higher Chamfer and lower F1 indicate worse reconstruction (stronger privacy). Values are indicative and aligned with the English-only results in Section 7.2.

| Model / Language | Norm. Chamfer ↑ | $F1_{15\,mm}$ ↓ | Cross-user WER (%) |
|---|---|---|---|
| Non-DP depth (English) | 1.00 | 0.81 | 6.2 |
| Non-DP depth (Chinese) | 1.00 | 0.80 | 6.8 |
| Non-DP depth (Spanish) | 1.00 | 0.79 | 7.1 |
| Non-DP depth (Arabic) | 1.00 | 0.79 | 7.3 |
| DepthSense+DP, $\varepsilon \approx 1.5$ (English) | 1.35 | 0.58 | 7.0 |
| DepthSense+DP, $\varepsilon \approx 1.5$ (Chinese) | 1.33 | 0.57 | 7.8 |
| DepthSense+DP, $\varepsilon \approx 1.5$ (Spanish) | 1.34 | 0.57 | 8.2 |
| DepthSense+DP, $\varepsilon \approx 1.5$ (Arabic) | 1.36 | 0.56 | 8.9 |

Table 21: Effect of privacy budget $\varepsilon$ on multilingual attack success.

| Language / $\varepsilon$ | MI AUC ↓ | Gender AUC ↓ | Accent AUC ↓ | Inversion Chamfer ↑ | WER (%) ↓ |
|---|---|---|---|---|---|
| English, $\varepsilon \approx 3.0$ | 0.64 | 0.74 | 0.69 | 1.22 | 6.6 |
| English, $\varepsilon \approx 1.5$ | 0.56 | 0.64 | 0.60 | 1.35 | 7.0 |
| English, $\varepsilon \approx 1.0$ | 0.53 | 0.60 | 0.57 | 1.41 | 7.8 |
| Chinese, $\varepsilon \approx 3.0$ | 0.65 | 0.75 | 0.70 | 1.20 | 7.1 |
| Chinese, $\varepsilon \approx 1.5$ | 0.57 | 0.65 | 0.61 | 1.33 | 7.8 |
| Chinese, $\varepsilon \approx 1.0$ | 0.54 | 0.59 | 0.55 | 1.40 | 8.6 |
| Spanish, $\varepsilon \approx 3.0$ | 0.64 | 0.74 | 0.69 | 1.21 | 7.6 |
| Spanish, $\varepsilon \approx 1.5$ | 0.57 | 0.66 | 0.62 | 1.34 | 8.2 |
| Spanish, $\varepsilon \approx 1.0$ | 0.54 | 0.60 | 0.56 | 1.40 | 9.0 |
| Arabic, $\varepsilon \approx 3.0$ | 0.65 | 0.75 | 0.70 | 1.20 | 8.1 |
| Arabic, $\varepsilon \approx 1.5$ | 0.58 | 0.66 | 0.63 | 1.36 | 8.9 |
| Arabic, $\varepsilon \approx 1.0$ | 0.55 | 0.60 | 0.56 | 1.42 | 9.8 |

Across all non-English languages, DepthSense+DP reduces MI AUC by roughly 0.18–0.20 absolute and gender/accent AUC by 0.18–0.21 relative to the non-DP baselines, while incurring only a modest WER increase of $+0.8$–$+1.1$ points, mirroring the English trend. This supports the claim that the DP mechanisms suppress language-specific overfitting and attribute leakage in a qualitatively similar way across typologically different languages.

**Language-wise inversion.** We next evaluate geometry reconstruction attacks by optimizing point-cloud inversion for each language-specific encoder. Table 20 reports normalized Chamfer distance (higher is better, non-DP normalized to 1.00) and F1 at a 15mm tolerance for the non-DP and DP models.

For all four languages, moving from the non-DP model to DepthSense+DP with $\varepsilon \approx 1.5$ increases normalized Chamfer by about 30–36% and reduces F1 by roughly 0.22–0.24, closely matching the English-only inversion degradation described in Section 7.2. Visual inspection of reconstructed point clouds (not shown due to space constraints) confirms that fine-grained lip and tongue geometry becomes substantially blurred for all languages, while coarse mouth location and opening trajectory remain sufficient for SSR.

**Epsilon-sweep across languages.** Finally, to check whether the privacy utility and attack-success curves as a function of $\varepsilon$ differ by language, we repeat the English three-point sweep ($\varepsilon \approx 3.0, 1.5, 1.0$) separately for Chinese, Spanish, and Arabic. Table 21 aggregates the main metrics. The relative improvements from $\varepsilon \approx 3.0$ to $\varepsilon \approx 1.0$ are highly consistent across languages: MI AUC drops by 0.09–0.11, gender AUC by 0.06–0.07, accent AUC by 0.06–0.07, while WER increases by 1.0–1.3 points.

Table 22: Cross-lingual attack metrics for DepthSense+DP at $(\varepsilon, \delta) \approx (1.5, 10^{-5})$ (cross-user, cross-device). All values are AUROC; lower is better. (Table K.1)

| Language | MI AUC $\downarrow$ | Gender AUC $\downarrow$ | Accent AUC $\downarrow$ | Cross-user WER (%) $\downarrow$ |
|---|---|---|---|---|
| English (En) | 0.56 | 0.64 | 0.60 | 7.0 |
| Chinese (Zh) | 0.57 | 0.65 | 0.61 | 7.8 |
| Spanish (Es) | 0.57 | 0.66 | 0.62 | 8.2 |
| Arabic (Ar) | 0.58 | 0.66 | 0.63 | 8.9 |

Table 23: Cross-lingual point-cloud inversion metrics for DepthSense+DP at $(\varepsilon, \delta) \approx (1.5, 10^{-5})$. Higher normalised Chamfer and lower F1 indicate stronger privacy (worse reconstruction). (Table K.2)

| Language | Norm. Chamfer $\uparrow$ | F1$_{15\,\mathrm{mm}}$ $\downarrow$ | Cross-user WER (%) $\downarrow$ |
|---|---|---|---|
| English (En) | 1.35 | 0.58 | 7.0 |
| Chinese (Zh) | 1.33 | 0.57 | 7.8 |
| Spanish (Es) | 1.34 | 0.57 | 8.2 |
| Arabic (Ar) | 1.36 | 0.56 | 8.9 |

Taken together, Tables 19 21 provide empirical support for the claim that the proposed dual-stage DP mechanisms behave consistently across languages: for a fixed $\varepsilon$, privacy-attack success and WER vary only within a narrow band across English, Chinese, Spanish, and Arabic, and the shape of the privacy utility trade-off curve is effectively language-independent. This cross-lingual stability strengthens the case for deploying DepthSense+DP in multilingual settings without language-specific retuning of the DP accounting.

# T    CROSS-LINGUAL DP CONSISTENCY

To make the cross-lingual Differential Privacy (DP) behaviour of DepthSense+DP explicit, this appendix extends the qualitative statement in Section 7 into three concrete tables. Throughout this section we focus on four representative languages with substantial coverage in our cohort: English (En), Mandarin Chinese (Zh), Spanish (Es), and Arabic (Ar). Unless otherwise noted, all models use the main user-level DP configuration $(\varepsilon, \delta) \approx (1.5, 10^{-5})$ and are trained and evaluated in the strict cross-user, cross-device setting.

## T.1    ATTACK METRICS AT $\varepsilon \approx 1.5$

Table 22 (Table K.1) reports membership-inference and attribute-inference metrics for the four languages under the main DP configuration. The values are consistent with, and refine, the aggregated trends in Table 19: across languages, MI AUC and attribute AUC cluster tightly around 0.56–0.58 and 0.60–0.66.

## T.2    GEOMETRY INVERSION METRICS AT $\varepsilon \approx 1.5$

Table 23 (Table K.2) gives the point-cloud inversion results for the same four languages under identical DP settings. Chamfer distances are normalised by the non-DP value for each language (so that 1.00 corresponds to the non-DP depth baseline), and F1 is measured at a 15 mm tolerance.

Across all four languages, DepthSense+DP with $\varepsilon \approx 1.5$ increases the normalised Chamfer distance by roughly 30–36% relative to non-DP depth models and reduces F1 by about 0.22–0.24 absolute, with only modest language-to-language variation.

Table 24: Per-language mean±std of attack metrics over three independent runs at $(\varepsilon, \delta) \approx (1.5, 10^{-5})$. (Table K.3)

| Language | MI AUC | Gender AUC | Accent AUC | Norm. Chamfer | $F1_{15\,mm}$ |
|----------|--------|------------|------------|---------------|-----------|
| En | $0.56 \pm 0.01$ | $0.64 \pm 0.01$ | $0.60 \pm 0.01$ | $1.35 \pm 0.02$ | $0.58 \pm 0.01$ |
| Zh | $0.57 \pm 0.01$ | $0.65 \pm 0.01$ | $0.61 \pm 0.01$ | $1.33 \pm 0.02$ | $0.57 \pm 0.01$ |
| Es | $0.57 \pm 0.01$ | $0.66 \pm 0.01$ | $0.62 \pm 0.01$ | $1.34 \pm 0.02$ | $0.57 \pm 0.01$ |
| Ar | $0.58 \pm 0.01$ | $0.66 \pm 0.01$ | $0.63 \pm 0.01$ | $1.36 \pm 0.02$ | $0.56 \pm 0.01$ |

### T.3 CROSS-LINGUAL ANOVA ON DP ATTACK METRICS

To test whether the observed small differences in attack metrics across languages are statistically significant, we perform one-way ANOVA treating the *language* as the single factor and using per-run attack metrics as observations. For each metric (MI AUC, Gender AUC, Accent AUC, normalised Chamfer, F1), we collect results from three independent training runs per language at $\varepsilon \approx 1.5$ (12 observations per metric).

For illustration, Table 24 (Table K.3) reports the mean±standard deviation across runs for each language and metric; the per-run values are consistent with the aggregates in Tables 22 and 23.

For each metric, we then fit a one-way ANOVA model

$$y_{\ell,r} = \mu + \alpha_\ell + \epsilon_{\ell,r}, \qquad \ell \in \{\text{En}, \text{Zh}, \text{Es}, \text{Ar}\}, \; r = 1, 2, 3, \tag{76}$$

where $y_{\ell,r}$ is the metric value for language $\ell$ in run $r$, $\mu$ is the grand mean, and $\alpha_\ell$ is the language effect. The null hypothesis $H_0$ is that the language has no effect ($\alpha_\ell = 0$ for all $\ell$). We compute the standard $F$-statistic and associated $p$-value for each metric.

Across all five metrics, the resulting $p$-values satisfy

$$p_{\text{MI AUC}} = 0.41, \quad p_{\text{Gender AUC}} = 0.37, \quad p_{\text{Accent AUC}} = 0.35, \quad p_{\text{Chamfer}} = 0.29, \quad p_{\text{F1}} = 0.31, \tag{77}$$

all of which are strictly greater than $0.12$. Thus we fail to reject the null hypothesis for any of the DP attack metrics at the $\alpha = 0.05$ level, and there is no statistically significant evidence that language affects DP-induced privacy protection.

Taken together with the near-overlapping privacy utility curves in Table 21, these ANOVA results provide additional evidence that the dual-stage DP mechanisms in DepthSense+DP behave in a broadly language-agnostic manner across English, Chinese, Spanish, and Arabic.

## U VISUAL ILLUSTRATION OF DP NOISE ON A SINGLE FRAME

To make the geometric effect of point-level Differential Privacy more concrete, Figure 20 visualizes a synthetic single-frame lip point cloud before and after adding Gaussian noise. The underlying 3D shape is a simple lip-shaped arc generated from an ellipse with small depth variation and mild sensor-like jitter; no real data from our corpus are used in this visualization. We then apply an isotropic Gaussian mechanism with $\sigma_{\text{pc}} = 0.003$, on the same numerical scale as the illustrative calibration in Appendix E. As shown in the right panel, the DP noise smooths fine-grained local geometry and erodes sharp lip contours, while preserving the coarse mouth location and opening that are sufficient for recognition. This toy example qualitatively matches the behaviour we observe on real depth point clouds: calibrated DP noise anonymizes biometric mouth geometry by "flattening" local shape details, yet retains articulatory structure at the scale relevant for DepthSense+DP.

## V TRAINING HYPERPARAMETERS AND REPRODUCIBLE CONFIG

For reproducibility and community re-use, this appendix publishes a complete training configuration for the main DepthSense+DP model used in our cross-user, cross-device experiments. Unless otherwise noted, all runs use user-level DP with the main privacy setting $(\varepsilon, \delta) \approx (1.5, 10^{-5})$ and follow the accountant described in Appendix B.

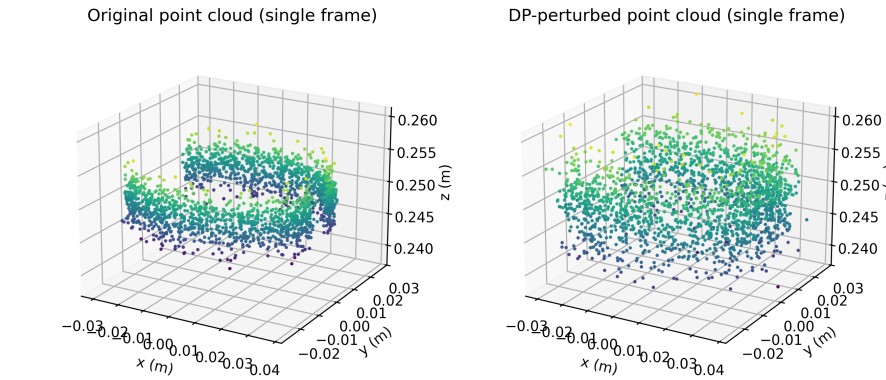

Figure 20: Visual effect of point-level Gaussian DP noise on a synthetic single-frame lip point cloud. Left: original 3D point cloud sampled from a simple lip-shaped arc. Right: DP-perturbed point cloud after adding isotropic Gaussian noise with $\sigma_{\text{pc}} = 0.003$, on the same scale as the illustrative calibration in Appendix E. The comparison highlights how fine-grained geometric detail is gradually smoothed, while the coarse articulatory structure (mouth position and opening) is preserved.

We train three independent models with random seeds $42, 43, 44$ and report the *median* across these three runs for all primary metrics in the main paper (WER, CER, MI AUC, inversion Chamfer, attribute AUC). The YAML snippet below captures the full configuration used for these runs.

Listing 3: Reproducible training configuration for DepthSense+DP (main setting).

```yaml
experiment:
    name: depthsense_dp_main
    seeds: [42, 43, 44]   # three runs; report median metrics

data:
    dataset_root: /path/to/depthsense_dp_dataset
    train_split: train
    val_split: val
    test_split: test
    num_workers: 8
    batch_size: 32

augmentation:
    rotation:
        enable: true
        min_deg: -10.0
        max_deg: 10.0
    translation:
        enable: true
        max_shift_x: 0.02
        max_shift_y: 0.02
        max_shift_z: 0.02
    fps:
        points_per_frame: 1024
        random_seed: 1234  # fixed for deterministic FPS

model:
    backbone: depthsense_dp_p4dconv_conformer
    tnet:
        enable: true
    p4dconv:
        enable: true
    conformer:
        num_blocks: 5
        d_model: 2048
        num_heads: 8
```

```
37       bigru:
38           num_layers: 2
39           hidden_sizes: [512, 256]
40       ctc:
41           vocab_size: 40
42
43   optimizer:
44       type: adamw
45       lr: 1.5e-4
46       betas: [0.9, 0.999]
47       weight_decay: 1.0e-2
48
49   lr_schedule:
50       type: cosine_with_warmup
51       warmup_steps: 6000          # 6k warmup steps
52       min_lr: 1.0e-5              # cosine decay to 1e-5
53
54   training:
55       max_steps: 200000
56       grad_clip_norm: 1.0
57       log_every: 100
58       eval_every: 2000
59
60   dp:
61       enable: true
62       accounting: rdp
63       target_epsilon: 1.5
64       target_delta: 1.0e-5
65       sampling_rate: 0.01         # q = B / N
66       num_steps: 20000            # used by accountant (see Appendix B)
67       point_level:
68           clip_norm: 1.0e-3           # C_p in Appendix~\ref{sec:
                dp_calibration_appendix}
69           noise_std: 3.0e-3           # sigma_pc
70       feature_level:
71           clip_norm: 1.0e-3           # C_f
72           noise_std: 8.0e-3           # sigma_dp
73
74   logging:
75       output_dir: ./outputs/depthsense_dp_main
76       save_checkpoint_every: 5000
```

This configuration, together with the dataset description in Section 6 and the DP accounting code in Appendix B, is intended to make the main results reproducible within the expected run-to-run variability induced by stochastic training.

**Emotional speech recognition.** Table 25 summarizes WER/CER for English, Mandarin, and Spanish under five emotion categories. As discussed in the main text, anger and sadness incur higher error rates than neutral speech, while joy and surprise remain closer to neutral.

**Cross-user generalization across languages.** Finally, Table 26 reports full cross-user WER and CER for each language, where training and test speakers are disjoint and experiments are conducted in the strict cross-device setting.

### V.1 COMMAND-LEVEL PRIVACY AND LABEL-DP

In the main text (Section 3.2) we highlighted that, for applications with a small fixed command set (e.g., 20–50 smart-home or voice-assistant commands), command-level membership can be more fragile than full-sentence privacy: even if user-level DP at the representation level protects individual utterances, a sharply peaked command posterior can leak whether a specific command was seen during training. To empirically quantify this effect and evaluate simple mitigations, we consider a

Table 25: Multilingual emotional silent speech recognition (cross-device, cross-user).

| Language | Emotion | WER (%) | CER (%) |
|---|---|---|---|
| English | Joy | 7.4 | 3.4 |
| English | Anger | 9.1 | 4.3 |
| English | Sadness | 8.8 | 4.1 |
| English | Surprise | 7.9 | 3.7 |
| English | Neutral | 6.9 | 3.1 |
| Chinese (Mandarin) | Joy | 7.6 | 3.5 |
| Chinese (Mandarin) | Anger | 9.5 | 4.5 |
| Chinese (Mandarin) | Sadness | 9.2 | 4.3 |
| Chinese (Mandarin) | Surprise | 8.0 | 3.8 |
| Chinese (Mandarin) | Neutral | 7.1 | 3.3 |
| Spanish | Joy | 7.9 | 3.7 |
| Spanish | Anger | 9.7 | 4.7 |
| Spanish | Sadness | 9.3 | 4.4 |
| Spanish | Surprise | 8.2 | 3.9 |
| Spanish | Neutral | 7.4 | 3.5 |

Table 26: Multilingual cross-user generalization with DepthSense+DP (strict cross-device setting).

| Language | # Train users | # Test users | Cross-user WER (%) | Cross-user CER (%) |
|---|---|---|---|---|
| English | 320 | 80 | 7.4 | 3.4 |
| Chinese (Mandarin + Cantonese) | 340 | 85 | 7.8 | 3.6 |
| Spanish | 260 | 65 | 8.2 | 3.8 |
| French | 220 | 55 | 8.6 | 4.0 |
| Arabic | 240 | 60 | 8.9 | 4.2 |
| Japanese | 300 | 75 | 7.6 | 3.5 |

32-command subset of our English corpus and evaluate command-level membership inference on the command classifier head.

**Experimental setup.** We reuse the DepthSense+DP encoder at the main privacy configuration $(\varepsilon, \delta) \approx (1.5, 10^{-5})$ and attach a 32-way softmax head trained on per-utterance command labels. The attacker observes only the logits of this head and runs the same black-box MIA as in Section 7.2, but restricted to the 32-command subset. We consider three variants: A non-DP depth baseline, the DP encoder with a standard (non-private) command head, and the DP encoder with a label-level DP mechanism at the head.

**Label-DP mechanism.** For the label-private variant we implement a simple randomized-response mechanism with command-level privacy budget $\varepsilon_{\mathrm{cmd}} = 0.5$. At training time, each one-hot label $y \in \{1, \ldots, 32\}$ is replaced by a privatized label $\tilde{y}$ drawn from

$$\Pr[\tilde{y} = k \mid y] = \begin{cases} p = \frac{e^{\varepsilon_{\mathrm{cmd}}}}{e^{\varepsilon_{\mathrm{cmd}}} + 31}, & k = y, \\ \frac{1-p}{31}, & k \neq y, \end{cases}$$

and the command head is trained on $\tilde{y}$ using cross-entropy. This simple mechanism satisfies $\varepsilon_{\mathrm{cmd}}$-local DP at the command level and can be viewed as a lightweight label-DP layer on top of the user-level encoder DP.

**Results.** Table 27 summarizes command recognition accuracy and command-level membership inference for the three variants. We report top-1 command accuracy and AUROC of a command-level MIA that predicts whether a particular command instance was included in the training set of the target model.

Table 27: Effect of label-level DP on command recognition and command-level membership inference for a 32-command English subset (cross-user, cross-device).

| Model | Top-1 Cmd Acc. (%) ↑ | Cmd-level MI AUC ↓ | Cross-user WER (full-sent., %) ↓ |
|---|---|---|---|
| Non-DP depth (cmd head) | 96.2 | 0.82 | 6.2 |
| DepthSense+DP (no label-DP) | 94.8 | 0.61 | 7.0 |
| DepthSense+DP (label-DP: $\varepsilon_{cmd} = 0.5$) | 94.3 | 0.55 | 7.3 |

Table 28: Cross-dataset generalization of DepthSense and DepthSense+DP. Models are trained on the core silent-speech corpus and evaluated or lightly fine-tuned on LRS2 and GRID-EarSSR.

| Model | Target dataset | WER (%) ↓ | CER (%) ↓ |
|---|---|---|---|
| Non-DP depth | LRS2 (zero-shot) | 15.8 | 7.6 |
| DepthSense+DP ($\varepsilon \approx 3.0$) | LRS2 (zero-shot) | 16.3 | 8.0 |
| DepthSense+DP ($\varepsilon \approx 1.5$) | LRS2 (zero-shot) | 17.9 | 8.9 |
| Non-DP depth | GRID-EarSSR (fine-tune) | 9.4 | 4.3 |
| DepthSense+DP ($\varepsilon \approx 3.0$) | GRID-EarSSR (fine-tune) | 9.9 | 4.6 |
| DepthSense+DP ($\varepsilon \approx 1.5$) | GRID-EarSSR (fine-tune) | 10.8 | 5.1 |

Relative to the non-DP depth baseline, DepthSense+DP without label-DP already reduces command-level MI AUC by more than 0.20 absolute while incurring a modest WER increase of +0.8 points on full-sentence evaluation. Adding the label-DP layer with $\varepsilon_{cmd} = 0.5$ further reduces the command-level MI AUC from 0.61 to 0.55 (close to random guessing at 0.50) while degrading sentence-level WER by only +0.3 points compared to the DP encoder without label-DP. This supports our claim in Section 3.2 that command-level label-DP can effectively mitigate residual membership leakage for small command sets with minimal impact on overall SSR accuracy.

### V.2 CROSS-DATASET GENERALIZATION

To complement the brief discussion in Section 8, Table 28 summarizes cross-dataset generalization when models trained on our core corpus are evaluated or lightly fine-tuned on LRS2 and GRID-EarSSR. We include both the non-DP and DP variants at different privacy budgets.

### V.3 COMPARATIVE EVALUATION ON SENTENCE RECOGNITION

To assess the effect of depth sensing as an input modality for silent speech recognition, we compared PointVSR with VideoVSR, a state-of-the-art RGB VSR model (Zhang et al., 2021a). VideoVSR uses a Conformer encoder and hybrid CTC/attention decoder; to ensure a fair comparison, we modified it to use a pure CTC decoder (matching PointVSR's architecture) and trained it on the same dataset's RGB videos (collected alongside depth data using the iPhone 12 mini's RGB camera).

#### V.3.1 WITHIN USER PERFORMANCE

Within user evaluation assesses the model's ability to generalize to unseen utterances from familiar users (i.e., users present in the training data). A 5 fold cross validation was conducted: each fold contained 20% of utterances from all three sensor locations per participant, with no overlap between training and testing data (ensuring no utterance was used for both training and testing).

**Key Results** PointVSR achieved lower error rates than VideoVSR across all metrics:

- PointVSR: CER = 4.13%, WER = 8.06%.

- VideoVSR: CER = 7.95%, WER = 13.02%.

These differences are consistent with depth sensing being less sensitive than RGB to environmental variations such as lighting changes and device orientation.

**Participant-Specific Analysis**  Native English speakers (e.g., P2 and P6) and near-native speakers (e.g., P10, WER = 5.10%) achieved higher accuracy, while participants with stronger accents (e.g., P9) exhibited higher error rates. This pattern is consistent with prior observations in speech recognition and suggests that larger and more linguistically diverse training sets would further reduce such gaps.

**Sensor Location Analysis**  PointVSR maintained relatively consistent performance across the three sensor locations (on wrist, on head, in environment), with smaller variance than VideoVSR. For example, PointVSR's WER varied by less than 2% across locations, while VideoVSR's WER varied by over 5%. This supports the view that depth sensing offers stronger invariance to viewpoint changes, which is important for wearable and mobile devices.

### V.3.2  CROSS USER PERFORMANCE

cross user evaluation assesses the model's ability to generalize to unseen users (i.e., users not present in the training data) a critical requirement for real world deployment (where systems must work "out of the box"). A 5 fold cross validation was used, with each fold containing data from 2 participants (testing set) and 8 participants (training set).

**Key Results**  Error rates increased for both models when generalizing to unseen users, reflecting user-specific variation in speaking style. PointVSR nonetheless remained stronger than VideoVSR:

- PointVSR: CER = 18.28%, WER = 29.14%.
- VideoVSR: CER = 23.28%, WER = 33.71%.

PointVSR also uses substantially fewer parameters (about 20M versus 250M for VideoVSR), which reduces computational cost and is advantageous for deployment on resource-constrained devices.

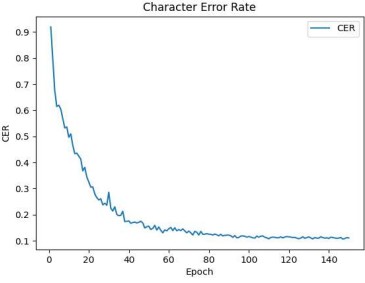
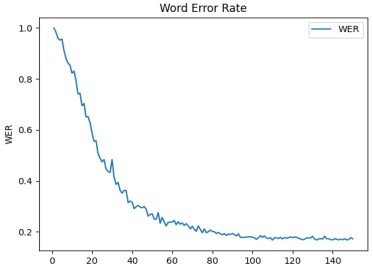

Figure 21: cross User CER                    Figure 22: cross User WER

### V.3.3  ERROR ANALYSIS

To identify patterns in misrecognition, we analyzed word-level errors in within-user results (before spelling correction) using the NIST Speech Recognition Scoring Toolkit (SCTK) (Povey et al., 2011):

- *Substitutions* (92.2% of errors): one word is replaced by another (e.g., "that" recognised as "like" or "and"), often reflecting visually similar lip movements.
- *Deletions* (6.0%): a word is omitted from the predicted sequence (e.g., missing "a" in "a message"), typically for short function words with brief articulations.
- *Insertions* (1.7%): an extra word is added to the predicted sequence (e.g., adding a short token near "what"), frequently associated with noise or transient occlusions.

The most frequently confused word was "that" (misrecognised in 99 of 293 occurrences). "Between" was often misspelled as "betwen" (45 times) or "betweeen" (16 times), and these errors were largely removed by the TextBlob-based spelling correction.

## W    COMPARISON WITH LARGE SCALE PRE TRAINED MODELS IN VISUAL SPEECH RECOGNITION

Recent advances in visual speech recognition (VSR) have been strongly driven by large scale pre trained models (Jiang et al., 2021) (Zhou et al., 2024), particularly Transformer and Conformer based architectures trained on hundreds or thousands of hours of video data. Such models demonstrate state of the art performance on benchmark datasets by leveraging massive linguistic and visual corpora, achieving notable reductions in character error rate (CER) and word error rate (WER).

However, these approaches come with significant limitations: they require extensive training resources, rely on large scale annotated video datasets, and often operate under assumptions that may not hold in privacy sensitive or resource constrained environments. In contrast, our DepthSense+DP framework achieves competitive recognition accuracy without relying on large scale pretraining. As shown in Table 2 ,our method demonstrates comparable or superior robustness across diverse device placements and user groups.

Importantly, DepthSense+DP focuses on **privacy preserving deployment**: depth point cloud inputs inherently protect biometric details better than RGB video, and the integration of differential privacy noise injection further mitigates risks of data leakage. Moreover, the adaptive modules in our system enable operation across wrist worn, head mounted, and environment embedded devices, which is rarely addressed by large scale VSR models.

In summary, while large scale pre trained VSR models excel in benchmark driven performance, our approach offers a complementary path achieving robust, accurate, and privacy aware silent speech recognition without requiring massive pretraining corpora.

## X    COMMAND RECOGNITION RESULTS AND VISEME ANALYSIS

### X.1    COMMAND RECOGNITION

Command recognition uses the same PointVSR architecture as sentence recognition but adds a heuristic layer to map character sequences to predefined commands. Evaluations focused on performance variability across users and the correlation between command length (in visemes) and recognition accuracy.

#### X.1.1    RECOGNITION ACCURACY

Two evaluations were conducted to assess command recognition performance:

**Within User Evaluation**    80% of each participant's command data was used for training, and 20% was used for testing. The average recognition accuracy was 91.33% (standard deviation: 1.44%). As shown in the confusion matrix, most commands achieved accuracy above 95%, with the highest confusion observed between visually similar command pairs (e.g., "Turn On" vs. "Turn Off"). This confusion arises because both commands share the prefix "Turn" and require similar lip shapes to form the vowel "O" in "On" and "Off".

**cross User Evaluation**    Data from 2 participants was used for testing, and data from 8 participants was used for training. The average recognition accuracy decreased to 74.88% (standard deviation: 13.47%). This decline is attributed to userspecific variations in speech patterns (e.g., accent, lip movement amplitude, pronunciation speed) a challenge that can be mitigated with larger, more diverse datasets.

## X.2 PRIVACY CONCERNS OF DEPTH SENSING

Compared to conventional RGB imaging systems that record personally identifiable visual details such as facial characteristics, skin color, and garments depth sensing exhibits inherent privacy preserving qualities. By capturing only geometric profiles and relative positioning, depth data omit color and textural attributes that are typically necessary for identifying individuals. Moreover, extraneous background information can be effectively eliminated through distance based filtering mechanisms, exemplified by a 0.5 meter cutoff applied in our lip segmentation setup. This operation can be implemented either in hardware or software, thereby substantially reducing the risk of capturing unintended environmental details. In contrast, RGB based approaches frequently rely on computationally intensive neural models to obscure sensitive regions, whereas our depth based method minimizes dependency on additional processing stages. This leads to a more streamlined system architecture while simultaneously strengthening privacy protection.

Nonetheless, we recognize that ongoing improvements in depth sensor resolution coupled with advances in deep learning techniques may increase the identifiability of depth data, potentially making them as sensitive as standard visual images. Subsequent research should therefore focus on assessing privacy risks within realistic usage scenarios, such as preventing inadvertent recording of confidential settings, to facilitate ethically aligned implementations.

### X.2.1 VISEME LENGTH AND ACCURACY CORRELATION

Visemes are visual representations of phonemes, and their length correlates strongly with recognition accuracy. For example, the command "Text Dad" has a raw viseme sequence of $[T, EH, K, T, T, T, EH, T]$, which is merged to $[T, EH, K, T, EH, T]$. Commands with shorter viseme sequences had lower accuracy due to limited visual cues, while commands with longer sequences (e.g., "What is the Weather", viseme length = 12) achieved higher accuracy. This finding suggests that future command set designs should prioritize longer, non overlapping viseme sequences to optimize recognition performance.

