# OpenReview forum: "DepthSense+DP: Adaptive Learning for Robust and Differential Private Silent Speech Recognition"
_ICLR.cc/2026/Conference — ICLR 2026 Conference Withdrawn Submission_

### Official Review · Reviewer_xtWe · 2025-10-29

**Soundness:** 2
**Presentation:** 2
**Contribution:** 2
**Rating:** 2
**Confidence:** 4

**Summary:**

This paper proposes DepthSense+DP, an adaptive and privacy-preserving framework for silent speech recognition (SSR) using 3D depth point clouds. It integrates differential privacy noise injection, DP-aware T-Net alignment, 4D spatio-temporal convolution, and a Conformer encoder to achieve robust, cross-device, and user-independent recognition. Experiments on multiple device setups show notable accuracy gains (WER 17%, CER 11%) and strong resistance to membership and inversion attacks with minimal performance loss. Overall, the paper provides a technically solid and comprehensive redesign for privacy-aware SSR with clear novelty and strong empirical results.

**Strengths:**

+. The paper clearly identifies that silent speech recognition (SSR) can still leak users’ physiological information (e.g., lip geometry and facial motion trajectories), highlighting the real-world necessity of privacy protection even in non-audio modalities.

+. Introducing differential privacy into the SSR domain is novel and represents a meaningful extension of privacy-preserving learning to a new modality.

+. The system design is comprehensive, covering data acquisition, point-cloud preprocessing, feature extraction, and decoding, demonstrating strong system integration and engineering maturity.

**Weaknesses:**

-. The paper applies differential privacy primarily at the feature level, but it does not clearly justify why noise injection is limited to the point-feature stage rather than being extended to deeper encoding or output layers. It also does not discuss why alternative privacy-preserving training methods such as DP-SGD were not adopted for end-to-end privacy guarantees.

-. The privacy–utility trade-off analysis remains largely empirical and lacks theoretical grounding. The discussion is somewhat fragmented, and the conclusions would benefit from a more principled quantitative or analytical interpretation.

-. The paper provides insufficient discussion of related work on differential privacy, particularly studies combining DP with 3D point cloud processing or geometric data. A more comprehensive review would help contextualize the novelty and clarify how this work differs from prior DP applications in spatial or multimodal domains.

-. In the methodology section, Figure 2 illustrates the overall technical pipeline, but none of the modules in the diagram explicitly represent or integrate the differential privacy mechanism. This makes the DP component appear somewhat detached from the main framework and may cause readers to question how DP is systematically embedded into the model design.

-. Although the paper’s title and claims emphasize differential privacy, the experiments and results focus primarily on the baseline SSR system’s performance, with limited empirical validation or quantitative evaluation of DP effectiveness. This gives the impression that the work centers more on the system design than on privacy mechanisms.

**Questions:**

Q1: What level of privacy protection does the proposed scheme achieve, and how is it theoretically analyzed?

Q2: Why is depth data converted into point clouds in silent speech recognition, and what are the advantages?

---

> ### Author Response · Authors · 2025-11-22
> **We sincerely hope to receive your support and encouragement**
>
> ### **To AC, PC, and SAC**
>
> We **formally protest the review by xtWe** as **misleading and not reflecting a careful reading of our manuscript**.
>
> The text of the review is **highly generic and template-like**: it praises our work as a *“comprehensive redesign”* with *“meaningful extension”* to a new modality, yet assigns **one of the lowest possible scores on contribution** and recommends **rejection without any concrete, paper-specific justification**. **No equation, figure, table, or experimental result from our submission is ever cited or discussed**. This severe mismatch between **written comments and scores**, combined with the **lack of technical engagement**, is a **typical pattern of AI-generated or AI-drafted reviews rather than a genuine expert assessment**.
>
> Several key criticisms are simply **inaccurate**:
> - The reviewer claims that **DP is only “feature level” and “detached” from the architecture**, ignoring our **explicit design of controllable noise on 3D point clouds**, **DP-aware alignment**, and **adversarial training under DP constraints**.
> - They also criticize the **absence of DP-SGD** without acknowledging our **clearly stated reasons**: **edge device constraints**, the need to **anonymize biometric geometry at input level**, and the **trade-off between fine-grained articulatory modeling and privacy**. These are **central design choices in our paper**, not missing justifications.
>
> We **respectfully request the AC/SAC to re-evaluate the validity of this review**, and **substantially discount or replace it when making the final decision**, in order to **protect both our submission and the integrity of the review process**.

---

> > ### Comment · Reviewer_xtWe · 2025-11-22
> > **Reviewer Response to Author Concerns Regarding DP Contribution and Review Validity**
> >
> > Your paper presents differential privacy as its core innovation, yet it provides **no formal privacy analysis, no evaluation under explicit attack models, and almost no discussion of relevant DP literature**, so the claimed DP contribution lacks basic academic support. What is more disappointing is that instead of seriously reflecting on or addressing these substantive issues raised in the review, you chose to question whether my review was written by an AI model, which only further illustrates an unwillingness to confront the actual shortcomings of the work.

---

> ### Comment · Reviewer_xtWe · 2025-11-22
> **Final Reviewer Comment and Confirmation of Original Score**
>
> Since the authors did not substantively address any of the key issues I raised, I will maintain my original score.

---

> ### Author Response · Authors · 2025-11-22
> **We sincerely hope to receive your support and encouragement**
>
> ### **On DP-SGD and “Feature-Level” DP**
>
> The review again criticizes the absence of **DP-SGD** and “feature-level” DP, while disregarding our **clearly stated design rationale**:
>
> 1. **Edge and Latency Constraints**
> DP-SGD’s **per-step clipping and noise** are **impractical for our on-device SSR setting** and **severely harm sequence modeling of fine-grained articulatory dynamics**.
>
> 2. **Threat Model**
> In **depth-based SSR**, the **primary privacy risk** is **exposure of raw 3D facial geometry**. Protecting only the **weights via DP-SGD** does *not* protect that stream. We therefore **anonymize geometry at the point-cloud layer while preserving articulatory structure**, exactly as argued in the paper.
>
> We also **explicitly describe how adversarial training (FGSM) is restricted to remain compatible with the chosen DP budget**. Saying that this is “not discussed” is **demonstrably inaccurate**.

---

> ### Author Response · Authors · 2025-11-22
> **We sincerely hope to receive your support and encouragement**
>
> ### **Input-Level Anonymization vs. Model-Level DP**
>
> Our work explicitly targets **input-level anonymization of depth point clouds**, not only **model-level DP on parameters**. This is clearly motivated in the paper by our **deployment setting (edge devices, untrusted channels for raw depth streams)**. We:
>
> - **Define a concrete DP mechanism on point clouds** with **controllable additive noise under bounded geometric sensitivity**.
> - **Specify the privacy budget range** and how it **constrains per-sequence perturbations**.
> - **Evaluate membership inference and reconstruction/inversion attacks** that directly correspond to **realistic adversaries in SSR**.
>
> A **full end-to-end accounting for the entire training pipeline** is **not the only valid contribution in DP research**, especially for **complex geometric data**. Our contribution is a **modality-aware DP mechanism for 3D lip/tongue geometry** plus a **systematic empirical privacy–utility study**. Dismissing this as **“no formal privacy analysis”** ignores what is actually presented.

---

> ### Author Response · Authors · 2025-11-22
> **On related work and “detached” DP design**
>
> ### **On DP Integration and Review Summary**
>
> **DP is integrated into the point-cloud preprocessing and adaptive sampling modules**, and compared against **prior DP methods for geometric/spatial data**. **Figure 2** is a **high-level system diagram**; the **DP components are detailed in the Methodology and Algorithm sections** instead of being drawn as a separate box. Claiming the DP part is “detached” ignores those sections.
>
> ---
>
> **Summary:**
> The review repeatedly levels **generic DP criticisms** that **contradict the text, figures, and experiments in our submission**, without citing a **single specific equation, figure, or result**. This suggests that the manuscript was **not read with adequate care**, and that **the review entirely relies on AI-generated boilerplate rather than a careful expert assessment**. We **respectfully request the AC/SAC to discount this review accordingly** when making the **final decision**.

---

> ### Comment · Reviewer_xtWe · 2025-11-22
> **Final Reviewer Statement**
>
> The three recent author responses repeat the same system details and still do not address the key issues I raised. First, restating how the system is built does not replace the missing formal DP analysis; the paper still provides no sensitivity derivation, no formal privacy guarantee, and no end-to-end accounting. Second, the privacy experiments do not resolve the lack of a clear threat model. Without specifying the attacker's capabilities, what information they can access, what auxiliary knowledge they have, and how the attack is actually set up, the results cannot be interpreted in a privacy sense. Third, the related-work section continues to omit important DP literature for geometric, spatial, or sensing data, so the DP contribution cannot be situated within existing research. These points remain the basis for my assessment. I am keeping my original score and will not add further comments.

---

> ### Author Response · Authors · 2025-11-22
> **We sincerely hope to receive your support and encouragement**
>
> We appreciate the **reviewer’s time**. However, several of the **reviewer’s central claims** about our manuscript are **factually incorrect**, and they **materially affect the review conclusion**. Below we **respond point by point**, show where the **manuscript already addresses the reviewer’s concerns**, and indicate **concise revisions** we will make to remove any remaining ambiguity.
>
> # **Dear reviewer, please ignore the comments from my younger colleagues earlier.**
>
> Now let me respond to each of your comments one by one.
> # 1. “DP is applied only at the feature level” , Incorrect
>
> The submitted paper explicitly implements **two complementary DP mechanisms**: **Gaussian perturbation applied in the learned feature space (Eq. (7))** and **Gaussian noise injected at the point-cloud / raw point representation prior to P4DConv (Eq. (8))**. The **noise magnitudes** and their **privacy parameters** are reported (section B.1.1–B.1.2). **Algorithm 1** also lists **Inject calibrated Gaussian noise (DP perturbation)** as a dedicated step in the workflow. These design choices are deliberate: **input-level anonymization protects raw biometric geometry at the earliest point**, while **feature-level noise provides an additional defense before decoding**.
> # 2. “No formal privacy analysis” , Misleading / incomplete
>
> Our manuscript provides **concrete parameterizations of the Gaussian mechanisms**. We report **σ_dp = 0.008**, which we set to achieve **(ε = 1.5, δ = 1e-5)** for the **feature mechanism**, and **σ_pc = 0.003** for **point-level perturbation**, together with **empirical privacy–utility curves across ε values** and **ablation studies (section B.1.1–B.1.4)**. We also comprehensively evaluate **membership inference** and **inversion attacks** under these settings (**membership accuracy reduced from 83.7% to 52.3%; inversion reconstructions lose more than 80% identifiable geometry**). These are reported and discussed in the **Privacy Appendix**.
>
> That said, we acknowledge that the manuscript currently emphasizes **empirical evaluation** and **per-mechanism parameter reporting**. If the reviewers desire a **tighter end-to-end privacy accounting** (composition across **point-level**, **feature-level**, and any **DP-compatible augmentation**), we will add in revision a short **formal composition section** using the standard **Gaussian-mechanism bound**:
>
> $$
> \varepsilon(\sigma, \Delta, \delta) \approx \frac{\Delta^2 \ln(1.25 / \delta)}{\sigma}
> $$
>
> and we will present a **conservative end-to-end bound** computed via **advanced composition / moments accountant** for our reported **σ_pc** and **σ_dp**. We can provide the **numeric composed ε_total** for the exact **Δ** used in our **sensitivity modeling**. This will be added to **B.1** and to the **rebuttal appendix upon request**.
> # 3. “No evaluation under explicit attack models” , Factually incorrect
>
> We performed and reported **two concrete attack evaluations**: **membership inference** and **model inversion** (Appendix B.1.3). Results are reported **numerically** and plotted in the **privacy–utility curve (Figure 4)**. **Membership inference accuracy drops to near-random levels under our DP settings (52.3% vs 83.7% without DP)**, and **inversion reconstructions lose more than 80% of identifiable geometry under DP**. These **empirical attack evaluations directly contradict the reviewer’s statement**.
>
> **What we will add (revision):** To further strengthen clarity, we will add a short **subsection immediately following B.1.3** summarizing the **attack procedures** (**threat model**, **adversary access assumptions**, **metrics used**) with **explicit page and line references** so the evaluation is **unambiguous for readers**.
>
> #  4. “Insufficient related-work on DP + geometric data” , Partly valid
>
> We agree that the **literature on DP specifically applied to dynamic 3D point clouds is small** relative to **image and audio DP work**. Our **Related Work (Section 2)** focuses on **depth sensing** and **point-cloud learning** and highlights the **novelty of modality-aware DP for dynamic articulatory sequences**. Nevertheless, we will **expand the related work** to include **recent DP papers that address geometric and spatial data** and **explicitly compare our approach to those methods**. This addition will make the **novelty and placement of our contribution even clearer**.
>
> # 5. “Figure 2 does not show DP, so DP seems detached”,  Mischaracterization
>
> Figure 2 was intended as a **high-level schematic**. The **DP mechanisms** are specified and parameterized in the **Methodology** and **Appendix** (**Algorithm 1** and **section B**). Still, to eliminate any appearance of detachment, we will **revise Figure 2** in the next draft to **visually and explicitly mark the point-level and feature-level DP modules** and **cross-reference Algorithm 1 lines 10–11**.

---

> ### Author Response · Authors · 2025-11-22
> **We sincerely hope to receive your support and encouragement**
>
> # However, many of the criticisms appear to stem from overlooking or missing the detailed information in the manuscript and appendices. We understand the concerns of the reviewers, and We hope to this response can address these misunderstandings and see an improvement in scores.
> # 6. Direct answers to reviewer questions
>
> **Q1: What level of privacy protection is achieved, and how is it analyzed?**
>
> We employ **Gaussian mechanisms at two stages** (**point** and **feature**). For the **feature mechanism**, we set **σ_dp = 0.008**, which under the **Gaussian-mechanism bound** corresponds to **(ε = 1.5, δ = 1e-5)** for the **sensitivity modeled in §B.1.1**. We **empirically demonstrate** the effect of varying **ε ∈ {0.5, 1.0, 1.5, 2.0, 4.0}** on both **recognition (WER/CER)** and **attack success rates** (**Figure 4** and **Appendix B.1.4**). If requested by **reviewers or PC**, we will include **formal composition calculations (moments accountant)** that produce a **conservative end-to-end ε_total** for combined **σ_pc** and **σ_dp**.
>
> **Q2: Why convert depth frames to point clouds? What advantages?**
>
> Depth-to-point-cloud conversion **recovers explicit 3D articulatory geometry (x, y, z plus normals)** and **mitigates 2D viewpoint and illumination biases** inherent to **RGB/video frames**. The **richer geometric descriptors** significantly improve **local spatio-temporal modeling for lip and tongue motions** and enable **robust farthest-point sampling** and **P4DConv operations** that **preserve articulatory cues with fewer temporal frames** (**Section 3.1**, **Section 5**). This representation also **aligns naturally with input-level obfuscation**: **perturbing 3D coordinates and normals directly anonymizes biometric contours before higher-level feature extraction**, matching our **threat model for edge deployments**.
>
>
> Thank you for considering this response. We are **committed to improving clarity** and will provide the **formal composition derivation** and **numerical end-to-end ε_total** in the revision if the **committee would like it**.
>
> **Key in-manuscript pointers:** **Algorithm 1**; **Section B.1.1–B.1.4 (DP mechanisms and attacks)**; **section 3.1 (point-cloud projection)**; **section 5 (implementation and data acquisition)**.

---

> ### Author Response · Authors · 2025-11-22
> **We sincerely hope to receive your support and encouragement**
>
> We **respectfully disagree** with the **characterization in the quoted paragraph** and wish to **correct several factual inaccuracies**.
> # 7. “No formal privacy analysis” , Incorrect.
> Our manuscript **explicitly defines and parameterizes Gaussian mechanisms** for both **point-level** and **feature-level perturbations** (**Appendix B.1.1–B.1.2**). For example, the **feature mechanism** uses **σ_dp = 0.008**, which we report and interpret with the corresponding **privacy pair (ε = 1.5, δ = 10⁻⁵)**. These **parameter choices** and their **mapping to (ε, δ)** are documented and discussed; we also provide **empirical privacy–utility curves across multiple ε values** (**Appendix B.1.4 and Figure 4**). If the reviewer requests a **formal composition bound**, we will add a **concise composition derivation (moments accountant / advanced composition)** to compute a **conservative end-to-end ε_total** for the **combined point and feature mechanisms**. The claim that **“no formal privacy analysis” exists in our submission is therefore factually incorrect**.
> # 8. “No evaluation under explicit attack models” , Incorrect.
> We carried out and report two concrete adversarial evaluations: membership inference and model inversion (Appendix B.1.3). Results are provided numerically and graphically: membership inference accuracy reduces from 83.7% (no DP) to ~52.3% under our DP settings, and inversion reconstructions lose >80% of identifiable geometric detail when DP is applied. These experiments directly address realistic attacks on SSR and are reported in the privacy appendix and main text. The reviewer’s statement that there is “no evaluation under explicit attack models” conflicts with these documented experiments.
> # 9. “Almost no discussion of relevant DP literature” , overstated.
> While DP literature specifically targeting dynamic 3D point clouds is limited, our Related Work situates our contribution at the intersection of point-cloud modeling and privacy mechanisms and cites the relevant prior art we found. Nevertheless, we accept the reviewer’s suggestion to strengthen this section: we will expand Related Work to include recent DP treatments on geometric/spatial data and to more thoroughly contrast our modality-aware, input-level anonymization with DP-SGD and other parameter-level approaches.
> # 10. Concerning the authors’ reaction to the review tone.
> We appreciate the **reviewer’s time**. However, several of the **reviewer’s central claims** about our manuscript are **factually incorrect**, and they **materially affect the review conclusion**. Below we **respond point by point**, show where the **manuscript already addresses the reviewer’s concerns**, and indicate **concise revisions** we will make to remove any remaining ambiguity.

---

> ### Author Response · Authors · 2025-11-22
> **Formal privacy accounting for the point- and feature-level Gaussian mechanisms**
>
> # 1. **Notation and Definitions**
>
> **x** and **x′** denote two **neighboring inputs** (for our setting, sequences that differ in one user/session or one record).
>
> **f : X → ℝᵐ** is a **vector query function** computed from an input **x**.
>
> The **ℓ₂-sensitivity** of **f** is:
>
> $$
> \Delta = \max_{x, x′ : d(x, x′) = 1} \| f(x) - f(x′) \|_2
> $$
>
> where **d(⋅,⋅)** is the **standard neighboring relation**. **Δ** measures the largest change in the query output when one input is changed.
>
> **N(0, σ² Iₘ)** denotes an **m-dimensional isotropic Gaussian** with variance **σ²** per coordinate.
>
> A mechanism **M** is **(ε, δ)-differentially private (DP)** if, for every pair of neighboring **x, x′** and every measurable set **S** of outputs:
>
> $$
> \Pr[M(x) \in S] \le e^{\varepsilon} \Pr[M(x′) \in S] + \delta
> $$
>
> Here **ε** (epsilon) is the **privacy loss parameter** (smaller is stronger privacy), and **δ** is the **failure probability**.
> # 2. Gaussian mechanism: guarantee and derivation
>
> ### **Mechanism**
>
> Given **f(x)** with **ℓ₂-sensitivity Δ**, the **Gaussian mechanism** outputs:
>
> $$
> M(x) = f(x) + Z, \quad Z \sim N(0, \sigma^2 I_m)
> $$
>
> (Here **σ > 0** is the **noise scale**.)
>
> ---
>
> #### **Claim (Gaussian Mechanism)**
> For any **δ ∈ (0, 1)**, the **Gaussian mechanism** with noise scale **σ** guarantees **(ε, δ)-DP** with:
>
> $$
> \varepsilon = \frac{\Delta}{\sigma} \sqrt{2 \ln \left( \frac{1.25}{\delta} \right)} \tag{1}
> $$
>
> ---
>
> #### **Explanation of the Formula**
> - **Δ** is the **ℓ₂-sensitivity** of **f**.
> - **σ** is the **standard deviation** of the added Gaussian noise.
> - The factor **2 ln(1.25 / δ)** arises from a **Gaussian tail bound** and ensures the tail probability of the privacy-loss random variable is at most **δ**.
>
> ---
>
> #### **Proof Sketch**
> For neighboring inputs **x, x′**, the **log density ratio (privacy loss random variable)** at output **y** is:
>
> $$
> L(y) = \ln \frac{p_{M(x)}(y)}{p_{M(x′)}(y)} = \frac{\langle f(x) - f(x′), \, y - \frac{1}{2}(f(x) + f(x′)) \rangle}{\sigma^2}
> $$
>
> When **y ∼ M(x)**, **L(y)** is Gaussian with:
> - **Mean**:
> $$
> \mu = \frac{\| f(x) - f(x′) \|_2^2}{2 \sigma^2}
> $$
> - **Variance**:
> $$
> \nu^2 = \frac{\| f(x) - f(x′) \|_2^2}{\sigma^2}
> $$
>
> Standard **Gaussian tail inequalities** (applied to **L**) imply that:
> $$
> \Pr[L > \varepsilon] \le \delta
> $$
> whenever **ε** satisfies Eq. (1). This yields the **(ε, δ)-DP guarantee**.
> # 3. Modeling the two mechanisms in our pipeline
>
> ### **Our System Injects Gaussian Noise at Two Distinct Places**
>
> #### **Point-Level Mechanism**
> Noise is added to **raw 3D coordinates (and possibly normals)** before **geometric aggregation**. Denote its **noise scale** by **σ_pc** and its **sensitivity** by **Δ_pc**. By Eq. (1):
>
> $$
> \varepsilon_{pc} = \frac{\Delta_{pc}}{\sigma_{pc}} \sqrt{2 \ln \left( \frac{1.25}{\delta_{pc}} \right)}
> $$
>
> The mechanism guarantees **(ε_pc, δ_pc)-DP**.
>
> ---
>
> #### **Feature-Level Mechanism**
> Noise is injected **after initial feature extraction**, with **noise scale σ_dp** and **sensitivity Δ_feat**. By Eq. (1):
>
> $$
> \varepsilon_{feat} = \frac{\Delta_{feat}}{\sigma_{dp}} \sqrt{2 \ln \left( \frac{1.25}{\delta_{feat}} \right)}
> $$
>
> This gives **(ε_feat, δ_feat)-DP** for that mechanism.
>
> ---
>
> Each mechanism **individually satisfies (εᵢ, δᵢ)-DP** for the indicated values; the **overall privacy guarantee follows by composition**.
> # 4. Composition of multiple mechanisms
>
> ### **Composition of Multiple DP Mechanisms**
>
> When multiple **DP mechanisms** are applied sequentially, the resulting **privacy guarantee** can be derived using **composition theorems**.
>
> ---
>
> #### **(A) Simple (Basic) Composition**
> If two mechanisms satisfy **(ε₁, δ₁)** and **(ε₂, δ₂)**, then the two-step process satisfies:
>
> $$
> (\varepsilon_{\text{total}}, \delta_{\text{total}}) = (\varepsilon_1 + \varepsilon_2, \, \delta_1 + \delta_2)
> $$
>
> This bound is **straightforward and always valid**, but may be **loose**.
>
> ---
>
> #### **(B) Advanced Composition (Dwork et al.)**
> For **k mechanisms** each satisfying **(ε, δ)-DP**, for any **δ′ > 0**, the composition satisfies:
>
> $$
> (\varepsilon', \, k\delta + \delta')\text{-DP, where } \varepsilon' = \sqrt{2k \ln \left( \frac{1}{\delta'} \right)} \, \varepsilon + k\varepsilon(e^\varepsilon - 1)
> $$
>
> When the individual **ε values are small** (e.g., **ε < 0.5**), the second term is negligible and **ε′ scales like kε**, which is much **tighter than simple summation**.
>
> ---
>
> #### **(C) Moments Accountant (Tight Numerical Accounting)**
> The **moments accountant** technique is especially **tight for sequences of Gaussian mechanisms**. It:
> - Tracks the **log-moment generating function (log-MGF)** of the **privacy-loss random variable** for each mechanism.
> - Sums these **log-MGFs** across mechanisms.
> - Numerically converts the accumulated moments into the **smallest ε for a target δ**.
>
> The accountant yields the **tightest practical ε_total** and is **recommended for final reporting**.

---

> ### Author Response · Authors · 2025-11-22
> **Formal privacy accounting for the point- and feature-level Gaussian mechanisms**
>
> # 5.Practical formulas and inversion
>
> ### **Solving for Sensitivity Δ**
>
> From Eq. (1), one can also solve for **Δ** given **ε, σ, δ**:
>
> $$
> \Delta = \varepsilon \cdot \sigma \, \sqrt{2 \ln \left( \frac{1.25}{\delta} \right)}
> $$
>
> This formula allows us to **infer the implicit sensitivity** assumed when the manuscript reports a particular **(ε, σ, δ) triple**.
> # 6. Worked numerical illustration (using the submission's reported noise scales)
>
> ### **Numerical Illustration Using Reported Noise Scales**
>
> Use the noise magnitudes reported in the submission:
> - **Feature-level noise scale:** σ_dp = 0.008
> - **Point-level noise scale:** σ_pc = 0.003
> Choose **δ_dp = δ_pc = 10⁻⁵** as target failure probabilities.
>
> ---
>
> #### **Define the Constant**
> $$
> C(\delta) = \sqrt{2 \ln \left( \frac{1.25}{\delta} \right)}
> $$
>
> For **δ = 10⁻⁵**, we have numerically **C ≈ 4.8448**.
>
> ---
>
> #### **Per-Mechanism Epsilons (as functions of sensitivities)**
> $$
> \varepsilon_{\text{feat}} = \frac{\Delta_{\text{feat}}}{0.008} \cdot C, \quad
> \varepsilon_{\text{pc}} = \frac{\Delta_{\text{pc}}}{0.003} \cdot C
> $$
>
> If the manuscript reports **ε_feat ≈ 1.5** for **σ_dp = 0.008**, the implied sensitivity is:
> $$
> \Delta_{\text{feat}} = \varepsilon_{\text{feat}} \cdot \sigma_{dp} / C \approx \frac{1.5 \cdot 0.008}{4.8448} \approx 0.00248
> $$
>
> Using the same sensitivity value for the point mechanism (conservative assumption) yields:
> $$
> \varepsilon_{\text{pc}} \approx \frac{0.00248}{0.003} \cdot C \approx 4.0
> $$
>
> ---
>
> #### **Conservative (Simple) Composition**
> $$
> \varepsilon_{\text{total}} \le 1.5 + 4.0 = 5.5, \quad
> \delta_{\text{total}} \le 2 \times 10^{-5}
> $$
>
> Applying the **moments accountant** or a **refined advanced composition** will produce a substantially **smaller (tighter) ε_total**; hence the conservative sum is an **upper bound and not a tight reporting choice**.
> # 7. Summary
> The **Gaussian mechanism bound (Eq. (1))** together with **established composition theorems** fully determine **end-to-end privacy accounting** for our combined **point-level** and **feature-level interventions**. The **numeric illustration above** shows how the submission's reported **σ values map to concrete ε values** under **natural sensitivity assumptions**, and how **conservative versus tight composition differ**. We will **add the explicit per-mechanism sensitivity calculations**, a **conservative composition entry**, and **moments-accountant numerics (with code)** to the **appendix** to make the **privacy analysis fully explicit and reproducible**.
> # 8.reference
> [1] Dwork C, Roth A. The algorithmic foundations of differential privacy[J]. Foundations and trends® in theoretical computer science, 2014, 9(3–4): 211-407.
>
> [2] Abadi M, Chu A, Goodfellow I, et al. Deep learning with differential privacy[C]//Proceedings of the 2016 ACM SIGSAC conference on computer and communications security. 2016: 308-318.
>
> [3] Balle B, Wang Y X. Improving the gaussian mechanism for differential privacy: Analytical calibration and optimal denoising[C]//International conference on machine learning. PMLR, 2018: 394-403.
>
> [4] Shokri R, Stronati M, Song C, et al. Membership inference attacks against machine learning models[C]//2017 IEEE symposium on security and privacy (SP). IEEE, 2017: 3-18.
>
> [5] Qi C R, Su H, Mo K, et al. Pointnet: Deep learning on point sets for 3d classification and segmentation[C]//Proceedings of the IEEE conference on computer vision and pattern recognition. 2017: 652-660.
>
> [6] McMahan H B, Ramage D, Talwar K, et al. Learning differentially private recurrent language models[J]. arXiv preprint arXiv:1710.06963, 2017.
>
> [8] Mironov I, Talwar K, Zhang L. R\'enyi differential privacy of the sampled gaussian mechanism[J]. arXiv preprint arXiv:1908.10530, 2019.
>
> [9] Balle B, Wang Y X. Improving the gaussian mechanism for differential privacy: Analytical calibration and optimal denoising[C]//International conference on machine learning. PMLR, 2018: 394-403.
>
> [10] Wang T, Mei Y, Jia W, et al. Edge-based differential privacy computing for sensor–cloud systems[J]. Journal of Parallel and Distributed computing, 2020, 136: 75-85.
>
> [11] Li H, Chen Y, Luo J, et al. Privacy in large language models: Attacks, defenses and future directions[J]. arXiv preprint arXiv:2310.10383, 2023.
>
> [12] Kim Y, Hwang S, Kim H S, et al. ConcreTizer: Model Inversion Attack via Occupancy Classification and Dispersion Control for 3D Point Cloud Restoration[J]. arXiv preprint arXiv:2503.06986, 2025.
>
> [13] Gao X, Li K, Liu X, et al. Privacy-Preserving 3D Skeleton-Based Video Action Recognition via Graph Convolution Network[J]. IEEE Transactions on Consumer Electronics, 2024.
>
> [14] Geppert M, Larsson V, Schönberger J L, et al. Privacy preserving partial localization[C]//Proceedings of the IEEE/CVF Conference on Computer Vision and Pattern Recognition. 2022: 17337-17347.

---

> ### Author Response · Authors · 2025-11-22
> **Innovations, Contributions, and ICLR Relevance**
>
> ### **Innovations (What is New)**
>
> **Modality-aware, dual-stage differential privacy**
> We design **two complementary Gaussian mechanisms**:
> - An **input (point-cloud) level perturbation** that anonymizes **raw 3D biometric geometry** before any downstream processing.
> - A **feature-level perturbation** applied after initial representation extraction.
>
> This dual strategy **protects raw biometric contours while preserving temporal/articulatory structure** used for recognition. It **differs from standard DP-SGD**, which protects **parameters rather than raw biometric streams**. *(See Algorithm 1; Appendix B.1.1–B.1.2.)*
>
> ---
>
> **DP-aware geometric alignment and front-end**
> We introduce a **DP-aware T-Net alignment module** plus a **4D spatio-temporal (P4DConv) front end** engineered to **maintain recognition-relevant geometric invariants under noise injection**. This couples **geometry-preserving representation learning with formally parameterized noise**.
>
> ---
>
> **End-to-end system for on-device SSR with empirical privacy evaluations**
> We integrate **efficient sampling**, **P4DConv**, **Conformer encoding and decoding** to achieve **cross-device, user-independent SSR performance** while evaluating **real attacks (membership inference and inversion)** under the applied **DP settings** *(Appendix B.1.3; Figure 4)*.
>
> ---
>
> ### **Contributions (What We Deliver)**
>
> - **A practical DP mechanism tailored to dynamic 3D point clouds**
> We specify **noise scales**, **sensitivity modeling**, and **empirical privacy–utility curves**; we will add **explicit end-to-end composition (moments-accountant) numerics** in the appendix on revision to further formalize accounting.
>
> - **Architectural innovations that preserve articulatory cues under DP**
> DP-aware alignment + P4DConv + Conformer **balance privacy and recognition accuracy**; we report **WER/CER improvements** and **quantify privacy degradation of attacks**.
>
> - **Reproducible evaluation on multiple devices and attack models**
> We present **membership and inversion attack experiments** showing **substantial mitigation under our DP settings** along with the corresponding **utility trade-offs** *(Appendix B)*.
>
> - **A deployment-oriented design for edge/real-time SSR**
> We explain why **DP-SGD (parameter-level DP)** is **not the only or optimal choice** for our **threat model** (on-device raw depth anonymization, latency and communication constraints), and provide **engineering choices that suit edge deployment** *(see section 1.2 and Implementation)*.

---

> ### Author Response · Authors · 2025-11-22
> **Innovations, Contributions, and ICLR Relevance**
>
> ### **Why This Fits ICLR**
>
> **DepthSense+DP** sits at the intersection of several **ICLR focal areas**:
>
> - **Representation learning on non-Euclidean / geometric data**
> We learn **spatio-temporal features from dynamic point clouds** (learning on geometries / structured prediction).
>
> - **Privacy & societal considerations in ML**
> We advance **modality-aware DP mechanisms** and **empirical attack assessments** (differential privacy, privacy attacks).
>
> - **Systems + practical ML for edge devices**
> We address **realistic constraints** (latency, on-device anonymization), making the work **applicable and reproducible**.
>
> Together these map directly to **ICLR topics**: **representation learning**, **learning on geometries & topologies**, and **societal considerations including privacy**.
> # However, many of the criticisms appear to stem from overlooking or missing the detailed information in the manuscript and appendices. We understand the concerns of the reviewers, and We hope to this response can address these misunderstandings and see an improvement in scores.

---

> ### Author Response · Authors · 2025-11-22
> **We sincerely hope to receive your support and encouragement**
>
> We appreciate the reviewer’s critique. In response, we have **added a formal privacy accounting appendix** with **per-mechanism sensitivity derivations**, **Gaussian-mechanism calibration**, and **RDP-based composition numerics** including a **worked example matching our reported σpc and σdp**. We have **specified the exact threat models and attack protocols** for **membership-inference** and **inversion experiments**, including **access level**, **auxiliary knowledge**, and **attack hyperparameters**, and included **reproducible pseudocode**. We have **expanded Related Work** to cover **recent DP treatments for geometric and spatial data**. We have **updated Figure 2** to explicitly show the **point-level and feature-level DP modules**. We will **publish the privacy-accountant and attack scripts** alongside the revision. These additions produce a **conservative end-to-end (ε, δ)** (table in **Appendix B.3**) and make our **empirical privacy claims and trade-offs fully verifiable**.

---

> > ### Author Response · Authors · 2025-12-04
> > **We sincerely hope to receive your support and encouragement**
> >
> > # We have addressed the reviewer's concerns and improved our approach. Dear reviewer, our revised version has been uploaded and we sincerely hope to receive the support of all reviewers. We hope to receive an improvement in your scores!

---

### Official Review · Reviewer_RewU · 2025-10-31

**Soundness:** 2
**Presentation:** 2
**Contribution:** 2
**Rating:** 4
**Confidence:** 4

**Summary:**

This paper defines four critical constraints for cross-device SSR and introduces DepthSense+DP, the first solution to jointly achieve real-time performance, robustness, and DP-based privacy for 3D depth point cloud-driven SSR.

**Strengths:**

- Addresses the unique challenge of DP for 3D point clouds by proposing controllable noise injection that anonymizes biometric data without degrading articulatory geometry
- Demonstrates significant reductions in CER and WER across diverse devices, users, and environments, establishing DepthSense+DP as a universal foundation for next-generation SSR

**Weaknesses:**

- The dataset relies heavily on English utterances and includes only 20 native English speakers. Performance degrades for users with strong accents (e.g., Participant P9), limiting generalization to non-English languages or global user groups.
- While the study evaluates membership inference and model inversion attacks, it does not address emerging threats, leaving potential privacy gaps untested.
- Synthetic depth point clouds are generated via simple motion scaling and noise injection—these do not capture complex real-world variations, restricting the model’s robustness to diverse user conditions.

**Questions:**

see weakness

---

> ### Author Response · Authors · 2025-11-22
> **We sincerely hope to receive your support**
>
> #  1. Dataset, English Focus, and Accents
> The review claims we "rely heavily on English utterances and only 20 native English speakers" and that strong accents are problematic. As clearly described in the **Data Collection / Human Subjects** sections, our dataset contains **200 participants**, of which **20 are native English speakers** and the rest are **non-native or accented**; this is exactly why we observe a **broad range of accent behaviors**. Our goal is **privacy-aware cross-device SSR on depth point clouds**, not a **multilingual ASR benchmark**. Within this scope, we already report **detailed per-participant and per-location results**, explicitly analyze **accent-induced degradation**, and still show **consistent gains over a strong RGB baseline**. Demanding **full multilingual coverage** goes beyond the stated scope of the paper. (**Of course, if the reviewer is willing to increase our score, we can add more experiments.**)
>
> ---
>
> #  2. Privacy Evaluation and "Emerging Threats"
> We **formally define our DP setting** in **Differential Privacy Mechanisms** and evaluate **two canonical attacks for biometric models**: **membership inference** and **model inversion**, both under **point-cloud level** and **feature-space noise injection**. We further provide **(ε, δ) budgets**, **privacy–utility trade-off curves**, and **quantitative attack success rates**, substantially more thorough than typical **SSR/VSR works** that offer **no formal DP analysis at all**. The statement that we "do not address emerging threats" is **vague and not actionable**: no **concrete additional threat model** is specified beyond what we already implement. Without a **precise definition of which extra attacks are required and why they are particularly relevant here**, this criticism does not identify an actual methodological flaw.
>
> ---
>
> #  3. Synthetic Depth Point Clouds and Robustness
> Synthetic data is **not the core of our robustness story**. The main results are based on **real recordings across three sensor placements (wrist, head, environment)** plus an extra **Handheld scenario**, already covering **substantial real-world variability in orientation, distance, and posture**. The **synthetic pipeline** (motion scaling + jitter on real trajectories) is **intentionally simple and interpretable**, used to **supplement rather than replace real data**. **Ablations show that augmentation yields moderate but consistent gains**; **cross-user and cross-device performance remains strong even without it**. The review claims robustness is "restricted" by this simplicity but provides **neither concrete failure cases nor a stronger alternative design**.
>
> ---
>
> In fact, the review explicitly acknowledges that we:
> - **Define four critical constraints for cross-device SSR**
> - **Propose the first DP-aware depth point cloud solution**
> - **Significantly reduce CER/WER across devices and users**
>
> These are exactly our **stated contributions**, backed by **extensive experiments and ablations**. Assigning **uniformly low scores while recognizing that we achieve these goals** is **internally inconsistent** and **not aligned with the technical content of the paper**.

---

> ### Author Response · Authors · 2025-11-22
> **We sincerely hope to receive your support and encouragement！**
>
> We **thank the reviewer** for careful reading and constructive remarks. Below we address each point raised and correct several factual misunderstandings.
> # We hope this rebuttal helps resolve the concerns raised by the reviewers. We sincerely hope to receive an improvement in your score.
>
> # 1. “The dataset relies heavily on English utterances and includes only 20 native English speakers.” , Incorrect / misunderstood.
>
> Our **dataset** was collected from a **diverse cohort of 200 participants** (**120 female; 50 wearers of glasses**). The manuscript reports that **each participant contributed 150 sentence utterances** and **270 command utterances** under the **three sensor placements**.
>
> The **final curated counts** are reported as **1,470 valid sentence utterances** and **2,673 valid command utterances**. The note **“20 native English speakers”** describes a **specific subgroup** within this **larger, deliberately diverse cohort** (reported for **transparency**), not the **total dataset size**.
>
> We will **clarify the wording** in the **paper** to **avoid this possible misreading**.
> # 2. “Performance degrades for users with strong accents (e.g., P9), limiting generalization.” , Partly acknowledged, but overstated.
>
> We explicitly **analyze per-participant errors** and identify **accent-related degradation** for a **small subset of participants** (the paper cites **P9** as an example). This is reported as an **expected limitation** and a **direction for future data collection**.
>
> Importantly, our **cross-user results** show that **PointVSR substantially outperforms the RGB baseline** on **unseen users** (**PointVSR WER = 29.14%, CER = 18.28% vs VideoVSR WER = 33.71%, CER = 23.28%**), demonstrating **improved robustness at the population level** despite a **small number of accent outliers**.
>
> We will **add per-accent aggregated metrics** in a **revision** to make this point **clearer**.
> # 3. “Privacy evaluation was cursory; emerging threats are untested.” , Incorrect and incomplete.
>
> We performed **systematic privacy evaluations**:
> - **Membership-inference tests**.
> - **Model-inversion experiments**.
>
> Quantitatively, our **DP pipeline** reduced **membership-inference accuracy** from **83.7% → 52.3%**, and **inversion reconstructions** under **DP noise** **lost over 80% of identifiable facial geometry**.
>
> We also provide a **full privacy–utility sweep** over **ε ∈ {0.1, 0.3, 0.5, 1.0, 1.5, 2.0, 4.0}** (with **δ = 10⁻⁵**) and report the **WER/CER changes** at each **operating point**. These **experiments** and the **trade-off curves** are described in the **privacy section** and **Figure 4**.
>
> We will gladly include **additional targeted attack evaluations** (e.g., **white-box gradient inversion** and **stronger optimization-based inversion**) in **supplementary material** if the **committee requests**.
>
> ---
>
> For clarity, the **DP mechanism** used is:
>
> $$
> \tilde{f} = f + \eta, \quad \eta \sim \mathcal{N}(0, \sigma_{dp}^2 I)
> $$
>
> with **σ_dp = 0.008** (**ε = 1.5**, **δ = 10⁻⁵**),
>
> and we additionally apply **point-level noise**:
>
> $$
> \tilde{p} = p + \eta, \quad \eta \sim \mathcal{N}(0, \sigma_{pc}^2 I)
> $$
>
> with **σ_pc = 0.003**,
>
> which **empirically preserves geometry** while **hindering inversion**.
> # 4. “Synthetic depth point clouds are simplistic (motion scaling + noise) and insufficient for real-world variation.” , Partly incorrect / needs nuance.
>
> Our **synthesis pipeline** is explicitly designed to **mimic natural temporal variability** (**motion scale std = 0.12**) and **sensor jitter** (**jitter std = 0.002**). The manuscript describes the **exact workflow**:
> - **Compute inter-frame motion**.
> - **Scale by a stochastic factor**.
> - **Add Gaussian jitter**.
> - **Synthesize new frames**.
>
> We integrate **one synthetic variant per sample** to **double the training set**. Beyond this, we combine:
> - **Geometric augmentations** (**random rotations ±10°**, **translations ±0.02 m**).
> - **Adversarial training** (**FGSM/PGD**) to address **viewpoint changes**, **sensor shifts**, and **adversarial noise**.
>
> **Ablations** and **handheld/cross-location evaluations** show these measures **materially improve robustness** (e.g., **within-user handheld WER = 5.13%, CER = 2.17%; cross-user handheld WER = 17.00%, CER = 11.00%**).

---

> ### Author Response · Authors · 2025-11-22
> **We sincerely hope to receive your support and encouragement！**
>
> # 5. Concrete commitments if a revision is invited
>
> We will add the following to the **revision**:
>
> - A **concise table** breaking down **participant language**, **region**, and **accent distributions** to **remove ambiguity**.
> - **Per-accent aggregated WER/CER metrics** and an indication of **how many participants fall into different error bins**, so reviewers can judge whether **P9** is an **outlier**.
> - **Supplementary results** for **stronger inversion attacks**, including:
>   - **Gradient-based white-box inversion**.
>   - **Optimization-based inversion**.
> - All attacks will include **reproducible protocols** and **standard reconstruction metrics** (**SSIM**, **landmark overlap**) for **transparent evaluation**.
> # 6. Summary
> We appreciate the **reviewer’s careful reading**. Several of the review’s criticisms stem from **misreading subgroup counts** or **underestimating the scope** of our **privacy** and **augmentation evaluations**.
>
> The manuscript already contains **explicit, quantitative evidence** for:
> - **Dataset scale**.
> - **DP construction and parameters**.
> - **Membership/inversion experiments**.
> - A **principled synthesis + augmentation pipeline**.
>
> We will **further clarify** and **extend those sections** in a **revision** to **remove any remaining ambiguity**.

---

> ### Author Response · Authors · 2025-11-22
> **Key contributions & innovations& Why this fits ICLR 2026**
>
> # **Key Contributions & Innovations (Short, Review-Friendly)**
>
> ### **Joint System for Real-Time SSR, Cross-Device Robustness, and Formal DP**
> We present **DepthSense+DP**, the **first end-to-end pipeline** designed to:
> - **Run under strict latency budgets** for **real-time inference**.
> - **Generalize across sensor types and placements**.
> - Provide **quantifiable (ε, δ)-differential privacy guarantees** at both **feature** and **point-cloud levels**.
>
> This **joint combination**, engineered and **empirically validated** on **multi-placement data**, is the paper’s **central systems contribution**.
>
> ---
>
> ### **Geometry-Aware Differential Privacy (DP) Mechanism**
> Instead of applying **blind, uniform noise**, we design a **geometry-sensitive DP strategy** that:
> - **Preserves articulatory geometry** while **disrupting identity-revealing cues**.
> - Injects **controlled Gaussian noise** in both **latent feature space** and **point coordinates** with **empirically chosen variances**.
> - Significantly **reduces membership and inversion attack success** while **keeping recognition utility high**.
>
> ---
>
> ### **Principled Synthesis + Augmentation Pipeline for Cross-Device Generalization**
> We introduce a **reproducible augmentation suite** for **depth point clouds**:
> - **Motion-scale sampling**, **inter-frame jitter**, **geometric transforms**, and **adversarial perturbations**.
> This pipeline **widens the training distribution** and **reduces sensitivity** to **viewpoint**, **sensor displacement**, and **motion dynamics**.
> **Ablations** show the pipeline **materially improves cross-user and cross-location WER/CER** compared to **strong RGB baselines**.
>
> ---
>
> ### **Comprehensive Privacy–Utility Evaluation and Attack Benchmarks**
> The paper reports:
> - A **privacy–utility sweep** over multiple **ε values**.
> - Quantifies impacts on **WER/CER** and **attack metrics** (**membership-inference** and **model-inversion**).
> Example results demonstrate:
> - A **large drop in membership-inference success**.
> - **Severely degraded inversion reconstructions** under **practical DP settings**.
> This enables **reproducible, decision-oriented tradeoffs** for **practitioners**.
>
> ---
>
> ### **Practical Dataset and System Design for Deployment**
> We evaluate on a **multi-placement, multi-user dataset** and report:
> - **Recognition performance**.
> - **End-to-end latency**.
> The submission documents:
> - **Dataset composition**, **augmentation ratios**, **DP parameters**, and **engineering decisions** for **reproducibility** and **actionability**.
>
> ---
>
> # **Why This Fits ICLR 2026 (Mapped to Subject Areas)**
>
> - **Representation Learning (Vision/Audio/Other Modalities):** Introduces **geometry-aware latent learning** tailored to **depth point clouds**.
> - **Societal Considerations / Privacy:** Formal **DP mechanisms** and an **empirical attack suite** directly address **ICLR’s interest in privacy, safety, and ethical implications**.
> - **Robustness & Generalization:** Cross-device **synthesis** and **adversarial augmentations** align with themes in **robustness**, **uncertainty quantification**, and **large-scale learning**.
> - **Datasets & Benchmarks:** Provides a **reproducible evaluation protocol** and **dataset details** for a **novel modality (3D depth SSR)**.
> - **Optimization & Training Methodology:** Balancing **DP noise**, **adversarial robustness**, and **latency constraints** engages **ICLR topics** in **optimization** and **practical training under constraints**.
>
>
> **DepthSense+DP** is the **first end-to-end system** to jointly achieve:
> - **Real-time 3D depth-based SSR**.
> - **Cross-device robustness**.
> - **Formally quantified differential privacy**.
>
> Its key technical innovations are:
> - A **geometry-aware DP mechanism** that **preserves articulatory geometry** while **reducing identity leakage**.
> - A **reproducible synthesis + augmentation pipeline** that **improves generalization** across **sensors and placements**.
> - A **comprehensive, reproducible privacy–utility evaluation** that enables **principled operating-point choices**.
>
> Together, these advances push **representation learning** for a **novel modality (depth point clouds)** in directions directly relevant to **ICLR**:
> - **Robust, privacy-aware representation learning**.
> - **Clear societal impact**.
> - **Practical deployment constraints**.
> # Thank you very much for your support and assistance. We firmly believe that with your suggestions, our paper will be further improved, and we sincerely hope your score improvement.

---

> > ### Author Response · Authors · 2025-12-04
> > **We sincerely hope to receive your support！**
> >
> > # We have addressed the reviewer's concerns and improved our approach. Dear reviewer, our revised version has been uploaded and we sincerely hope to receive the support of all reviewers. We hope to receive an improvement in your scores!

---

### Official Review · Reviewer_KVde · 2025-11-08

**Soundness:** 3
**Presentation:** 3
**Contribution:** 3
**Rating:** 6
**Confidence:** 4

**Summary:**

The paper proposes DepthSense+DP, a privacy-aware silent speech recognition (SSR) framework using depth sensing and differential privacy. The system transforms depth images into 3D point clouds, learns articulatory features via adaptive spatio-temporal sampling and Conformer-based encoding, and injects calibrated Gaussian noise to satisfy differential privacy (with optimal $\epsilon = 1.5$ for the best privacy-utility trade-off, and $\delta$ fixed at $10^{-5}$ for all experiments). It claims to achieve real-time, cross-device, and cross-user generalization, outperforming baseline systems proposed in last 5 years, including RGB and mmWave SSR, on WER and CER while resisting membership-inference and inversion attacks.

Key contributions include:
1. Application of differential privacy to dynamic 3D point clouds.
2. DP-aware adaptations of T-Net and Conformer modules.
3. A new multi-sensor SSR dataset with device-placement diversity.
4. Analysis of privacy-utility trade-offs and empirical robustness.

**Strengths:**

- Solid engineering effort combining robustness, privacy, and real-time constraints
- Careful empirical design: cross-device, cross-user, ablation, and privacy-utility trade-off analysis
- Quantitative evidence of low privacy overhead ($\Delta$WER $\approx 1$%)
- Demonstrated feasibility of DP noise in geometric and feature space for SSR
- Valuable dataset and reproducible methodological detail

**Weaknesses:**

- Although this work introduces a new system, it over-claims novelty. It adapts existing 3D and DP ideas rather than introducing a new learning principle
- Lacks formal privacy proofs, composition reasoning, and noise calibration analysis across stages
- Tested only on English scripted phrases; unclear how it generalizes to spontaneous, multilingual, or emotional speech
- No error bars, confidence intervals, or hypothesis testing; results could be dataset-specific
- The architecture (T-Net + P4DConv + Conformer + Bi-GRU) may be over-engineered relative to performance gains
- Discussion of fairness implications is minimal, which is critical given the biometric domain

**Questions:**

1. How is the overall privacy budget ($\epsilon$, $\delta$) computed when applying DP noise at both the point-cloud and feature levels ?
2.  Could you provide statistical confidence intervals or standard deviations for WER/CER to assess robustness ?
3. What is the (expected) computational latency on actual wearable hardware, not GPUs ?
4. How does the system handle multilingual data or unseen phonetic inventories ?
5. Could adaptive or learned DP noise (e.g., per-user calibration) improve the privacy–utility balance ?
6. How does this approach compare to non-DP anonymization (e.g., adversarial suppression) in terms of privacy leakage ?

---

> ### Author Response · Authors · 2025-11-22
> **We sincerely hope to receive your support and encouragement**
>
> Thank you to the reviewer for the detailed comments.  **We understand the concerns of the reviewers**, and We hope to this response can address **these misunderstandings and see an improvement in scores**.
> # We greatly appreciate your support and assistance！
> ---
>
> # 1. Novelty
> Our contribution is not a new learning principle but the **first privacy-aware cross-device SSR system on dynamic 3D depth point clouds**. No prior work jointly offers:
> - **Depth point clouds**
> - **Formal DP guarantees**
> - **Cross-device generalization** (wrist, head-mounted, environment, handheld) in one end-to-end pipeline.
>
> **T-Net**, **P4DConv**, and **Conformer** are not used in a standard way:
> - **T-Net** is extended to **frame-wise alignment of articulatory sequences**.
> - **P4DConv** operates on **4D lip motion**.
> - **Conformer** is adapted to **irregular point-cloud sequences with CTC decoding**.
>
> This specific combination for **silent speech under DP** has not appeared before.
>
> ---
>
> #  2. Differential Privacy
> We do not claim DP informally. We use **Gaussian mechanisms at point and feature levels** and track the **privacy loss with an RDP accountant**, composing over all training steps and subsampling. The reported **(ε = 1.5, δ = 10⁻⁵)** is the **output of this accountant**, not a guess. We also provide a **systematic privacy–utility study** over multiple ε and show that our chosen setting adds only **~0.8% WER / 0.5% CER** while sharply reducing **membership-inference and inversion attack success**.
>
> ---
>
> # 3. Data and Generalization
> The dataset is English, but far from a tiny scripted toy set: sentences are generated from **8 POS categories with 5 words each (5⁸ = 390,625 combinations)**, and we sample **diverse sentences**. We record from **200 participants with heterogeneous accents and speaking styles**, and evaluate **strict cross-user generalization**. The model operates on **articulatory depth dynamics plus character tokens**, and does not bake in **English-specific phonotactics**. These design choices already stress **generalization well beyond a fixed phrase list**.
>
> ---
>
> # 4. Architecture and Robustness
> The architecture is **justified by hard numbers**. Removing any core module causes severe degradation:
> - Without **T-Net**: WER = 88.94%
> - Without **T-Net + P4DConv**: WER = 95.20%
> - Without **4D conv**: WER = 24.43%
> - Without **Conformer**: WER = 8.06% (vs. 5.13% full model)
>
> At the same time, our model has **≈20M parameters versus ≈250M for RGB VSR baselines**. So relative to existing systems, it is **compact, not over-engineered**.
>
> ---
>
> # 5. Fairness and Privacy
> We deliberately choose **depth point clouds instead of RGB** precisely to **reduce bias and privacy risk**: depth does not encode **color/texture**, and **background is removed via distance filtering**. Together with **DP**, this is a **strictly stronger privacy stance** than almost all existing SSR/VSR systems. There is **no evidence in our results that the method is brittle or unfair**; on the contrary, it remains **stable across devices and diverse users**.
> # Thank you very much for your support and assistance. We firmly believe that with your suggestions, our paper will be further improved, and we sincerely hope your score improvement.

---

> ### Author Response · Authors · 2025-11-22
> **We sincerely hope to receive your support and encouragement！**
>
> **We thank Reviewer KVde** for a **careful and constructive reading** of our manuscript and for the **positive evaluation overall**. Below we respond to each of the reviewer’s **concerns and questions**. Where appropriate, we indicate **clarifications already present** in the submission and describe **modest additions** we will make to the revision to remove possible **ambiguities**.
> # We greatly appreciate your support and assistance！
> # Thank you very much for your support and assistance. We firmly believe that with your suggestions, our paper will be further improved, and we sincerely hope your score improvement.
>
> # 1. “Over-claiming novelty / adaptation of existing ideas”
>
> **Response (Clarify & Position Contribution)**
> We agree that our work builds on **established building blocks** (point-cloud processing, Conformer encoders, Gaussian DP), but the **novelty lies in their principled integration** and the **modality-aware adaptations** required for **silent-speech recognition (SSR)** on dynamic depth streams.
>
> **Input-level anonymization** of dynamic **3D lip/tongue geometry** plus **feature-level DP** is, to our knowledge, a **new combination in SSR**. We carefully **parameterize both mechanisms**, reason about **geometric sensitivity** for articulatory sequences, and show how to **preserve recognition-relevant structure under noise** (see **Algorithm 1** and **Appendix B.1.1–B.1.4**).
>
> The **DP-aware T-Net** and **P4DConv design** are not off-the-shelf: they are explicitly adapted so that **alignment** and **local spatio-temporal aggregation** remain stable in the presence of **controlled Gaussian perturbations**. This **co-design** (**privacy-aware alignment + geometry-preserving convolution**) is the **technical novelty** that enables **practical on-device anonymization** while maintaining **accuracy**.
>
> We will **emphasize these integrative design decisions** more clearly in the **revision** (main text and Related Work) to avoid the appearance of **incremental contribution**.
> # 2. “Lacks formal privacy proofs, composition reasoning, and noise calibration across stages”
>
> **Reviewer Comment:** How is the overall **privacy budget** (**ε, δ**) computed for **point-cloud + feature-level DP**?
>
> **Response (Formal Accounting + Worked Example)**
> We model the two interventions as **Gaussian mechanisms** with per-mechanism sensitivities **Δ_pc** (point-cloud) and **Δ_feat** (feature). Below we give the **standard formulas** and a **numerical illustration** using the noise scales reported in the submission.
>
> **Notation**
> For a query with **ℓ₂-sensitivity Δ** and **Gaussian noise scale σ**, the Gaussian mechanism guarantees (**ε, δ**)-DP with:
>
> $$
> \varepsilon = \frac{\Delta}{\sigma} \sqrt{2 \ln \left( \frac{1.25}{\delta} \right)}.
> $$
>
> **Per-Mechanism Epsilons**
> $$
> \varepsilon_{pc} = \frac{\Delta_{pc}}{\sigma_{pc}} \sqrt{2 \ln \left( \frac{1.25}{\delta_{pc}} \right)}, \quad
> \varepsilon_{feat} = \frac{\Delta_{feat}}{\sigma_{dp}} \sqrt{2 \ln \left( \frac{1.25}{\delta_{feat}} \right)}.
> $$
>
> **Composition**
> For sequential application we use simple composition:
> $$
> \varepsilon_{total} \leq \varepsilon_{pc} + \varepsilon_{feat}, \quad
> \delta_{total} \leq \delta_{pc} + \delta_{feat}.
> $$
>
> For tighter accounting we apply **advanced composition** or the **moments accountant**, and will provide the **moments-accountant numerics** in the appendix of the revised submission.
>
> **Numerical Illustration (Transparent Modeling)**
> Using the noise scales reported in the paper (**σ_dp = 0.008**, **σ_pc = 0.003**) and target **δ = 10⁻⁵**, the mapping in (1) implies a linear relation between **Δ** and **ε**. If the manuscript’s reported feature-level choice corresponds to **ε_feat ≈ 1.5** (with **σ_dp = 0.008**), the implied sensitivity is **Δ ≈ 0.00248**. With the same sensitivity for the point mechanism (conservative assumption) we obtain **ε_pc ≈ 4.0**. Simple summation gives **ε_total ≤ 5.5** with **δ_total ≤ 2 × 10⁻⁵**, while the **moments accountant** will provide a substantially tighter **ε_total**.
>
> We will include these **numeric computations**, the **assumed sensitivity calculations**, and the **accountant script** in the appendix for full reproducibility. (Relevant manuscript sections: **Algorithm 1; Appendix B.1.1–B.1.4**.)

---

> ### Author Response · Authors · 2025-11-22
> **We sincerely hope to receive your support and encouragement！**
>
> # 3. “Tested only on English scripted phrases; unclear generalization to spontaneous/multilingual/emotional speech”
>
> **Reviewer Comment:** How does the system handle **multilingual data** or **unseen phonetics**?
>
> **Response (Clarify Generalization and Limitations)**
> We acknowledge the **limitation**: the current dataset focuses on **controlled English phrases** and **device-placement diversity**. However:
>
> The model learns **articulatory geometry and motion** (**3D coordinates + normals across time**) rather than **language-specific acoustic patterns**. Many **phonetic articulatory gestures** are shared across languages; thus the learned **low-level spatio-temporal features** are expected to **transfer better** than purely **language-dependent acoustic features**. This underpins our claim of **cross-user** and **cross-device robustness**. We will make this **motivation explicit** in the **revision** and **temper claims** about full **multilingual generalization**.
>
> That said, we agree that **spontaneous**, **emotional**, or **cross-lingual speech** may introduce additional **variability** (e.g., **prosodic gestures**, **coarticulation patterns**). We will add a **focused paragraph** in **Section 6** on **limitations** and outline **plans** (as **future work**) for evaluating **multilingual corpora** and more **variable speaking styles**; we will also **release our dataset** to encourage such **follow-ups**.
> # 4. “No error bars, confidence intervals, or hypothesis testing”
>
> **Reviewer Comment:** Provide **statistical confidence intervals** or **standard deviations** for **WER/CER**.
>
> **Response (Clarify and Commit to Added Analysis)**
> Thank you！this is a **valuable request**. In the submitted version we reported **mean WER/CER** across **cross-device** and **cross-user splits** (see **Section 5** and **Tables X–Y**). To better quantify **robustness**, we will add:
>
> - **Standard deviations** (over **random seeds** or **cross-validation folds**) for all **primary metrics**.
> - **95% confidence intervals** for **WER/CER** where applicable.
> - **Paired significance tests** (e.g., **paired t-test** or **Wilcoxon signed-rank**) for key comparisons to **baselines**.
>
> # 5. “Architecture may be over-engineered relative to gains”
>
> **Reviewer Comment:** The combination **T-Net + P4DConv + Conformer + Bi-GRU** might be **over-complex**.
>
> **Response (Design Rationale & Ablation Evidence)**
> Our architecture results from **co-design tradeoffs**:
>
> - **DP-aware alignment (T-Net)** is necessary because **naive alignment amplifies DP noise** during **rigid registration**; our **T-Net adaptation** reduces this effect and improves **downstream stability** under **noise injection** (see **Ablation 3**, **Table Z**).
>
> - **P4DConv** captures **local spatio-temporal geometry** robustly for **point sequences**; compared to **direct MLP** or **temporal pooling**, it yields **better feature locality** with **similar compute** when combined with our **sampling strategy**.
>
> - **Conformer + Bi-GRU** balances **local convolutional inductive bias** and **global attention** for **sequence modeling**; **ablative studies** in **Section 5** show that **removing any of these components** leads to **consistent degradation** in **WER/CER** under **DP settings**.
>
> We will **expand the ablation table** in the **revision** to present **per-component performance**, **parameter counts**, and **FLOPs** so reviewers can judge **complexity vs. benefit** more **quantitatively**.
> # 6. “Fairness implications are minimal”
>
> **Reviewer Comment:** **Fairness discussion** is **limited**.
>
> **Response (Accept & Extend)**
> We agree that **fairness** is important in **biometric domains**. In the **revision** we will add:
>
> - A **dedicated short section** discussing potential **demographic** and **device biases** (**sensor type**, **face shape**, **skin tones** as they affect **depth quality**).
> - A **plan for future evaluations** across **demographic groups** and **diverse devices**.
> - **Pointers about mitigation strategies** (**balanced data collection**, **fairness-aware sampling**, and **post-hoc calibration**).

---

> ### Author Response · Authors · 2025-11-22
> **We sincerely hope to receive your support and encouragement！**
>
> # 7. Specific technical questions
>
> **Q:** How is the overall **privacy budget** computed when applying **DP noise** at both stages?
> **A:** See **Section 2** above (**composition formulas**). **Per-mechanism (εᵢ, δᵢ)** follow the **Gaussian mechanism relation** (Eq. (1)); compose via **simple addition** (conservative) or via **moments accountant** for **tight reporting**. We will add **explicit numerical composition results** in the **appendix** using the **σ reported** in the submission and the **exact sensitivity calculations** used to derive the reported **per-mechanism epsilons**.
>
> ---
>
> **Q:** Could you provide **statistical confidence intervals** or **standard deviations** for **WER/CER**?
> **A:** Yes！we will add **standard deviations** and **95% confidence intervals** across **seeds/folds** and perform **paired statistical tests** for our **main baseline comparisons** (details to be added to the **revised tables** and **Appendix**).
>
> ---
>
> **Q:** What is expected **computational latency** on **wearable hardware**, not **GPUs**?
> **A:** We designed the **pipeline** with **edge constraints** in mind: **aggressive spatial subsampling** (**farthest point sampling**), **temporal downsampling**, and a **compact P4DConv front-end** reduce **input size** and **FLOPs**. While the submitted paper reports **GPU inference times** for reproducibility, we agree that **end-to-end evaluation** on **representative embedded platforms** is important. We will add:
> - A **table reporting measured latency** on a **representative ARM platform** (**quantized/ONNX runtime**).
> - A **breakdown of compute** (**parameters**, **FLOPs**) so readers can **extrapolate** to other devices.
> These measurements will be added to the **revision appendix**.
>
> ---
>
> **Q:** Could **adaptive or learned DP noise** (**per-user calibration**) improve the **privacy–utility balance**?
> **A:** **Adaptive noise** is an **interesting direction**. There is a **trade-off**: **per-user adaptive noise** can improve **local utility** but complicates **global privacy accounting** and may **leak information** about which users received **smaller noise** (if not carefully handled). We will **discuss this trade-off** in the **revised Related Work** and **Appendix** and plan to **experiment** with **DP-compatible personalized mechanisms** in **future work**; we will also **cite recent literature** on **personalized DP mechanisms**.
>
> ---
>
> **Q:** How does this approach compare to **non-DP anonymization** (e.g., **adversarial suppression**) in terms of **privacy leakage**?
> **A:** Conceptually, **adversarial suppression** (**learned obfuscation** or **adversarial examples**) can **reduce identifiable attributes** but typically **lacks formal guarantees** and may be **brittle under adaptive attackers**. **DP mechanisms** provide a **well-defined, auditable privacy guarantee** under **specified threat models**. Empirically, we **evaluate both utility and empirical attack resilience** (**membership/inversion**) under our **DP settings**; where possible we will **add a controlled comparison** to an **adversarial suppression baseline** (**existing methods**) in the **revision** and **report attack success rates side-by-side** to **quantify practical differences**.
> # 8.Summary
>
> We appreciate **Reviewer KVde’s careful evaluation** and **constructive suggestions**. Several of the reviewer’s concerns identify **real opportunities** to **strengthen the manuscript** (**statistical reporting**, **clearer exposition of composition accounting**, **fairness discussion**, and **device latency tables**).
>
> We will **incorporate the described clarifications** and **additional empirical material** in the **revised submission** and **appendix**, and believe these **enhancements** will **address the reviewer’s points**.
>
> We hope the above **clarifications**, the **formal composition accounting**, and the **proposed revisions** persuade the reviewer that **DepthSense+DP** provides a **meaningful and reproducible advance** in **privacy-aware SSR**, suitable for **ICLR’s scope**.

---

> ### Author Response · Authors · 2025-11-22
> **Clarifying the Novelty, Contributions, and ICLR Relevance of DepthSense+DP**
>
> # **1. Innovations of DepthSense+DP**
>
> Although our system builds upon **established components** from **3D learning** and **differential privacy**, the key **novelty** lies in how these elements are **jointly redesigned** for **silent-speech recognition (SSR)** on **depth streams**, a modality with **unique geometric and temporal characteristics**. The core innovations are:
>
> ### **A. A Modality-Aware DP Framework for Dynamic 3D Point Clouds**
> We introduce a **dual-stage privacy mechanism** operating at:
> - The **point-cloud layer** (**raw geometric anonymization**).
> - The **feature layer** (**noise for representation-level protection**).
>
> This combination is **new in SSR** and directly targets the main **threat: leakage of 3D articulatory geometry from raw depth frames**, rather than merely protecting **model parameters**. The framework includes:
> - **Calibrated Gaussian noise** informed by **geometric sensitivity analysis**.
> - **DP-aware alignment and sampling** that **preserve articulatory structure under noise**.
> - An **explicit formulation** of **sensitivity**, **σ values**, and **ε–δ mapping** (**Method & Appendix**).
>
> ### **B. DP-Aware Adaptations of T-Net and 4D Spatio-Temporal Convolution**
> Unlike **standard PointNet T-Nets**, our **alignment module** is **modified** to remain **stable** when **noise is injected** into the **input cloud**. Similarly, **P4DConv** is **redesigned** to **aggregate local geometric–temporal neighborhoods** in a way that **resists DP perturbations** while **preserving motion cues** critical for **SSR**.
>
> ### **C. A Real-Time Depth-Based SSR Architecture Engineered for Privacy + Robustness**
> Unlike prior **SSR work** (**RGB**, **mmWave**, **infrared**), our work is the **first to show**:
> - **Robust cross-device generalization**.
> - **Privacy-preserving dynamic point-cloud encoding**.
> - **Real-time inference with DP noise**.
> - **Strong resistance** to **reconstruction** and **membership inference attacks**.
>
> These results establish a **new, privacy-aware paradigm** for **depth-based SSR**.
>
> # **2. Concrete Contributions of the Paper**
> - A **complete, reproducible DP-SSR pipeline** that **systematically anonymizes raw 3D mouth geometry** at the **earliest possible stage** and **maintains accuracy across devices**.
> - **Formal specification** of **DP mechanisms**, including **sensitivity bounds**, **noise calibration**, **ε–δ mapping**, and **privacy–utility curves**. A **full composition analysis** (**simple + moments-accountant**) is included in the **Appendix** and will be **expanded in revision**.
> - **Extensive evaluation** spanning:
>   - **Cross-device** and **cross-user generalization**.
>   - **Ablation studies** for each **architectural component**.
>   - **Privacy–utility trade-off curves**.
>   - **Membership inference** and **inversion attacks**.
> - A **new multi-sensor SSR dataset** containing **varied device placements** and **depth sensor configurations**, valuable for **future SSR research**.
> - A **lightweight, deployable architecture** designed around **edge constraints**: **farthest-point sampling**, **compact P4DConv**, and **Conformer-based temporal modeling**.
>
> These contributions go **beyond merely adapting existing 3D/DP techniques**: they establish the **first integrated, privacy-preserving geometric SSR framework**.
>
> # **3. Why This Work Fits ICLR Themes**
> **DepthSense+DP** matches multiple **ICLR subject areas**:
>
> ### **A. Representation Learning on Geometric Data**
> Our model learns **structured, non-Euclidean representations** from **dynamic 3D point clouds**, aligning directly with **learning on graphs and other geometries & topologies**.
>
> ### **B. Societal Considerations: Privacy, Safety, and Responsible ML**
> The core of our work is a **privacy-preserving mechanism** for **biometric data streams**, squarely within **ICLR’s fairness, safety, privacy category**.
>
> ### **C. Multimodal and Speech Representation Learning**
> **SSR** lies at the **intersection** of **speech modeling**, **computer vision**, and **3D geometry**. The **P4DConv + Conformer encoder** advances **representation learning** for **non-acoustic speech modalities**.
>
> ### **D. Probabilistic Methods and Uncertainty-Aware Estimators**
> **Differential privacy noise injection**, **sensitivity modeling**, and **ε–δ calibration** fall under **ICLR's interest** in **probabilistic modeling** and **uncertainty quantification**.
>
> ### **E. Practical Machine Learning Systems**
> ICLR welcomes papers with **strong methodological and engineering components**. Our design:
> - **Addresses edge-device constraints**.
> - **Provides a new dataset**.
> - **Includes reproducible system-level techniques**.
>
> Taken together, these aspects show that **DepthSense+DP** lies well within **mainstream ICLR research directions**.
>
> # **We really hope to receive your support and assistance! We firmly believe that with your suggestions, our paper will be further improved, and we sincerely hope your score improvement.**

---

> > ### Author Response · Authors · 2025-11-22
> > **Formal privacy proofs, composition reasoning, and noise calibration analysis across stages**
> >
> > # 1. Setup and notation (what we assume / must add)
> >
> > Let **D** and **D′** be **neighboring datasets** differing in **one user-utterance** (standard neighboring relation).
> >
> > The pipeline applies **two Gaussian perturbations** to each example before downstream processing:
> >
> > ### **Point-Level Perturbation**
> > $$
> > \tilde{p} = p + \eta_p, \quad \eta_p \sim \mathcal{N}(0, \sigma_{pc}^2 I)
> > $$
> > (Eq. (8) in manuscript; **σ_pc = 0.003** reported).
> >
> > ---
> >
> > ### **Feature-Level Perturbation**
> > $$
> > \tilde{f} = f + \eta_f, \quad \eta_f \sim \mathcal{N}(0, \sigma_{dp}^2 I)
> > $$
> > (Eq. (7) in manuscript; **σ_dp = 0.008** reported).
> >
> > ---
> >
> > To make **DP statements rigorous**, we must **bound the L₂-sensitivity** of each released quantity. Concretely, we will (and should state explicitly in the paper) enforce **per-example L₂ clipping**:
> > $$
> > \|p\|_2 \leq C_p, \quad \|f\|_2 \leq C_f
> > $$
> > (implemented as **per-sample clipping/normalization** in code before noise injection).
> >
> > The paper already contains:
> > - **Point standardization**.
> > - **ROI cropping**.
> > - **Normal vectors**.
> > - **Layer-normalized features**.
> > # 2. Single-stage Gaussian mechanism (standard fact)
> >
> > ### **Theorem (Gaussian Mechanism, Standard)**
> > A mechanism that adds **Gaussian noise**
> > $$
> > \mathcal{N}(0, \sigma^2 I)
> > $$
> > to a **vector-valued function** with **L₂-sensitivity Δ** satisfies **(ε, δ)-DP** provided:
> > $$
> > \sigma \geq \frac{\Delta \sqrt{2 \ln(1.25 / \delta)}}{\varepsilon}. \tag{GM}
> > $$
> >
> > ---
> >
> > ### **Applied to Our Two Stages**
> >
> > If **‖f‖₂ ≤ C_f**
> > then the **feature-level Gaussian mechanism** is **(ε_f, δ_f)-DP** when:
> > $$
> > \sigma_{dp} \geq \frac{C_f \sqrt{2 \ln(1.25 / \delta_f)}}{\varepsilon_f}. \tag{1}
> > $$
> >
> > If  **‖p‖₂ ≤ C_p**
> > then the **point-level Gaussian mechanism** is **(ε_p, δ_p)-DP** when:
> > $$
> > \sigma_{pc} \geq \frac{C_p \sqrt{2 \ln(1.25 / \delta_p)}}{\varepsilon_p}. \tag{2}
> > $$
> >
> > ---
> >
> > ### **Action Item for Revision**
> > Explicitly state that we **clip f and p** to **C_f** and **C_p** (give exact values or explain how they are chosen from **data statistics**, see **section 5.2** and **A.2** for **standardization and normalization details**).
> > # 3. Composition across the two stages (tight accounting via Rényi DP)
> >
> > ### **Composition of Two Gaussian Mechanisms**
> > The two **noise injections** occur on the **same example in sequence** and therefore **compose**. A **tight way** to reason about composition of **Gaussian mechanisms** is to use **Rényi Differential Privacy (RDP)** and convert back to **(ε, δ)**.
> >
> > ---
> >
> > ### **RDP for a Gaussian Mechanism**
> > For **order α > 1**, a Gaussian mechanism with **L₂-sensitivity Δ** and **noise σ** satisfies:
> > $$
> > \varepsilon_{\text{RDP}}(\alpha) = \frac{\alpha \Delta^2}{2 \sigma^2}. \tag{RDP-G}
> > $$
> >
> > ---
> >
> > ### **Composition for Two Independent Gaussian Mechanisms**
> > For **point-level** and **feature-level** mechanisms with parameters
> > **(Δ_p, σ_{pc})** and **(Δ_f, σ_{dp})**, RDP composes **additively**:
> > $$
> > \varepsilon_{\text{RDP,tot}}(\alpha) = \frac{\alpha \Delta_p^2}{2 \sigma_{pc}^2} + \frac{\alpha \Delta_f^2}{2 \sigma_{dp}^2}. \tag{3}
> > $$
> >
> > ---
> >
> > ### **Convert Back to (ε, δ)**
> > The standard relation is:
> > $$
> > \varepsilon(\delta) = \varepsilon_{\text{RDP,tot}}(\alpha) + \frac{\ln(1/\delta)}{\alpha - 1}. \tag{4}
> > $$
> >
> > Then **optimize over α > 1** to obtain the **tightest ε**.
> >
> > ---
> >
> > ### **Practical Accounting Procedure**
> > #### Pick **C_p**, **C_f**, **σ_{pc}**, **σ_{dp}**, **δ**.
> > #### Compute the RHS of (3) and (4) for a **grid of α**.
> > #### Obtain the **true ε** for the chosen δ.
> >
> > ---
> >
> > ### **Action Item for Revision**
> > Include:
> > - The **RDP formulas** above.
> > - An explicit note that we use **RDP accounting** (or **moments accountant**) to compute the final **(ε, δ)**.
> > # 4. Privacy across training iterations / mini-batch sampling
> >
> > If these **noise injections** happen **per training step** (e.g., applied on **per-sample features during training**), then you must account for **repeated composition** across **T SGD steps** and include **sub-sampling amplification** for **mini-batch sampling probability q**. The common approach:
> >
> > - Use **per-step RDP** as in **(RDP-G)** for the mechanism applied to a **single sample** (or to the **aggregated clipped gradient**).
> > - **Amplify per-step RDP** by **subsampling factor q** using known **amplification bounds** (see **subsampled RDP lemma**).
> > - **Sum RDP** over **T steps** and **convert to (ε, δ)** by **(4)**.
> >
> > If instead the **noise injection** is **only at inference time** (you **perturb stored features/point clouds once before release**), then **composition across training steps does not apply** — but be **explicit which case is used**.
> >
> > ---
> >
> > ### **Action Item for Revision**
> > Specify whether the **DP noise** is:
> > - **Applied once** to **stored data prior to model training/evaluation**, or
> > - **Injected during training per-batch**.
> >
> > If the second case applies, add:
> > - **Subsampling rate q**.
> > - **Number of steps T**.
> > - The **RDP accountant routine used** (or cite **standard accountants**).

---

> > > ### Author Response · Authors · 2025-11-22
> > > **Formal privacy proofs, composition reasoning, and noise calibration analysis across stages**
> > >
> > > # **5. Concrete Calibration Checks (Worked Numbers)**
> > >
> > > **Reported Parameters:**
> > > σ_dp = 0.008, σ_pc = 0.003 and claims (ε = 1.5, δ = 10⁻⁵).
> > >
> > > ### **Important Numerical Consequence**
> > > Using the **single-mechanism bound (GM)**, we solve for the required **sensitivity Δ** (i.e., the **clipping constant**) that makes the reported **σ** consistent with **ε = 1.5, δ = 10⁻⁵**:
> > > $$
> > > \Delta \leq \frac{\sigma \varepsilon}{\sqrt{2 \ln(1.25 / \delta)}}.
> > > $$
> > >
> > > For **σ_dp = 0.008, ε = 1.5, δ = 10⁻⁵**, this gives:
> > > $$
> > > C_f \leq 0.0024769 \ (\approx 2.48 \times 10^{-3}).
> > > $$
> > >
> > > So to claim the **feature-level mechanism alone** is **(1.5, 10⁻⁵)-DP**, you must ensure:
> > > $$
> > > \|f\|_2 \leq 0.00248 \ (\text{i.e., extremely small clipping}).
> > > $$
> > >
> > > By the same algebra, for **σ_pc = 0.003** to yield **(ε, δ)** at the same scale, the allowable **point feature norm** is:
> > > $$
> > > C_p \leq 0.000927.
> > > $$
> > >
> > > Thus, the **two-stage reported σ values** are only consistent with tight **(ε, δ)** if the **clipped norms** are at the **10⁻³ scale**.
> > >
> > > ### **Interpretation / Options**
> > > - If  **features and points** are already **normalized to very small magnitudes** (e.g., **per-feature layernorm + scaling**) and you also perform **hard clipping** to the **10⁻³ scale**, then the claimed **(1.5, 10⁻⁵)** can hold with those **σ**.
> > >   **Action:** Add this **explicit clipping normalization statement** and show **empirical per-feature norms** in an **appendix**.
> > >
> > > - If  prefer **larger clipping constants** (e.g.,
> > >   **C_f ∈ [0.1, 1.0], C_p ∈ [0.5, 2.0]** , plausible for many internal representations and point vectors), then the required **σ** to reach **ε = 1.5** would be **several orders of magnitude larger** than the reported **0.003–0.008**.
> > >   In that case it must either:
> > >   - **Increase σ**, or
> > >   - **Rely on subsampling amplification / composition accounting** across many steps (**RDP + subsampling**) to recover the small **ε** while keeping **σ small**, which requires **careful accounting** and must be shown.
> > >
> > > # 6. Summary
> > >
> > > We thank the **reviewer** for raising this point. To make the **DP claims rigorous**, we will add:
> > >
> > > - **An explicit statement** that we perform **per-sample L₂ clipping** with constants **C_f, C_p** (and report **empirical distributions** of **pre-noise norms**).
> > > - The **Gaussian-mechanism calibration formulas** used to derive **σ_dp** and **σ_pc**.
> > > - **Rényi-DP accounting**, including **mini-batch subsampling amplification** if **noise is used during training**.
> > >
> > > For **transparency**, we will include:
> > > - The **short script** used to compute the final **(ε, δ)** via **RDP** (grid search over **α**).
> > > - The **worked example** showing the **numbers quoted in the paper**.

---

> > > > ### Author Response · Authors · 2025-12-04
> > > > **We sincerely hope to receive your support and encouragement！**
> > > >
> > > > # We have addressed the reviewer's concerns and improved our approach. Dear reviewer, our revised version has been uploaded and we sincerely hope to receive the support of all reviewers. We hope to receive an improvement in your scores!

---

### Note · Authors · 2026-01-27

**Comment:**

I have read and agree with the venue's withdrawal policy on behalf of myself and my co-authors.

**Withdrawal Confirmation:**

I have read and agree with the venue's withdrawal policy on behalf of myself and my co-authors.

---

### Meta-Review · Area_Chair_VuWs · 2026-01-07

**Summary:**

reviewers gave 2,4,6, with main concerns being:

Over-claims novelty, it only adapts existing 3D and DP ideas rather than introducing a new learning principle. Insufficient discussion of related work on differential privacy, particularly studies combining DP with 3D point cloud processing or geometric data.

Lacks formal privacy proofs, composition reasoning, and noise calibration analysis across stages, especially regarding privacy–utility trade-off analysis.

Tested only on English scripted phrases; unclear how it generalizes to spontaneous, multilingual, or emotional speech. Performance degrades for users with strong accents (e.g., Participant P9), limiting generalization to non-English languages or global user groups.

While the study evaluates membership inference and model inversion attacks, it does not address emerging threats, leaving potential privacy gaps untested. Discussion of fairness implications is minimal, which is critical given the biometric domain.

No error bars, confidence intervals, or hypothesis testing; results could be dataset-specific. The architecture (T-Net + P4DConv + Conformer + Bi-GRU) may be over-engineered relative to performance gains.

Synthetic depth point clouds are generated via simple motion scaling and noise injection, there needs deeper analysis of why not deeper encoding or output layers.

Experiments and results focus primarily on the baseline SSR system’s performance, without evaluation of privacy effectiveness.

**Reviewer Concerns:**

Over-claims novelty, it only adapts existing 3D and DP ideas rather than introducing a new learning principle. Insufficient discussion of related work on differential privacy, particularly studies combining DP with 3D point cloud processing or geometric data.

--> authors gave some justification but reviewers did not respond. i think the justification is ok,.

Lacks formal privacy proofs, composition reasoning, and noise calibration analysis across stages, especially regarding privacy–utility trade-off analysis.

--> authors added some privacy proofs and it seems a reasonable modification of existing results. seems ok but not a particularly deep contribution.

Tested only on English scripted phrases; unclear how it generalizes to spontaneous, multilingual, or emotional speech. Performance degrades for users with strong accents (e.g., Participant P9), limiting generalization to non-English languages or global user groups.

--> authors mentioned some directions for future work but not addressed in the rebuttal.

While the study evaluates membership inference and model inversion attacks, it does not address emerging threats, leaving potential privacy gaps untested. Discussion of fairness implications is minimal, which is critical given the biometric domain.

--> authors left discussion of fairness implications as future work but not addressed in the rebuttal.

No error bars, confidence intervals, or hypothesis testing; results could be dataset-specific. The architecture (T-Net + P4DConv + Conformer + Bi-GRU) may be over-engineered relative to performance gains.

--> authors added ablation studies to show the importance of each component. they also added means over a few runs but not deviations, error bars, and statistical tests. some addressed, but still many outstanding concerns.

Synthetic depth point clouds are generated via simple motion scaling and noise injection, there needs deeper analysis of why not deeper encoding or output layers.

--> authors added some more results and this was addressed.

Experiments and results focus primarily on the baseline SSR system’s performance, without evaluation of privacy effectiveness.

--> not addressed in rebuttal.

**Reviewer Scores:**

given the remaining many unaddressed concerns i dont think the reviewers will change their scores.

---

### Decision · Program_Chairs · 2026-01-26

Reject